# TRANSFORMERS WITH RL OR SFT PROVABLY LEARN SPARSE BOOLEAN FUNCTIONS, BUT DIFFERENTLY

## ABSTRACT

Transformers can acquire Chain-of-Thought (CoT) capabilities to solve complex reasoning tasks through fine-tuning. Reinforcement learning (RL) and supervised fine-tuning (SFT) are two primary approaches to this end, yet their underlying mechanisms and differences remain theoretically unclear. In this work, we examine these aspects specifically for learning $k$-sparse Boolean functions with a one-layer transformer and intermediate supervision that is akin to CoT. In particular, we consider $k$-sparse Boolean functions that can be recursively decomposed into fixed 2-sparse Boolean functions. We analyze the learning dynamics of fine-tuning the transformer via either RL or SFT with CoT to identify sufficient conditions for it to provably learn these functions. We verify that these conditions hold for three basic examples, including $k$-PARITY, $k$-AND, and $k$-OR, thus demonstrating the learnability of both approaches. Notably, we reveal that RL and SFT exhibit distinct learning behaviors: RL learns the whole CoT chain simultaneously, whereas SFT learns the CoT chain step-by-step. Overall, our findings provide theoretical insights into the underlying mechanisms of RL and SFT as well as how they differ in triggering the CoT capabilities of transformers.

## 1 INTRODUCTION

Large language models (LLMs), with the transformer architecture being their core building block, are remarkably successful across a wide range of tasks, in particular reasoning. LLMs excel in solving complex reasoning tasks by iteratively generating intermediate steps (Wei et al., 2022) — an intriguing approach known as Chain-of-Thought (CoT). Fine-tuning has been shown to be a powerful method to enhance efficient CoT generation in LLMs, which in turn improves the multi-step reasoning performance of LLMs significantly (Wei et al., 2022; Zelikman et al., 2022; Lightman et al., 2024).

A widely adopted approach for fine-tuning to generate CoT is supervised fine-tuning (SFT), where the transformers are trained to minimize a loss over pairs of inputs and labeled outputs. While straightforward, SFT is restricted by the demand of a large amount of labeled CoT data. As a result, fine-tuning approaches based on reinforcement learning (RL) (DeepSeek-AI et al., 2025; Ouyang et al., 2022; Bai et al., 2022; Christiano et al., 2023; Kumar et al., 2024) are increasingly prevalent. Instead of minimizing a loss over labeled CoT data, RL guides transformers to generate CoT to solve complex reasoning tasks by maximizing a reward function via policy gradient methods (Mnih et al., 2016; Schulman et al., 2017; DeepSeek-AI et al., 2025), which has shown significant potential for improving the reasoning capabilities of LLMs.

The remarkable success of fine-tuned transformers has spurred significant interest in understanding their underlying mechanisms. A large number of works have investigated various aspects such as expressivity (Li et al., 2024; Merrill & Sabharwal, 2024) and estimation error (Hu et al., 2024). For SFT, Kim & Suzuki (2025) studied the emergence of CoT by formalizing the mechanism specifically in the bit subset parity problem. They showed that a one-layer transformer fine-tuned via SFT with teacher-forcing provably learns the parity in a way analogous to CoT, providing promising theoretical insights into reasoning capabilities acquired by SFT. Nevertheless, the theoretical understandings of RL and of how SFT and RL differently enhance the reasoning capabilities of transformers are inadequate or even absent. This lack of theoretical understandings stands in stark

contrast to the success of fine-tuning via RL and the growing body of empirical observations for the comparison between RL and SFT, such as SFT memorizes but RL generalizes (Chu et al., 2025).

In this work, we take the first step towards addressing this gap. We study the problem of learning a broad class of functions (Bhattamishra et al., 2023)—$k$-*sparse Boolean functions* (Definition 2.1). We specifically focus on $k$-sparse Boolean functions that can be recursively decomposed into fixed 2-sparse Boolean functions, which includes the bit subset parity problem as an example. These functions are hard to be learned in an end-to-end manner but can be provably learned with intermediate supervision (Kim & Suzuki, 2025; Wies et al., 2023; Shamir, 2017; Hu et al., 2025) that is akin to CoT, and hence provide suitable case studies for the emergence of reasoning capabilities for transformers through fine-tuning. In particular, we examine the learning dynamics of fine-tuning via *RL optimized by vanilla policy gradient method* (Mnih et al., 2016), which has not been covered by prior works despite its importance, or via *SFT without teacher-forcing as well as augmented data*, which relaxes the constraints in Kim & Suzuki (2025). For theoretical tractability, we analyze a one-layer transformer, consisting of a positional encoding, a self-attention layer, and a feedforward layer. This transformer is iteratively applied to its own output to generate intermediate steps before arriving at the final answer. Our contributions are summarized below.

**Our contribution.**

- For $k$-sparse Boolean functions (Definition 2.1) which are recursively decomposable via a binary tree structure using a fixed 2-sparse Boolean function, we decompose the learning into multi-step reasoning tasks (Fig. 1a), allowing the one-layer transformer to be fine-tuned via either RL or SFT. The transformer learns the functions by generating intermediate steps, akin to CoT generation. We specifically study $k$-PARITY (Sec. 4.1), $k$-AND, and $k$-OR (Sec. 4.2) as examples.

- For fine-tuning via RL optimized by policy gradient, we establish sufficient conditions (Theorem 3.1) that guarantee the learnability of $k$-sparse Boolean functions for the transformer after *one gradient update*, indicating that the whole reasoning chain is learned simultaneously. In addition, we verify that these conditions can be satisfied by $k$-PARITY (Theorem 4.1), $k$-AND, and $k$-OR (Claim 4.2), validating that CoT enables the transformer to learn these sparse Boolean functions via RL.

- For SFT without teacher forcing or data augmentation, we prove that the transformer can also acquire the CoT generation capabilities to learn $k$-sparse Boolean functions under similar conditions as RL (Theorem 3.2), which are satisfied by $k$-PARITY (Claim 4.1), $k$-AND, and $k$-OR (Claim 4.2). In contrast to RL, the transformer naturally exhibits a step-wise learning behavior: it learns the reasoning chain *step-by-step* from the beginning, requiring one gradient update per step and therefore demanding a total number of updates equal to the chain's length.

**Organization.** We first present the problem setup in Sec. 2, including the definition of $k$-sparse Boolean functions (Sec. 2.1), the decomposition of them into sub-tasks, and the transformer architecture (Sec. 2.2). We then establish sufficient conditions for the transformer to learn $k$-sparse Boolean functions through fine-tuning via RL (Sec. 3.1) and SFT (Sec. 3.2) with CoT, providing an overview of the learnability. Finally, we discuss the learnability of $k$-PARITY (Sec. 4.1), $k$-AND, and $k$-OR (Sec. 4.2) in detail, proving that both RL and SFT enable the transformer to learn these functions. Throughout the paper, we use SFT to refer to SFT without teacher forcing.

**Related Works.** We discuss related works in Appendix A due to space limitation.

## 2 FRAMEWORK SETUP

**Notations.** For a vector $\boldsymbol{a} \in \mathbb{R}^d$, we use $a_j$ to denote its $j$-th component and $\|\boldsymbol{a}\|$ its $\ell_2$-norm. Applying a scalar function $g(\cdot)$ to $\boldsymbol{a}$ gives $g(\boldsymbol{a}) = (g(a_1), \ldots, g(a_d))^\top \in \mathbb{R}^d$. We use $\text{sign}(\cdot)$ for the sign function that takes the sign of each component of a vector. For a matrix $\boldsymbol{W} \in \mathbb{R}^{d_1 \times d_2}$, we use $W_{i,j}$ to denote its $i$-th row $j$-th column element. Given an integer $N$, we define $[N] := \{1, \ldots, N\}$. We define $\delta_{ij} = 1$ if $i = j$, otherwise $\delta_{ij} = 0$. For simplicity, we use the convention $\mathbf{x} = (x_1, \ldots, x_d) \in \{-1, +1\}^d$, instead of $\mathbf{x}^\top$, to represent the input and output sequences, which can be distinguished from a regular vector according to the context.

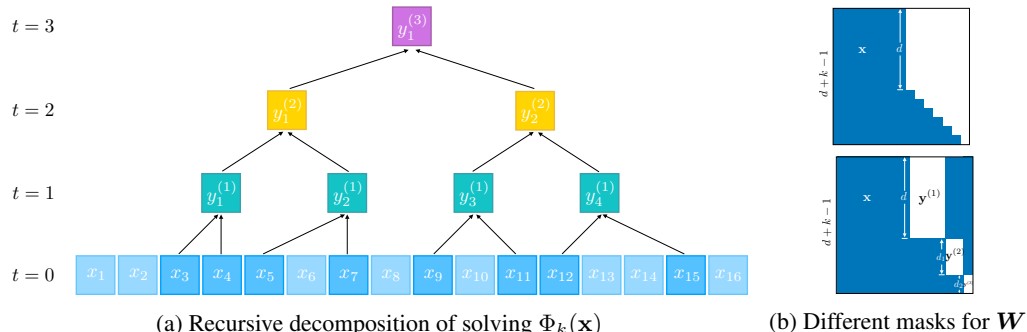

(a) Recursive decomposition of solving $\Phi_k(\mathbf{x})$       (b) Different masks for $\boldsymbol{W}$

Figure 1: **(a)** Recursive decomposition of learning a $k$-sparse Boolean function $\Phi_k(\mathbf{x})$ with a random set $B \subseteq [d]$ (shaded boxes in the lowest level) into solving sub-tasks by following a reasoning chain (bottom to top). Each level of the binary tree corresponds to a step of the reasoning chain, where each node in a level computes a 2-sparse Boolean function $\phi_2(\cdot, \cdot)$ over its two child nodes. **(b)** The self-attention weight $\boldsymbol{W}$ with different masks (blue entries are set as $-\infty$): **top:** the mask for $\mathbf{x}$ and causal mask; **bottom:** the mask for $\mathbf{x}$, causal mask, and pretrained mask.

## 2.1 SPARSE BOOLEAN FUNCTIONS AND RECURSIVE DECOMPOSITION

We first present the definition of the sparse Boolean functions that we attempt to learn.

**Definition 2.1** ($k$-sparse Boolean functions). *For a $d$-bit input sequence $\mathbf{x} \in \{+1, -1\}^d$ that follows the uniform distribution, let $B = \{i_1, i_2, \ldots, i_k\} \subseteq [d]$ with $2 \leq |B| = k \leq d$, then $\Phi_k : \{+1, -1\}^d \to \{\pm 1\}$ is a $k$-sparse Boolean function if it takes the input $\mathbf{x}$ and is determined by coordinates in $B$, i.e., $\Phi_k(\mathbf{x}) = \phi_k(x_{i_1}, x_{i_2}, \ldots, x_{i_k})$ for some fixed $\phi_k : \{-1, +1\}^k \to \{\pm 1\}$.*

In this paper, we only consider $k$-sparse Boolean functions that can be recursively decomposed into fixed 2-sparse Boolean functions $\phi_2$. Given sufficient training samples, the goal of learning is to predict $\Phi_k(\mathbf{x})$ for any test $\mathbf{x}$. Specifically, we will study three such $k$-sparse Boolean functions:

- $k$-PARITY (i.e., the bit subset parity): $\Phi_k^{\text{parity}}(\mathbf{x})$ equals $+1$ if the number of $-1$ of $\mathbf{x}$ in $B$ is even and equals $-1$ otherwise, i.e., $\Phi_k^{\text{parity}}(\mathbf{x}) = \prod_{i \in B} x_i$.

- $k$-AND: $\Phi_k^{\text{and}}(\mathbf{x})$ equals $+1$ if $x_i = 1$ for all coordinates $i \in B$ and equals $-1$ otherwise, i.e., $\Phi_k^{\text{and}}(\mathbf{x}) = 2 \prod_{i \in B} \left( \frac{x_i + 1}{2} \right) - 1$.

- $k$-OR: $\Phi_k^{\text{or}}(\mathbf{x})$ equals $+1$ if there is at least one $x_i = 1$ for the coordinates $i \in B$ and equals $-1$ otherwise, i.e., $\Phi_k^{\text{or}}(\mathbf{x}) = 1 - 2 \prod_{i \in B} \left( \frac{1 - x_i}{2} \right)$.

**Recursive Decomposition.** $k$-PARITY is known to be hard to learn with gradient-based methods in an end-to-end manner, as it has been proved that the gradient contains negligible information of the objective function (Shalev-Shwartz et al., 2017). To enable efficient learning, Wies et al. (2023); Kim & Suzuki (2025) decomposed $k$-PARITY into sub-tasks that only perform 2-parity computations, allowing the model to learn in a sequential manner. Inspired by them, we consider $k$-sparse Boolean functions $\Phi_k(\cdot)$ that can be recursively decomposed into fixed 2-sparse Boolean functions $\phi_2$. In particular, given $\Phi_k$, we assume $k = 2^T$ for an integer $T$, and recursively decompose the calculation of $\Phi_k(\cdot)$ into $T$ intermediate steps. This forms a length-$T$ reasoning chain in which each step $t \in [T]$ performs a series of fixed 2-sparse Boolean function calculations.

Formally, given an input $\mathbf{x} \in \{-1, +1\}^d$, a random subset $B \subseteq [d]$ with $|B| = k$, and a $k$-sparse Boolean function $\Phi_k(\mathbf{x})$, the decomposition can be expressed by a complete binary tree with $2k - 1$ nodes (Fig. 1a), where the lowest level has $k$ nodes $x_{i_j}$ for $i_j \in B$ and the $t$-th level has $d_t = k/2^t$ nodes and accounts for the $t$-th step of the reasoning chain. Additionally, each node in the $t$-th level depends on its two child nodes, demonstrating how $\Phi_k(\mathbf{x})$ is recursively solved: denoting the sequence at the $t$-th level as $\mathbf{y}^{(t)} = (y_1^{(t)}, \ldots, y_{d_t}^{(t)}) \in \{-1, +1\}^{d_t}$ and the initial input as $\mathbf{y}^{(0)} = \mathbf{x}$, let $\phi_2(z_1, z_2)$ be the fixed function that computes the 2-sparse Boolean function over $(z_1, z_2)$, then

$$\forall t \in [T], \ j \in [d_t] : \ y_j^{(t)} = \phi_2 \left( y_{i_1^j}^{(t-1)}, y_{i_2^j}^{(t-1)} \right), \tag{1}$$

where $i_1^j$ and $i_2^j$ are two child nodes of the $j$-th node of level $t$ and are named *relevant positions* as they determine $y_j^{(t)}$. The final answer is $\mathbf{y}_1^{(T)} = \phi_2(y_1^{(T-1)}, y_2^{(T-1)})$ such that $\mathbf{y}_1^{(T)} = \Phi_k(\mathbf{x})$.

## 2.2 PRETRAINED TRANSFORMER AND CHAIN-OF-THOUGHTS

For theoretical tractability, we investigate a one-layer transformer, consisting of the positional encoding, a self-attention layer, and a feedforward layer. Recall that we use the convention $\mathbf{x}$ for sequences. We will denote the number of tokens of $\mathbf{y}^{(t)}$ by $d_t = k/2^t$ for $t \in [T]$ with $d_0 = d$ and the total number of tokens of the sequence $(\mathbf{x}, \mathbf{y}^{(1)}, \ldots, \mathbf{y}^{(t)})$ by $N_t = \sum_{\tau=0}^{t} d_\tau$ and $N_{-1} = 0$.

**Positional encoding.** Specifically, given an input sequence $\mathbf{x}$ and the recursive decomposition of the $k$-sparse Boolean function $\Phi_k(\mathbf{x})$, we add tokens $\mathbf{y} = (\mathbf{y}^{(1)}, \ldots, \mathbf{y}^{(T)})$ to $\mathbf{x}$ which are all initialized as $\mathbf{0}$ and will be iteratively updated by the transformer to serve as intermediate steps of the reasoning chain (Fig. 1a). And we denote $\mathbf{z} = (\mathbf{x}, \mathbf{y}) \in \{-1, +1, 0\}^{d+k-1}$ for convenience. We then concatenate the positional encoding $\mathbf{e}_j \in \mathbb{R}^{d+k-1}$, the $j$-th standard basis vector of $\mathbb{R}^{d+k-1}$, to $z_j$ to form the complete encoding of the input sequence (with $\mathbf{x}$ and $\mathbf{y}$ explicitly written as components)

$$\boldsymbol{Z} = \begin{pmatrix} x_1 & \cdots & x_d & y_1^{(1)} & y_2^{(1)} & \cdots & y_1^{(T)} \\ \mathbf{e}_1 & \cdots & \mathbf{e}_d & \mathbf{e}_{d+1} & \mathbf{e}_{d+2} & \cdots & \mathbf{e}_{d+k-1} \end{pmatrix} \in \mathbb{R}^{(d+k) \times (d+k-1)}. \tag{2}$$

**Self-Attention.** A single-head self-attention layer (without residual connection) updates $\boldsymbol{Z}$ to $f^{\mathrm{SA}}(\boldsymbol{Z}) = \boldsymbol{W}_V \boldsymbol{Z}\, \mathrm{softmax}\left[(\boldsymbol{W}_K \boldsymbol{Z})^\top (\boldsymbol{W}_Q \boldsymbol{Z})\right] \in \mathbb{R}^{1 \times (d+k-1)}$, where $\boldsymbol{W}_V \in \mathbb{R}^{1 \times (d+k)}$ is the value matrix, $\boldsymbol{W}_K, \boldsymbol{W}_Q \in \mathbb{R}^{d' \times (d+k)}$ are the key and query matrices, respectively, and softmax is applied column-wise. In this paper, we simplify the self-attention by merging the key matrix and query matrix as a single matrix $\boldsymbol{W}_{KQ} := \boldsymbol{W}_K^\top \boldsymbol{W}_Q \in \mathbb{R}^{(d+k) \times (d+k)}$ to focus only on the positional encoding while setting $\boldsymbol{W}_V$ to preserve only the $\mathbf{x}$ component, namely $\boldsymbol{W}_{KQ} = \begin{pmatrix} 0 & \mathbf{0}_{1 \times (d+k-1)} \\ \mathbf{0}_{(d+k-1) \times 1} & \boldsymbol{W} \end{pmatrix}$ and $\boldsymbol{W}_V = \begin{pmatrix} 1 & \mathbf{0}_{1 \times (d+k-1)} \end{pmatrix}$. This has been a common choice in recent works due to its theoretical tractability (e.g., Kim & Suzuki, 2025; von Oswald et al., 2023; Zhang et al., 2023; Lyu et al., 2025; Huang et al., 2023).

**Feedforward layer.** The feedforward layer includes an activation function $\psi : [-1, 1] \rightarrow [0, 1]$ that applies to $f^{\mathrm{SA}}(\boldsymbol{Z})$ element-wise and the form of $\psi(\cdot)$ depends on that of the 2-sparse Boolean function $\phi_2(\cdot, \cdot)$. This is because $\psi([f^{\mathrm{SA}}(\boldsymbol{Z})]_j)$ is related to how the token at the position $j$ depends on its prior tokens, and such dependence is determined by $\phi_2(\cdot, \cdot)$.

**"Pretrained" transformer and its forward pass.** Taken together, the transformer $f(\cdot; \boldsymbol{W})$ takes into an input sequence $\mathbf{z} = (\mathbf{x}, \mathbf{y})$ and outputs $f(\mathbf{z}; \boldsymbol{W}) = \psi(f^{\mathrm{SA}}(\boldsymbol{Z}))$, where the causal mask $W_{i,j} = -\infty$ for $i \geq j$ is implicitly added and we only consider $j \geq d+1$ ($W_{i,j} = -\infty$ for any $j \leq d$) as only the reasoning chain $\mathbf{y}$ in the input $\mathbf{z}$ will be iteratively generated by the transformer. Furthermore, since we will consider fine-tuning via RL, the output $f(\mathbf{z}; \boldsymbol{W})$ should correspond to the next action for the current state[1], which is related to an intermediate reasoning step in Fig. 1a. Thus, it is natural for each action of RL to account for a reasoning step $\mathbf{y}^{(t)}$ for the reasoning chain $\mathbf{y}$. In light of that $\mathbf{y}^{(t)}$ only depends on $\mathbf{y}^{(t-1)}$ for any $t \in [T]$ (Fig. 1a) and tokens in the same sequence $\mathbf{y}^{(t)}$ are independent of each other, aside from the causal masks, we impose "pretrained masks" to $\boldsymbol{W}$ that explicitly exploit such dependence to control the error propagation of the reasoning chain as follows (Fig. 1b): given $t \in [T], \forall j \in [N_{t-1} + 1, N_t]$,

$$W_{i,j} = -\infty \quad \text{when} \quad \begin{cases} i > d, & \text{if } t = 1, \\ i > N_{t-1} \text{ or } i \leq N_{t-2}, & \text{otherwise}. \end{cases} \tag{3}$$

We heuristically view the transformer $f(\cdot; \boldsymbol{W})$ with such masks as our "pretrained transformer", which is a result of masking and is distinct from the conventional pretraining of transformers. $f(\cdot; \boldsymbol{W})$ takes an input sequence $\mathbf{z} = (\mathbf{x}, \ldots, \mathbf{y}^{(t-1)}, \mathbf{y}^{(t)}, \ldots, \mathbf{y}^{(T)})$ and gives

$$[f(\mathbf{z}; \boldsymbol{W})]_{N_{t-1} + l^{(t)}} = \psi\left(\xi_{l^{(t)}}\right), \quad \xi_{l^{(t)}} := \sum_{i=1}^{d_{t-1}} y_i^{(t-1)} \sigma_{N_{t-1} + l^{(t)}}^{N_{t-2} + i}, \quad t \in [T] \tag{4}$$

---

[1]This will be discussed in detail in Sec. 3.1.

Figure 2: The pretrained transformer iteratively uses its output to solve $\Phi_k(\mathbf{x})$ in a CoT manner.

at the $(N_{t-1}+l^{(t)})$-th position of the output to update the $l^{(t)} \in [d_t]$ token of the reasoning sequence $\mathbf{y}^{(t)}$, where we define the attention score as $\sigma_j^i := \exp(W_{i,j})/\sum_m \exp(W_{m,j})$ which determines the positions that the current token attends to. Intuitively, $\xi_{l^{(t)}}$ indicates how tokens of $\mathbf{y}^{(t-1)}$ contribute to the calculation of $y_{l^{(t)}}^{(t)}$.

**Chain-of-Thoughts.** We now indicate how the pretrained transformer iteratively exploits its own output to generate intermediate steps to solve the $k$-sparse Boolean function, akin to CoT (Fig. 2). Specifically, given the input sequence $\mathbf{z} = (\mathbf{x}, \mathbf{y}^{(1)}, \dots, \mathbf{y}^{(t)}, \dots, \mathbf{y}^{(T)})$ with all intermediate reasoning steps $\mathbf{y}^{(t)}$ initialized as $\mathbf{0}$, starting from $t = 1$ to $T$, the pretrained transformer $f(\cdot; \boldsymbol{W})$ iteratively generates the token $\hat{y}_{l^{(t)}}^{(t)}$ of $\hat{\mathbf{y}}^{(t)}$ from $l^{(t)} = 1$ to $l^{(t)} = d_t$ according to Eq. (4), e.g., by sampling from $f(\hat{\mathbf{z}}; \boldsymbol{W})$ if it is a distribution, and applies the generated token to update $\hat{\mathbf{z}} = (\mathbf{x}, \dots, \hat{\mathbf{y}}^{(t-1)}, \hat{y}_1^{(t)}, \dots, \hat{y}_{l^{(t)}-1}^{(t)}, y_{l^{(t)}}^{(t)}, \dots, \mathbf{y}^{(T)})$. This is repeated until all steps of the reasoning sequence $\hat{\mathbf{z}} = (\mathbf{x}, \dots, \hat{\mathbf{y}}^{(t-1)}, \hat{\mathbf{y}}^{(t)}, \dots, \hat{\mathbf{y}}^{(T)})$ are updated, and $\hat{\mathbf{y}}^{(T)}$ will be the final answer.

# 3 SUFFICIENT CONDITIONS FOR PROVABLE LEARNING

We consider fine-tuning the transformer (Sec. 2.2) to learn the $k$-sparse Boolean function $\Phi_k(\cdot)$ via RL or SFT. For convenience, we denote $\mathbf{y}^{(:t)} := (\mathbf{y}^{(1)}, \dots, \mathbf{y}^{(t)})$ and $\mathbf{y}^{(t:)} := (\mathbf{y}^{(t)}, \dots, \mathbf{y}^{(T)})$.

## 3.1 RL FINE-TUNING OF THE PRETRAINED TRANSFORMER

**Problem setup.** We view each input $\mathbf{z} = (\mathbf{x}, \mathbf{y})$ as a state and each output of the transformer as a distribution of the token for the action $\mathbf{y}^{(t)}$ (a reasoning step of the CoT chain) under the state $\mathbf{z}$, i.e., $p_{\boldsymbol{W}}(y_j^{(t)} = 1|\mathbf{z}) = [f(\mathbf{z}; \boldsymbol{W})]_{N_{t-1}+j}$. The formulation of the pretrained transformer Eq. (4) implies that the generation of $\mathbf{y}^{(t)}$ only depends on $\mathbf{y}^{(t-1)}$, allowing us to write $p_{\boldsymbol{W}}(y_j^{(t)} = 1|\mathbf{z}) = p_{\boldsymbol{W}}(y_j^{(t)} = 1|\mathbf{y}^{(:t-1)}) = p_{\boldsymbol{W}}(y_j^{(t)} = 1|\mathbf{y}^{(t-1)})$. The distribution over one whole CoT reasoning chain $\mathbf{y}$ conditioned on the initial state $\mathbf{x}$ is defined by

$$p_{\boldsymbol{W}}(\mathbf{y}|\mathbf{x}) = \prod_{t=1}^{T} p_{\boldsymbol{W}}(\mathbf{y}^{(t)}|\mathbf{y}^{(t-1)}), \tag{5}$$

where the sequence $\mathbf{y}^{(t)}$ can be heuristically viewed as an action as well as a state. Then, starting from the initial state $\mathbf{z} = (\mathbf{x}, \mathbf{y})$ with $\mathbf{y}$ initialized as 0, each sampled action $\hat{\mathbf{y}}^{(t)}$ updates the state $\hat{\mathbf{z}} = (\mathbf{x}, \hat{\mathbf{y}}^{(:t-1)}, \mathbf{y}^{(t:)})$ to $(\mathbf{x}, \hat{\mathbf{y}}^{(:t)}, \mathbf{y}^{(t+1:)})$ in a CoT manner (Fig. 2). RL now aims to maximize the *expected reward* $\mathcal{R}(\boldsymbol{W})$ w.r.t the transformer $f(\cdot; \boldsymbol{W})$:

$$\mathcal{R}(\boldsymbol{W}) := \mathop{\mathbb{E}}_{\mathbf{x}}[R(\mathbf{x}; \boldsymbol{W})], \quad R(\mathbf{x}; \boldsymbol{W}) := \mathbb{E}_{\mathbf{y} \sim p_{\boldsymbol{W}}(\cdot|\mathbf{x})}[r(\mathbf{x}, \mathbf{y})], \quad r(\mathbf{x}, \mathbf{y}) := \sum_{t=1}^{T} r_t\left(\mathbf{y}^{(t)}, \mathbf{y}^{(t-1)}\right),$$

where $r : \{-1, +1\}^{d+k-1} \to [-1, 1]$ is a reward function while the $t$-th step reward is defined by

$$r_t\left(\mathbf{y}^{(t)}, \mathbf{y}^{(t-1)}\right) := \frac{1}{k-1}\sum_{j=1}^{d_t} y_j^{(t)}\bar{y}_j^{(t)}, \quad \bar{y}_j^{(t)} = \phi_2\left(y_{i_1^j}^{(t-1)}, y_{i_2^j}^{(t-1)}\right), \tag{6}$$

with $i_1^j$ and $i_2^j$ being the two child nodes of the $j$-th node of level $t$ in Fig. 1a, i.e., $\bar{y}_j^{(t)}$ is the correct label of $y_j^{(t)}$ given $\mathbf{y}^{(:t-1)}$. Given a sparse Boolean function $\Phi(\cdot)$ and its recursive decomposition as in Def. 2.1, if we let $\boldsymbol{W}^\star = \arg\max_{\boldsymbol{W}} \mathcal{R}(\boldsymbol{W})$ maximize the expected reward such that $\mathcal{R}(\boldsymbol{W}^\star) = 1$, then the transformer $f(\cdot; \boldsymbol{W}^\star)$ exhibits a unique self-attention score that only attends to the relevant positions and ignore all irrelevant ones, i.e.,

$$\forall t \in [T], l^{(t)} \in [d_t]: \ [\mathrm{softmax}(\boldsymbol{W}^\star)]_{N_{t-1}+l^{(t)}}^{N_{t-2}+p} = \begin{cases} \frac{1}{2}, & \text{if } p \in \{i_1^{l^{(t)}}, i_2^{l^{(t)}}\}, \\ 0, & \text{otherwise.} \end{cases}$$

We consider RL optimized by policy gradient descent, which employs the gradient of the expected reward that is computed by (Lem. 1)

$$\nabla_{\boldsymbol{W}} R(\mathbf{x}; \boldsymbol{W}) = \mathbb{E}_{\mathbf{y} \sim p_{\boldsymbol{W}}(\cdot|\mathbf{x})} \left[ \sum_{t=1}^{T} \nabla_{\boldsymbol{W}} \ln p_{\boldsymbol{W}} \left( \mathbf{y}^{(t)} | \mathbf{y}^{(t-1)} \right) \sum_{\tau=t}^{T} r_\tau \left( \mathbf{y}^{(\tau)}, \mathbf{y}^{(\tau-1)} \right) \right]. \quad (7)$$

Furthermore, when optimizing $p_{\boldsymbol{W}}\left(\mathbf{y}^{(t)}|\mathbf{y}^{(t-1)}\right)$, the $t$-th step reward $r_t$ is considerably more important than other $r_{t'}$ with $t \neq t'$ (e.g., the model can achieve high $r_t$ even when it fails at later steps), as the correctness of a token $y_j^{(t)}$ generated in the current action $t$ only depends on its two child nodes $y_{i_1^j}^{(t-1)}$ and $y_{i_2^j}^{(t-1)}$ (Fig. 1a) generated in its last step. Thus we consider optimizing $\max_{\boldsymbol{W}} \mathcal{R}(\boldsymbol{W})$ with the policy gradient below, which is equivalent to optimizing RL with immediate reward,

$$\nabla_{\boldsymbol{W}} R(\mathbf{x}; \boldsymbol{W}) = \mathbb{E}_{\mathbf{y} \sim p_{\boldsymbol{W}}(\cdot|\mathbf{x})} \left[ \sum_{t=1}^{T} \nabla_{\boldsymbol{W}} \ln p_{\boldsymbol{W}} \left( \mathbf{y}^{(t)} | \mathbf{y}^{(t-1)} \right) r_t \left( \mathbf{y}^{(t)}, \mathbf{y}^{(t-1)} \right) \right]. \quad (8)$$

**Intuitive analysis.** During policy gradient training, $f(\cdot; \boldsymbol{W})$ generates a large number of trajectories (reasoning sequences) and receives an immediate reward for each action it takes. Thus, for the $t$-th action $\mathbf{y}^{(t)}$, as long as the policy gradient Eq. (8) contains sufficient information to tell the relevant positions (the child nodes of a token of $\mathbf{y}^{(t)}$) from the irrelevant ones (other nodes), $f(\cdot; \boldsymbol{W})$ can be optimized to obtain higher $t$-th step reward $r_t$. Interestingly, such separation of the policy gradient can be shown to be equivalent to the separation of the *critical gradient component*[2] (Lem. 2)

$$\textbf{Critical gradient component:} \ \gamma_{l^{(t)}}^j(\mathbf{y}^{(t-1)}) := \frac{2}{k-1} \psi'(\xi_{l^{(t)}}) \phi_2\left(y_{i_1^{l^{(t)}}}^{(t-1)}, y_{i_2^{l^{(t)}}}^{(t-1)}\right) y_j^{(t-1)} \quad (9)$$

which is determined by the activation function $\psi(\cdot)$ and the 2-sparse Boolean function $\phi_2(\cdot, \cdot)$ given $t \in [T]$ and $l^{(t)} \in [d_t]$. In addition, the $t$-th step reward $r_t$ does not depend on rewards of other steps according to the form of the output of the transformer Eq. (4). As a result, rewards for different steps can be optimized simultaneously under the separation of $\gamma_{l^{(t)}}^j$ for all tokens, which guarantees the learnability.

Below we present a more accurate characterization for optimizing $\max_{\boldsymbol{W}} \mathcal{R}(\boldsymbol{W})$ with the sign of the policy gradient Eq. (8) as it will have positive values at relevant positions and negative values at irrelevant ones under the separation condition.

**Theorem 3.1** (Learnability of $k$-sparse Boolean functions via RL)**.** *Given integers $d \geq k \geq 2$, consider a $k$-sparse Boolean function $\Phi_k(\cdot)$ with any subset $B \subseteq [d]$ as in Def. 2.1. Let $\boldsymbol{W}(0) = \mathbf{1}$ be the initialization and let $\boldsymbol{W}^\star = \arg\max_{\boldsymbol{W}} \mathcal{R}(\boldsymbol{W})$ be the optimal parameter that solves $\max_{\boldsymbol{W}} \mathcal{R}(\boldsymbol{W})$. Set learning rate $\eta = \Omega\left(\ln(d/\epsilon)\right)$ for any $\epsilon > 0$. If the separation of the critical gradient component $\gamma_{l^{(t)}}^p$ is satisfied for $\forall t \in [T], l^{(t)} \in [d_t]$ and $\forall p \in \{i_1^{l^{(t)}}, i_2^{l^{(t)}}\}$, $p' \in [d_{t-1}] \backslash \{i_1^{l^{(t)}}, i_2^{l^{(t)}}\}:$*

$$\mathbb{E}_{\mathbf{x}, \mathbf{y}^{(:t-1)} \sim p_{\boldsymbol{W}}(\cdot|\mathbf{x})} \left[ \gamma_{l^{(t)}}^p(\mathbf{y}^{(t-1)}) - \gamma_{l^{(t)}}^{p'}(\mathbf{y}^{(t-1)}) \right] > 0 \quad (10)$$

*($p$ is a child node of $y_{l^{(t)}}^{(t)}$ while $p'$ is not), then fine-tuning the transformer $f(\cdot; \boldsymbol{W})$ via RL optimized by the sign of the policy gradient Eq. (8) after one update $\boldsymbol{W}(1) = \boldsymbol{W}(0) + \eta\,\mathrm{sign}\left(\nabla_{\boldsymbol{W}} \mathcal{R}(\mathbf{x}; \boldsymbol{W})\right)$ achieves $\|\mathrm{softmax}(\boldsymbol{W}(1)) - \mathrm{softmax}(\boldsymbol{W}^\star)\|_1 \leq \epsilon$.*

---

[2]This name is because the policy gradient can be expressed by the critical gradient component, see Lem. 2.

To the best of our knowledge, Thm. 3.1 studies the learnability of $k$-sparse Boolean functions for transformers with CoT through fine-tuning via RL for the first time. The learnability is guaranteed by the *separation of the critical gradient component* Eq. (9). Thm. 3.1 reveals the benefit of RL— learning the whole CoT chain in one-update without teacher-forcing or any additional training data.

**Immediate reward versus final reward.** Thm. 3.1 is established for RL with immediate reward, as the recursive decomposition (Fig. 1a) inherently demands the verification of sequential steps. As a comparison, another common approach is RL with final reward $r^{\mathrm{F}} = \mathbf{y}^{(T)} \Phi_k(\mathbf{x})$ that is provided only when the final answer is finalized (DeepSeek-AI et al., 2025). This approach admits a different objective $\max_{\boldsymbol{W}} \mathcal{R}^{\mathrm{F}}(\boldsymbol{W}) := \max_{\boldsymbol{W}} \mathbb{E}_{\mathbf{x}, \mathbf{y} \sim p_{\boldsymbol{W}}(\cdot|\mathbf{x})}[r^{\mathrm{F}}(\mathbf{x}, \mathbf{y})]$. Below we show its hardness in contrast to the learnability of RL with immediate reward inspired by Shalev-Shwartz et al. (2017).

**Proposition 3.1** (Hardness of RL with final reward). *Let $\mathscr{H}$ be a class of functions $h :$ $\{-1, +1\}^d \to \{-1, +1\}$ such that $\mathbb{E}_{\mathbf{x}}[h(\mathbf{x})h'(\mathbf{x})] = 0$ for any two distinct $h, h' \in \mathscr{H}$. Then for the model $p_{\boldsymbol{W}}(\cdot|\mathbf{x})$ with bounded gradient of the final output $\mathbb{E}_{\mathbf{x}}[\|\nabla_{\boldsymbol{W}} \mathbb{E}_{\mathbf{y} \sim p_{\boldsymbol{W}}(\cdot|\mathbf{x})}[\mathbf{y}^{(T)}]\|^2] \leq M$ and the objective of RL with final reward $\max_{\boldsymbol{W}} \mathcal{R}_h^{\mathrm{F}}(\boldsymbol{W}) = \mathbb{E}_{\mathbf{x}, \mathbf{y} \sim p_{\boldsymbol{W}}(\cdot|\mathbf{x})}[r_h^{\mathrm{F}}(\mathbf{x}, \mathbf{y})]$ where $r_h^{\mathrm{F}}(\mathbf{x}, \mathbf{y}) = \mathbf{y}^{(T)} h(\mathbf{x})$, the variance of the policy gradient $\nabla_{\boldsymbol{W}} \mathcal{R}_h^{\mathrm{F}}(\boldsymbol{W})$ is bounded as*

$$\mathrm{Var}(\mathscr{H}; \boldsymbol{W}) := \mathbb{E}_{h \in \mathscr{H}} \left[ \left\| \nabla_{\boldsymbol{W}} \mathcal{R}_h^{\mathrm{F}}(\boldsymbol{W}) - \mathbb{E}_{h' \in \mathscr{H}}[\nabla_{\boldsymbol{W}} \mathcal{R}_{h'}^{\mathrm{F}}(\boldsymbol{W})] \right\|^2 \right] \leq \frac{M}{|\mathscr{H}|}. \quad (11)$$

We consider $k$-sparse parity. Suppose that $\mathscr{H}$ is the collection of all possible sparse parity functions with $|\mathscr{H}| = 2^d$, which satisfies the condition of Prop. 3.1 (Shalev-Shwartz et al., 2017), and our target function is uniformly chosen from $\mathscr{H}$, then $\mathrm{Var}(\mathscr{H}; \boldsymbol{W})$ is exponentially small in $d$, indicating that the signal of target function contained in the policy gradient is drowned out by noise. This leads to the hardness of learning relying on such gradient, demonstrating the benefit of immediate reward.

## 3.2 SFT of the Pretrained Transformer

**Problem setup.** SFT is a straightforward approach that aims to minimize a loss over pairs of the labeled sequence and generated output. For each input $\mathbf{z} = (\mathbf{x}, \mathbf{0})$ with the CoT chain $\mathbf{y}$ initialized as $\mathbf{0}$, we let $\tilde{\mathbf{z}} = (\mathbf{x}, \tilde{\mathbf{y}}) = (\mathbf{x}, \tilde{\mathbf{y}}^{(1)}, \ldots, \tilde{\mathbf{y}}^{(T)})$ be the labeled sequence where $\tilde{\mathbf{y}}$ is the ground-truth CoT chain for solving $\Phi_k(\mathbf{x})$. Following the CoT generation discussed in Fig. 2, given the input sequence $\mathbf{z} = (\mathbf{x}, \mathbf{0})$, the transformer solves $\Phi_k(\mathbf{x})$ by iteratively applying its output to generate

$$\hat{y}_{l^{(t)}}^{(t)} = \mathrm{sign}\left(\hat{q}_{l^{(t)}}^{(t)}\right), \quad \hat{q}_{l^{(t)}}^{(t)} := 2[f(\hat{\mathbf{z}}; \boldsymbol{W})]_{N_{t-1}+l^{(t)}} - 1 \quad (12)$$

and update $\hat{\mathbf{z}} = (\mathbf{x}, \hat{\mathbf{y}}^{(:t-1)}, \hat{y}_1^{(t)}, \ldots, \hat{y}_{l^{(t)}-1}^{(t)}, y_{l^{(t)}}^{(t)}, \ldots, \mathbf{y}^{(t+1:)})$ from $l^{(t)} = 1$ to $d_t$ and from $t = 0$ to $T$, until the whole CoT chain $\hat{\mathbf{y}}$ is generated such that $\hat{\mathbf{z}} = (\mathbf{x}, \hat{\mathbf{y}})$. The labeled sequence and the generated output pair is $(\tilde{\mathbf{z}}, \hat{\mathbf{z}})$. We use the hinge-loss $\ell(\hat{a}, a) := \max(0, 1 - \hat{a}a)$ for $a \in \{-1, +1\}$ and $\hat{a} \in \mathbb{R}$, then the population loss $\mathcal{L}(\boldsymbol{W})$ of our problem will be ($\hat{q}_j^{(t)}$ can be viewed as a score)

**Population loss:** $\mathcal{L}(\boldsymbol{W}) := \mathbb{E}_{\mathbf{x}} \left[ \sum_{t=1}^{T} L_t(\mathbf{x}, \boldsymbol{W}) \right], \quad L_t(\mathbf{x}, \boldsymbol{W}) := \frac{1}{k-1} \sum_{j=1}^{d_t} \ell\left(\hat{q}_j^{(t)}, \tilde{y}_j^{(t)}\right).$ (13)

The objective of SFT is $\min_{\boldsymbol{W}} \mathcal{L}(\boldsymbol{W})$ by, e.g., gradient descent. Similar to the case of RL, $\boldsymbol{W}^{\star} = \arg\min_{\boldsymbol{W}} \mathcal{L}(\boldsymbol{W})$ is uniquely determined given a sparse Boolean function $\Phi(\cdot)$ and its recursive decomposition as in Def. 2.1, with $\mathcal{L}(\boldsymbol{W}^{\star}) = 0$. Specifically, the transformer $f(\cdot; \boldsymbol{W}^{\star})$ also exhibits a self-attention score that only attends to the relevant positions and ignore all irrelevant ones, i.e.,

$$\forall t \in [T], l^{(t)} \in [d_t] : \ [\mathrm{softmax}(\boldsymbol{W}^{\star})]_{N_{t-1}+l^{(t)}}^{N_{t-2}+p} = \begin{cases} \frac{1}{2}, & \text{if } p \in \{i_1^{l^{(t)}}, i_2^{l^{(t)}}\}, \\ 0, & \text{otherwise.} \end{cases}$$

We highlight that our approach is different from the teacher-forcing technique used in prior works (Wies et al., 2023; Kim & Suzuki, 2025), although both approaches employ intermediate steps of the CoT chain as supervision. Specifically, the input sequence for the generation of CoT tokens Eq. (12) is $\hat{\mathbf{z}}$ whose intermediate tokens are all generated by the transformer in a natural auto-regressive manner, while the teacher-forcing technique employs the ground-truth sequence $\tilde{\mathbf{z}}$ as input sequence to generate each token of the CoT. As a result, SFT with teacher forcing has a mismatch between training and inference, and we consider SFT without teacher forcing.

**Intuitive analysis.** SFT admits a fundamentally distinct training objective compared to RL: the transformer generates a whole CoT chain following a natural auto-regressive way and compares it with the ground-truth label to minimize the hinge loss. Thus, the ground-truth label of later CoT steps can be used to minimize the loss only if the prior steps are generated correctly (Fig. 1a). This suggests learning dynamics driven by induction: **(1)** suppose that at the first gradient update SFT only considers $\min_{\boldsymbol{W}} \mathbb{E}_{\mathbf{x}}[L_1(\mathbf{x}, \boldsymbol{W})]$ to learn the first step of the CoT chain $\mathbf{y}^{(1)}$, then its last step (i.e., $\mathbf{x}$) is correct (as $\mathbf{x}$ itself is the ground-truth), thus the ground-truth label $\tilde{\mathbf{y}}^{(1)}$ can be faithfully used; **(2)** similar to the case for RL, as long as the gradient $\nabla_{\boldsymbol{W}} \mathbb{E}_{\mathbf{x}}[L_1(\mathbf{x}, \boldsymbol{W})]$ can sufficiently separate the relevant positions from irrelevant ones, the population loss of the first step $\mathbb{E}_{\mathbf{x}}[L_1(\mathbf{x}, \boldsymbol{W})]$ can be minimized such that the generated $\hat{\mathbf{y}}^{(1)}$ can approximate the ground-truth $\tilde{\mathbf{y}}^{(1)}$ after one-gradient update; **(3)** now at the second update, if SFT only considers $\mathbf{y}^{(2)}$, then its last step $\hat{\mathbf{y}}^{(1)}$ is correctly generated and the ground-truth label $\tilde{\mathbf{y}}^{(2)}$ can be used as in the learning of $\mathbf{y}^{(1)}$; and **(4)** repeating this argument suggests that one gradient update solves one step $\mathbf{y}^{(t)}$, and thus $T$ gradient updates solve the whole CoT chain.

Below we present a more accurate characterization, which also depends on the critical gradient component Eq. (9) yet in a slightly different way. Our analysis reveals that it is sufficient and necessary for SFT to take $T$ steps to learn the whole CoT chain, requiring one gradient update per CoT step. In addition, we will show that this stepwise learning naturally emerges from SFT in transformer— it is an intrinsic property that does not rely on data augmentation with the corresponding filter or curriculum learning. To be consistent with that for RL, we use sign gradient descent, which can be viewed as a special version of Adam (Kingma & Ba, 2017).

**Theorem 3.2** (Learnability of $k$-sparse Boolean functions via SFT)**.** *Given integers $d \geq k \geq 2$, consider a $k$-sparse Boolean function $\Phi_k(\cdot)$ with any subset $B \in [d]$ as in Def. 2.1. Let $\boldsymbol{W}(0) = \mathbf{1}$ be the initialization and let $\boldsymbol{W}^{\star} = \arg\min_{\boldsymbol{W}} \mathcal{L}(\boldsymbol{W})$ be the optimal parameter that solves $\min_{\boldsymbol{W}} \mathcal{L}(\boldsymbol{W})$. Set learning rate $\eta = \Omega\left(\ln(d/\epsilon)\right)$ for any $\epsilon > 0$. Let the transformer $f(\cdot; \boldsymbol{W})$ be fine-tuned via SFT by running sign gradient descent $\boldsymbol{W}(s+1) = \boldsymbol{W}(s) - \eta \operatorname{sign}\left(\nabla_{\boldsymbol{W}} \mathcal{L}(\boldsymbol{W}(s))\right)$. If the separation of the critical gradient component is satisfied for any $l^{(t)} \in [d_t]$ and any $t \in [T]$ in the sense that*

$$\forall p \in \{i_1^{l^{(t)}}, i_2^{l^{(t)}}\}, p' \in [d_{t-1}]\setminus\{i_1^{l^{(t)}}, i_2^{l^{(t)}}\} : \ \mathbb{E}_{\mathbf{x}}\left[\gamma_{l^{(t)}}^{p}(\tilde{\mathbf{y}}^{(t-1)}) - \gamma_{l^{(t)}}^{p'}(\tilde{\mathbf{y}}^{(t-1)})\right] > 0 \quad (14)$$

*where $\tilde{\mathbf{y}}^{(t-1)}$ is the ground-truth label of $\mathbf{y}^{(t-1)}$ given an input $\mathbf{x}$, then running sign gradient descent for $T$ iterations achieves $\|\operatorname{softmax}(\boldsymbol{W}(T)) - \operatorname{softmax}(\boldsymbol{W}^{\star})\|_1 \leq \epsilon$.*

## 3.3 SUMMARY OF THE LEARNABILITY

Thm. 3.1 and 3.2 show that both RL and SFT enable the pretrained transformer to acquire CoT generation ability to learn $k$-sparse Boolean functions under the separation of $\gamma_{l^{(t)}}^{p}$ between the case when $p$ is a child node of $l^{(t)}$ and that when $p$ is not. Given $t \in [T]$ and $l^{(t)} \in [d_t]$, when generating the $l^{(t)}$-th token of the reasoning sequence $\mathbf{y}^{(t)}$, it is crucial for the self-attention to correctly focus on the relevant positions $i_1^{l^{(t)}}$ and $i_2^{l^{(t)}}$ (the child nodes of $l^{(t)}$ (Fig. 1a)) such that the sub-task Eq. (1) can be faithfully solved by the transformer. Our established conditions ensure that only attention scores of these relevant positions will be increased while that of all irrelevant positions will be decreased during the sign gradient training, hence, the learnability of the transformer is obtained.

Notably, while the condition for fine-tuning via RL and that via SFT are similar, the transformer exhibits distinct learning behaviors for these two approaches due to their different training objectives: RL allows the transformer to learn the whole CoT chain simultaneously, as it can receive an immediate reward for each reasoning step $\mathbf{y}^{(t)}$ during training regardless of the learning of its prior steps, and thus one-update is sufficient; as a comparison, the transformer learns the reasoning chain *step-by-step* via SFT and demands $T$-updates for the whole chain. We discuss the intuition behind this distinction. On one hand, the ground-truth labels of all steps of the CoT are fixed and determined by the input $\mathbf{x}$; on the other hand, the generation of later reasoning steps by transformers depends on the prior generated steps. If the prior steps are not correctly generated by the transformer, then it cannot approximate ground-truth labels of later steps, since these labels are obtained from the correct prior steps which the transformer cannot generate. As a result, SFT must ensure the learning of prior steps before proceeding to later steps, leading to a step-wise learning behavior.

# 4 PARITY, AND, AND OR CAN SATISFY THE LEARNABILITY CONDITIONS

Thm. 3.1 and Thm. 3.2 establish the conditions for the learnability of general $k$-sparse Boolean functions, i.e., the separation of the critical gradient component $\gamma_{l^{(t)}}^p$ Eq. (9), which depend on the formulations of the activation $\psi(\cdot)$ and the 2-sparse Boolean function $\phi_2(\cdot, \cdot)$ that vary for different $k$-sparse Boolean functions. In Tab. 1, we summarize $\psi(\cdot)$ and $\phi(\cdot, \cdot)$ for $k$-PARITY, $k$-AND and $k$-OR. In

| Function | $\psi(z)$ | $\phi_2(z_1, z_2)$ |
|---|---|---|
| $k$-PARITY | $z^2$ | $z_1 z_2$ |
| $k$-AND | $\max(z, 0)$ | $\frac{z_1 z_2 + z_1 + z_2 - 1}{2}$ |
| $k$-OR | $\min(z, 0) + 1$ | $\frac{-z_1 z_2 + z_1 + z_2 + 1}{2}$ |

Table 1: Formulations of $\psi(\cdot)$ and $\phi_2(\cdot, \cdot)$ combinations for different $k$-sparse Boolean functions.

particular, the activation function $\psi : [-1, 1] \to [0, 1]$ ensures that the output of the transformer Eq. (4) can be seen as the probability of generating $y = 1$ for RL as well as a score of the token for SFT. $\psi(\cdot)$ is *designed* to guarantee that $\psi((z_1 + z_2)/2)$ is large if $\phi_2(z_1, z_2) = 1$ such that the transformer has sufficient expressibility to capture the targeted $k$-sparse Boolean functions.

In this section, we apply the conditions in Thm. 3.1 and Thm. 3.2 to verify that they are satisfied by the critical gradient component Eq. (9) induced by the $\psi(\cdot)$ and $\phi_2(\cdot, \cdot)$ combinations in Tab. 1, revealing learnability of these sparse Boolean functions for transformers via RL or SFT. Note that there are other possible $\psi(\cdot)$ besides those in Tab. 1 that can ensure the expressibility of the transformer, and their corresponding learnability can be verified by examining the separation of the resulted critical gradient component following a similar approach. We present numerical experiments in App. E to support the theoretical claims.

## 4.1 PARITY

The $k$-PARITY has $\Phi_k^{\text{parity}}(\mathbf{x}) = \prod_{i \in B} x_i$ for any subset $B \subseteq [d]$, which returns $+1$ if the number of 1 of $\mathbf{x} \in \{-1, +1\}^d$ in $B$ is even and $-1$ otherwise. This gives us the 2-PARITY function $\phi_2(z_1, z_2) = z_1 z_2$ for the recursive decomposition (Sec. 2.1). We use $\psi(z) = z^2$ for $z \in [-1, 1]$. In the following, aside from discussing the separation of the critical gradient component $\gamma_{l^{(t)}}^p$ Eq. (10), we provide a more precise characterization for the learning dynamics of fine-tuning via RL for arbitrary iterations of the policy gradient training.

**Theorem 4.1** ($k$-PARITY learning dynamics of fine-tuning via RL). *Under the setting of Thm. 3.1, for $k$-PARITY $\Phi_k^{\text{parity}}(\mathbf{x})$, let $\psi(z) = z^2$, $\phi_2(z_1, z_2) = z_1 z_2$, and $\eta > 0$ be the learning rate. If we run RL optimized by sign policy gradient for $S \in \mathbb{Z}^+$ updates, then $\forall t \in [T]$, $l^{(t)} \in [d_t]$, the separation of the critical gradient component $\gamma_{l^{(t)}}^p$ Eq. (10) is satisfied $\forall s \in [S]$. Furthermore, the attention score $\sigma_j^i = \exp(W_{i,j}) / \sum_m \exp(W_{m,j})$ has the formulation of*

$$\sigma_{N_{t-1}+l^{(t)}}^{N_{t-2}+p}(s) = \begin{cases} \frac{1}{2} \frac{1}{1 + \frac{d_{t-1}-2}{2} \exp(-2\eta s)}, & p \in \{i_1^{l^{(t)}}, i_2^{l^{(t)}}\}, \\ \frac{1}{d_{t-1} - 2 + 2\exp(2\eta s)}, & \text{otherwise} . \end{cases} \tag{15}$$

Thm. 4.1 reveals that the separation condition for the critical gradient component $\gamma_{l^{(t)}}^p$ Eq. (10) is satisfied, hence one-step update of RL can enable the transformer to learn the $k$-PARITY. In addition, Thm. 4.1 also gives the exact learning dynamics of the model parameter for arbitrary iterations, which highlights that either a large learning rate $\eta$ or sufficient iteration number $S$ can lead the self-attention to focus on the relevant positions and ignore irrelevant ones, i.e.,

$$\sigma_{N_{t-1}+l^{(t)}}^{N_{t-2}+p}(S) \to \begin{cases} \frac{1}{2}, & \text{if } p \in \{i_1^{l^{(t)}}, i_2^{l^{(t)}}\}, \\ 0, & \text{otherwise} \end{cases}$$

for sufficiently large $S$ or $\eta$. In the following, we confirm that $k$-PARITY can also be learned by transformers via SFT in a step-wise learning manner, rather than the one-step learning of the RL.

**Claim 4.1** (Transformers with CoT can learn $k$-PARITY via SFT). *Under the setting of Thm. 3.2, for $k$-PARITY $\Phi_k^{\text{parity}}(\mathbf{x})$, let $\psi(z) = z^2$ and $\phi_2(z_1, z_2) = z_1 z_2$, then the separation of critical gradient component $\gamma_{l^{(t)}}^p$ Eq. (14) is satisfied, thus the learnability of Thm. 3.2 for SFT is guaranteed.*

## 4.2 AND AND OR

We study two more $k$-sparse Boolean functions, $k$-AND with $\Phi_k^{\mathrm{and}}(\mathbf{x}) = 2\prod_{i\in B} \frac{x_i+1}{2} - 1$ and $k$-OR with $\Phi_k^{\mathrm{or}}(\mathbf{x}) = 1 - 2\prod_{i\in B} \frac{1-x_i}{2}$, and the functions $\psi(\cdot)$ and $\phi_2(\cdot,\cdot)$ are listed in Tab. 1. We show that the condition in Thm. 3.1 for RL and that in Thm. 3.2 are satisfied, confirming the learnability of $k$-AND and $k$-OR for transformers via either RL or SFT.

**Claim 4.2** (Transformers with CoT can learn $k$-AND and $k$-OR via RL or SFT)**.** *Under the setting of Thm. 3.1 for RL (Thm. 3.2 for SFT), for $k$-AND $\Phi_k^{\mathrm{and}}(\mathbf{x})$ with $\psi(z) = \max(z,0)$ and $\phi_2(z_1, z_2) = (z_1 z_2 + z_1 + z_2 - 1)/2$ and $k$-OR $\Phi_k^{\mathrm{or}}(\mathbf{x})$ with $\psi(z) = \min(z,0)+1$ and $\phi_2(z_1, z_2) = (-z_1 z_2 + z_1 + z_2 + 1)/2$, the separation of $\gamma_{l(t)}^p$ Eq. (10) for RL (Eq. (14) for SFT) is satisfied; thus the learnability of Thm. 3.1 for RL (Thm. 3.2 for SFT) is guaranteed for both $\Phi_k^{\mathrm{and}}(\cdot)$ and $\Phi_k^{\mathrm{or}}(\cdot)$.*

## 5 DISCUSSION AND LIMITATION

In this paper, we have investigated the learning dynamics of fine-tuning transformers with CoT via either RL or SFT for learning $k$-sparse Boolean functions, including $k$-PARITY, $k$-AND, and $k$-OR. We have established sufficient conditions for the provable learning of both RL and SFT—the separation of the critical gradient component. Furthermore, our results reveal that, while both RL and SFT are capable of learning these $k$-sparse Boolean functions, they exhibit distinct learning behaviors: RL learns the whole CoT chain simultaneously but SFT must solve the prior steps of CoT chain before learning the later steps, leading to a step-wise learning phenomenon. Compared to Kim & Suzuki (2025); Wang et al. (2025), step-wise learning of CoT naturally arises from transformer without relying on curriculum learning, indicating that it can be an intrinsic property of SFT. Our findings take the first step towards understanding the mechanism of fine-tuning transformers with CoT via RL and provide additional theoretical insights on that via SFT by removing the constraints of teacher-forcing as well as additional data augmentation, resulting in a tractable comparison between them.

**Limitation and future directions.** We only take the first step of comparing RL from SFT, while we still cannot answer whether or not SFT memorizes but RL generalizes, since we do not consider generalization in this work. In addition, our results are established for population gradient for both RL and SFT, rather than giving a finite-sample version. Future works can generalize our results to gradient obtained from finite-sample. There are also more variants of vanilla policy gradient method, e.g., PPO (Schulman et al., 2017), and it will be interesting to compare different policy gradient algorithms on the CoT capabilities of transformers based on our results. Finally, we only consider a one-layer transformer with designed activation functions; future works can study multi-layer transformers to provide more general results.

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

APPENDIX

The appendix is organized as follows:

## A  RELATED WORKS

**Transformers with CoT.**   Recently, the success of CoT reasoning in transformer models has attracted a lot of attention. Toward this direction, a series of existing works have investigated the improvement of transformer expressiveness by providing CoT (Merrill & Sabharwal, 2024; Chen et al., 2024a; Li et al., 2024), while some other works aimed to reveal the inherent limitations (Barceló et al., 2025; Amiri et al., 2025). One crucial aspect of analyzing the transformers with CoT is optimization dynamics (Huang et al., 2025; Kim & Suzuki, 2025), which typically focused on the single-head transformer. A recent progress (Yang et al., 2025) generalized the analysis to mulit-head transformers and showed that a one-layer transformer can learn symbolic multi-step reasoning aided by sufficient intermediate reasoning steps of CoT.

**Learning Boolean functions with transformers.**   (Sparse) Boolean functions have been showed to be fundamentally hard for transformers to learn in an end-to-end manner. This can be attributed to the "simplicity bias" that encourages transformers to prefer low-degree functions (Vasudeva et al., 2025; Hahn & Rofin, 2024). However, when providing a "scratchpad" (CoT) as intermediate supervision, the problem is decomposed into easier sub-tasks and a one-layer transformer can learn the sparse PARITY in one gradient update via teacher forcing (Kim & Suzuki, 2025). For other sparse Boolean functions AND and OR, Hu et al. (2025) showed a similar result: a single-head softmax-attention cannot solve these sparse Boolean functions without any additional supervision, while it is capable of learning them via teacher-forcing. Our work also focuses on the learnability of transformers for sparse Boolean functions. Compared to these works, we take the first step to investigate the underlying mechanism of fine-tuning via RL, which has not been covered by prior works, and we relax several constraints of SFT such as teacher-forcing, allowing us to reveal a distinction between the learning behavior of RL and that of SFT. Wang et al. (2025) studied the learnability of $k$-fold compositional functions by transformers, focusing on how data difficulty affects training. They demonstrated that gradient-based learning (in an SFT manner) with curriculum or data mixture can enable efficient learning. The curriculum learning in Wang et al. (2025) is also stepwise, similar to our stepwise learning conclusion for SFT. The difference is that our stepwise learning behavior can emerge naturally in SFT from transformers, an intrinsic property that does not rely on external curriculum learning.

**Learning dynamics of transformers.**   Learning dynamics is a fundamental aspect for deep learning. With the increasing importance of transformers, a lot of existing works started to investigate the learning dynamics of transformers across a wide range of tasks, especially the special in-context learning (Yang et al., 2024; Chen et al., 2024b; Huang et al., 2023; Zhang et al., 2023) and the corresponding neural scaling laws (Lyu et al., 2025). In this work, we also analyze the learning dynamics

of transformers, while our focus is fine-tuning via RL or SFT rather than in-context learning, which stands at the core of our characterization of the corresponding learnability.

## B PROOFS OF SECTION 3.1

This section presents proofs for Sec. 3.1:

- App. B.1 establishes the formulation of the policy gradient Eq. (8);

- App. B.2 builds the equivalence between the separation of the policy gradient and that of the critical gradient component;

- App. B.3 proves Thm. 3.1;

- App. B.4 shows the hardness of RL with only final reward.

### B.1 FORMULATION OF THE POLICY GRADIENT

**Additional notation.** For the whole CoT reasoning sequence $\mathbf{y}$, we denote its trajectory space by $\mathcal{Y}^{(:T)}$, and the trajectory space of $\mathbf{y}^{(:t)}$ by $\mathcal{Y}^{(:t)}$.

**Lemma 1** (Formulation of the policy gradient). *The gradient for the expected reward*

$$\mathcal{R}(\boldsymbol{W}) := \mathbb{E}_{\mathbf{x}}\left[R(\mathbf{x}, \boldsymbol{W})\right] := \mathbb{E}_{\mathbf{x}}\left[\mathbb{E}_{\mathbf{y} \sim p_{\boldsymbol{W}}(\cdot|\mathbf{x})}\left[\sum_{t=1}^{T} r_t\left(\mathbf{y}^{(t)}, \mathbf{y}^{(t-1)}\right)\right]\right] \tag{16}$$

*is computed by*

$$\nabla_{\boldsymbol{W}} R(\mathbf{x}; \boldsymbol{W}) = \mathbb{E}_{\mathbf{y} \sim p_{\boldsymbol{W}}(\cdot|\mathbf{x})}\left[\sum_{t=1}^{T} \nabla_{\boldsymbol{W}} \ln p_{\boldsymbol{W}}\left(\mathbf{y}^{(t)}|\mathbf{y}^{(t-1)}\right) \sum_{\tau=t}^{T} r_\tau\left(\mathbf{y}^{(\tau)}, \mathbf{y}^{(\tau-1)}\right)\right]. \tag{17}$$

*Proof.* According to the definition of $R(\mathbf{x}; \boldsymbol{W})$, we can write its gradient as

$$\nabla_{\boldsymbol{W}} R(\mathbf{x}; \boldsymbol{W})$$

$$= \sum_{\mathbf{y} \in \mathcal{Y}^{(:T)}} \nabla_{\boldsymbol{W}} p_{\boldsymbol{W}}(\mathbf{y}|\mathbf{x}) \sum_{\tau=1}^{T} r_\tau\left(\mathbf{y}^{(\tau)}, \mathbf{y}^{(\tau-1)}\right)$$

$$= \sum_{t=1}^{T} \sum_{\mathbf{y} \in \mathcal{Y}^{(:T)}} p_{\boldsymbol{W}}(\mathbf{y}|\mathbf{x}) \nabla_{\boldsymbol{W}} \ln p_{\boldsymbol{W}}\left(\mathbf{y}^{(t)}|\mathbf{y}^{(t-1)}\right)\left(r_{:t-1}(\mathbf{y}^{(:t-1)}) + r_{t:}(\mathbf{y}^{(t-1:)})\right), \tag{18}$$

where we apply

$$\nabla_{\boldsymbol{W}} p_{\boldsymbol{W}}(\mathbf{y}|\mathbf{x}) = p_{\boldsymbol{W}}(\mathbf{y}|\mathbf{x}) \nabla_{\boldsymbol{W}} \ln p_{\boldsymbol{W}}(\mathbf{y}|\mathbf{x})$$

and $p_{\boldsymbol{W}}(\mathbf{y}|\mathbf{x}) = \prod_{t=1}^{T} p_{\boldsymbol{W}}\left(\mathbf{y}^{(t)}|\mathbf{y}^{(t-1)}\right)$ with $\mathbf{y}^{(0)} = \mathbf{x}$ in the second equality, and we define

$$r_{:t}(\mathbf{y}^{(:t)}) := \sum_{\tau=1}^{t} r_\tau\left(\mathbf{y}^{(\tau)}, \mathbf{y}^{(\tau-1)}\right),$$

$$r_{t:}(\mathbf{y}^{(t-1:)}) := \sum_{\tau=t}^{T} r_\tau\left(\mathbf{y}^{(\tau)}, \mathbf{y}^{(\tau-1)}\right). \tag{19}$$

For any $t \in [T]$, we can derive

$$\sum_{\mathbf{y} \in \mathcal{Y}^{(:T)}} p_{\boldsymbol{W}}(\mathbf{y}|\mathbf{x}) \nabla_{\boldsymbol{W}} \ln p_{\boldsymbol{W}}\left(\mathbf{y}^{(t)}|\mathbf{y}^{(t-1)}\right) r_{:t-1}(\mathbf{y}^{(:t-1)})$$

$$= \sum_{\mathbf{y}^{(:t)} \in \mathcal{Y}^{(:t)}} p_{\boldsymbol{W}}\left(\mathbf{y}^{(:t)}|\mathbf{x}\right) \nabla_{\boldsymbol{W}} \ln p_{\boldsymbol{W}}\left(\mathbf{y}^{(t)}|\mathbf{y}^{(t-1)}\right) r_{:t-1}(\mathbf{y}^{(:t)}) \left[ \sum_{\mathbf{y}^{(t+1:)} \in \mathcal{Y}^{(:t+1)}} p_{\boldsymbol{W}}\left(\mathbf{y}^{(t+1:)}|\mathbf{y}^{(:t)}\right) \right]$$

$$= \sum_{\mathbf{y}^{(:t-1)} \in \mathcal{Y}^{(:t-1)}} p_{\boldsymbol{W}}\left(\mathbf{y}^{(:t-1)}|\mathbf{x}\right) r_{:t-1}(\mathbf{y}^{(:t-1)}) \sum_{\mathbf{y}^{(t)} \in \mathcal{Y}^{(t)}} p_{\boldsymbol{W}}\left(\mathbf{y}^{(t)}|\mathbf{y}^{(t-1)}\right) \nabla_{\boldsymbol{W}} \ln p_{\boldsymbol{W}}\left(\mathbf{y}^{(t)}|\mathbf{y}^{(t-1)}\right)$$

$$= \sum_{\mathbf{y}^{(:t-1)} \in \mathcal{Y}^{(:t-1)}} p_{\boldsymbol{W}}\left(\mathbf{y}^{(:t-1)}|\mathbf{x}\right) r_{:t-1}(\mathbf{y}^{(:t-1)}) \nabla_{\boldsymbol{W}} \left[ \sum_{\mathbf{y}^{(t)} \in \mathcal{Y}^{(t)}} p_{\boldsymbol{W}}\left(\mathbf{y}^{(t)}|\mathbf{y}^{(t-1)}\right) \right] = 0,$$

$$(20)$$

where we apply $\sum_{\mathbf{y}^{(t+1:)}} p_{\boldsymbol{W}}\left(\mathbf{y}^{(t+1:)}|\mathbf{y}^{(:t)}\right) = 1$ in the second equality. Thus, Eq. (18) only has the term $r_{t:}$ and the claim of the lemma is established. $\qquad \square$

## B.2 SEPARATION OF THE CRITICAL GRADIENT COMPONENT

For convenience, we recall that the critical gradient component is defined by

$$\textbf{Critical gradient component: } \gamma_{l^{(t)}}^j(\mathbf{y}^{(t-1)}) := \frac{2}{k-1} \psi'\left(\xi_{l^{(t)}}\right) \phi_2\left(y_{i_1^{l^{(t)}}}^{(t-1)}, y_{i_2^{l^{(t)}}}^{(t-1)}\right) y_j^{(t-1)}. \quad (21)$$

Below we discuss the relation between the policy gradient Eq. (8) and the critical gradient component as well as the equivalence between their separations.

**Lemma 2.** $\forall t \in [T], l^{(t)} \in [d_t]$, let $i_1^{l^{(t)}}$ and $i_2^{l^{(t)}}$ be two child nodes of the $l^{(t)}$-th node of $\mathbf{y}^{(t)}$, then the policy gradient $\mathbb{E}_{\mathbf{x}}\left[\nabla_{\boldsymbol{W}} R(\mathbf{x}; \boldsymbol{W})\right]$ Eq. (8) can be expressed by the critical gradient component as $(p \in [d_{t-1}])$

$$\partial_{W_{N_{t-2}+p, N_{t-1}+l^{(t)}}} R(\mathbf{x}; \boldsymbol{W})$$

$$= \mathbb{E}_{\mathbf{y}^{(:t-1)} \sim p_{\boldsymbol{W}}(\cdot|\mathbf{x})} \left[ \gamma_{l^{(t)}}^p(\mathbf{y}^{(t-1)}) - \sum_{i=1}^{d_{t-1}} \sigma_{N_{t-1}+l^{(t)}}^{N_{t-2}+i} \gamma_{l^{(t)}}^i(\mathbf{y}^{(t-1)}) \right] \sigma_{N_{t-1}+l^{(t)}}^{N_{t-2}+p}. \quad (22)$$

Furthermore, if $\boldsymbol{W} = c\mathbf{1}$ for an arbitrary constant $c$, the separation of the policy gradient $\forall p \in \{i_1^{l^{(t)}}, i_2^{l^{(t)}}\}$, $p' \in [d_{t-1}] \backslash \{i_1^{l^{(t)}}, i_2^{l^{(t)}}\}$:

$$\mathbb{E}_{\mathbf{x}}\left[\partial_{W_{N_{t-2}+p, N_{t-1}+l^{(t)}}} R(\mathbf{x}; \boldsymbol{W}) - \partial_{W_{N_{t-2}+p', N_{t-1}+l^{(t)}}} R(\mathbf{x}; \boldsymbol{W})\right] > 0 \quad (23)$$

is equivalent to the separation of the critical gradient component $\forall p \in \{i_1^{l^{(t)}}, i_2^{l^{(t)}}\}$, $p' \in [d_{t-1}] \backslash \{i_1^{l^{(t)}}, i_2^{l^{(t)}}\}$ :

$$\mathbb{E}_{\mathbf{x}, \mathbf{y}^{(:t-1)} \sim p_{\boldsymbol{W}}(\cdot|\mathbf{x})} \left[\gamma_{l^{(t)}}^p(\mathbf{y}^{(t-1)}) - \gamma_{l^{(t)}}^{p'}(\mathbf{y}^{(t-1)})\right] > 0. \quad (24)$$

*Proof.* To prove this claim, we start from analyzing the detailed formulation of the policy gradient with all components written explicitly. Specifically, given $t \in [T]$, only the components with $p \in [d_{t-1}]$ and $l^{(t)} \in [d_t]$,

$$W_{N_{t-2}+p, N_{t-1}+l^{(t)}} \neq -\infty \quad (25)$$

given the pretrained mask discussed in Sec. 2.2. As a result, we only need to analyze the policy gradient for these components. According to

$$\partial_{W_{N_{t-2}+p,N_{t-1}+l^{(t)}}} R(\mathbf{x};\boldsymbol{W})$$

$$\stackrel{(i)}{=} \sum_{\mathbf{y}\in\mathcal{Y}^{(:t)}} p_{\boldsymbol{W}}(\mathbf{y}^{(:t-1)}|\mathbf{x})p_{\boldsymbol{W}}(\mathbf{y}^{(t+1:)}|\mathbf{y}^{(t)})\partial_{W_{N_{t-2}+p,N_{t-1}+l^{(t)}}}p_{\boldsymbol{W}}(\mathbf{y}^{(t)}|\mathbf{y}^{(t-1)})r_t(\mathbf{y}^{(t)},\mathbf{y}^{(t-1)})$$

$$= \sum_{\mathbf{y}^{(:t-1)}\in\mathcal{Y}^{(:t-1)}} p_{\boldsymbol{W}}(\mathbf{y}^{(:t-1)}|\mathbf{x}) \sum_{\mathbf{y}^{(t:)}\in\mathcal{Y}^{(t:)}} \partial_{W_{N_{t-2}+p,N_{t-1}+l^{(t)}}}p_{\boldsymbol{W}}(\mathbf{y}^{(t)}|\mathbf{y}^{(t-1)})r_t(\mathbf{y}^{(t)},\mathbf{y}^{(t-1)})p_{\boldsymbol{W}}(\mathbf{y}^{(t+1:)}|\mathbf{y}^{(t)})$$

$$= \sum_{\mathbf{y}^{(:t-1)}\in\mathcal{Y}^{(:t-1)}} p_{\boldsymbol{W}}(\mathbf{y}^{(:t-1)}|\mathbf{x})$$

$$\times \left[ \sum_{\mathbf{y}^{(t)}\in\mathcal{Y}^{(t)}} \partial_{W_{N_{t-2}+p,N_{t-1}+l^{(t)}}}p_{\boldsymbol{W}}(\mathbf{y}^{(t)}|\mathbf{y}^{(t-1)})r_t(\mathbf{y}^{(t)},\mathbf{y}^{(t-1)}) \sum_{\mathbf{y}^{(t+1:)}\in\mathcal{Y}^{(t+1:)}} p_{\boldsymbol{W}}(\mathbf{y}^{(t+1:)}|\mathbf{y}^{(t)}) \right]$$

$$\stackrel{(ii)}{=} \mathbb{E}_{\mathbf{y}^{(:t-1)}\sim p_{\boldsymbol{W}}(\cdot|\mathbf{x})} \left[ \sum_{\mathbf{y}^{(t)}\in\mathcal{Y}^{(t)}} \partial_{W_{N_{t-2}+p,N_{t-1}+l^{(t)}}}p_{\boldsymbol{W}}(\mathbf{y}^{(t)}|\mathbf{y}^{(t-1)})r_t(\mathbf{y}^{(t)},\mathbf{y}^{(t-1)}) \right], \tag{26}$$

where $(i)$ is a result of the definition of $\nabla_{\boldsymbol{W}} R(\mathbf{x};\boldsymbol{W})$, the fact that each $\mathbf{y}^{(t)}$ only depends on $\mathbf{y}^{(t-1)}$ due to the pretrained mask

$$p_{\boldsymbol{W}}(\mathbf{y}|\mathbf{x}) = p_{\boldsymbol{W}}(\mathbf{y}^{(:t-1)}|\mathbf{x})p_{\boldsymbol{W}}(\mathbf{y}^{(t:)}|\mathbf{x},\mathbf{y}^{(:t-1)}) = p_{\boldsymbol{W}}(\mathbf{y}^{(:t-1)}|\mathbf{x})p_{\boldsymbol{W}}(\mathbf{y}^{(t:)}|\mathbf{y}^{(t-1)}),$$

and that $W_{N_{t-2}+i,N_{t-1}+l^{(t)}}$ only determines $p_{\boldsymbol{W}}(\mathbf{y}^{(t)}|\mathbf{y}^{(t-1)})$; $(ii)$ is because the law of total probability. Now, as the formulation of the transformer implies Eq. (4) that

$$p_{\boldsymbol{W}}(y_{l^{(t)}}^{(t)} = 1|\mathbf{y}^{(t-1)}) = [f(\mathbf{x},\mathbf{y};\boldsymbol{W})]_{N_{t-1}+l^{(t)}} = \psi\left(\xi_{l^{(t)}}\right) \tag{27}$$

with $\xi_{l^{(t)}} := \sum_{i=1}^{d_{t-1}} y_i^{(t-1)}\sigma_{N_{t-1}+l^{(t)}}^{N_{t-2}+i}$, we need the gradient of the attention score $\sigma_i^j$, which is computed as

$$\partial_{W_{i,j}}\sigma_n^m = \delta_{jn}(\delta_{im} - \sigma_j^i)\sigma_n^m. \tag{28}$$

This gives us (with some straightforward algebra)

$$\partial_{W_{N_{t-2}+p,N_{t-1}+l^{(t)}}}p_{\boldsymbol{W}}(y_{l^{(t)}}^{(t)} = 1|\mathbf{y}^{(t-1)})$$

$$= \psi'(\xi_{l^{(t)}}) \sum_{i=1}^{d_{t-1}} y_i^{(t-1)}\partial_{W_{N_{t-2}+p,N_{t-1}+l^{(t)}}}\sigma_{N_{t-1}+l^{(t)}}^{N_{t-2}+i}$$

$$= \psi'(\xi_{l^{(t)}}) \sum_{i=1}^{d_{t-1}} y_i^{(t-1)}\left(\delta_{ip} - \sigma_{N_{t-1}+l^{(t)}}^{N_{t-2}+p}\right)\sigma_{N_{t-1}+l^{(t)}}^{N_{t-2}+i} \tag{29}$$

$$= \psi'(\xi_{l^{(t)}}) \left(y_p^{(t-1)} - \sum_{i=1}^{d_{t-1}} y_i^{(t-1)}\sigma_{N_{t-1}+l^{(t)}}^{N_{t-2}+i}\right)\sigma_{N_{t-1}+l^{(t)}}^{N_{t-2}+p}.$$

On the other hand, by using the fact that $p_{\boldsymbol{W}}(y_{l^{(t)}}^{(t)} = -1|\mathbf{y}^{(t-1)}) = 1 - p_{\boldsymbol{W}}(y_{l^{(t)}}^{(t)} = 1|\mathbf{y}^{(t-1)})$, we are able to conclude that

$$\partial_{W_{N_{t-2}+p,N_{t-1}+l^{(t)}}}p_{\boldsymbol{W}}(y_{l^{(t)}}^{(t)} = -1|\mathbf{y}^{(t-1)})$$

$$= -\psi'(\xi_{l^{(t)}}) \left(y_p^{(t-1)} - \sum_{i=1}^{d_{t-1}} y_i^{(t-1)}\sigma_{N_{t-1}+l^{(t)}}^{N_{t-2}+i}\right)\sigma_{N_{t-1}+l^{(t)}}^{N_{t-2}+p}. \tag{30}$$

Thus, we can summarize the gradient as

$$\partial_{W_{N_{t-2}+p,N_{t-1}+l^{(t)}}}p_{\boldsymbol{W}}(y_{l^{(t)}}^{(t)}|\mathbf{y}^{(t-1)})$$

$$= \psi'(\xi_{l^{(t)}}) \left(y_p^{(t-1)} - \sum_{i=1}^{d_{t-1}} y_i^{(t-1)}\sigma_{N_{t-1}+l^{(t)}}^{N_{t-2}+i}\right)\sigma_{N_{t-1}+l^{(t)}}^{N_{t-2}+p}y_{l^{(t)}}^{(t)}. \tag{31}$$

**Expressing the policy gradient with the critical gradient component.** We now express the policy gradient using the critical gradient component to prove the first claim of the lemma. Applying Eq. (31), we are able to rewrite Eq. (26) as (recall that all tokens $y_j^{(t)}$ for $j \in [d_{t-1}]$ of the sequence $\mathbf{y}^{(t)}$ are independent of each other hence $p_{\mathbf{W}}(\mathbf{y}^{(t)}|\mathbf{y}^{(t-1)}) = \prod_{l=1}^{d_t} p_{\mathbf{W}}(y_l^{(t)}|\mathbf{y}^{(t-1)})$):

$$\sum_{\mathbf{y}^{(t)} \in \mathcal{Y}^{(t)}} \partial_{W_{N_{t-2}+p, N_{t-1}+l^{(t)}}} p_{\mathbf{W}}(\mathbf{y}^{(t)}|\mathbf{y}^{(t-1)}) r_t(\mathbf{y}^{(t)}, \mathbf{y}^{(t-1)})$$

$$= [\psi'(\xi_{l^{(t)}})] \left( y_p^{(t-1)} - \sum_{i=1}^{d_{t-1}} y_i^{(t-1)} \sigma_{N_{t-1}+l^{(t)}}^{N_{t-2}+i} \right) \sigma_{N_{t-1}+l^{(t)}}^{N_{t-2}+p} \sum_{\mathbf{y}^{(t)} \in \mathcal{Y}^{(t)}} \frac{p_{\mathbf{W}}(\mathbf{y}^{(t)}|\mathbf{y}^{(t-1)})}{p_{\mathbf{W}}(y_{l^{(t)}}^{(t)}|\mathbf{y}^{(t-1)})} r_t(\mathbf{y}^{(t)}, \mathbf{y}^{(t-1)}) y_{l^{(t)}}^{(t)}$$

$$\stackrel{(i)}{=} \frac{2}{k-1} [\psi'(\xi_{l^{(t)}})] \left( y_p^{(t-1)} - \sum_{i=1}^{d_{t-1}} y_i^{(t-1)} \sigma_{N_{t-1}+l^{(t)}}^{N_{t-2}+i} \right) \sigma_{N_{t-1}+l^{(t)}}^{N_{t-2}+p} \bar{y}_{l^{(t)}}^{(t)}$$

$$\stackrel{(ii)}{=} \frac{2}{k-1} \left[ \psi'(\xi_{l^{(t)}}) \phi\left( y_{i_1^{l^{(t)}}}^{(t-1)}, y_{i_2^{l^{(t)}}}^{(t-1)} \right) \right] \left( y_p^{(t-1)} - \sum_{i=1}^{d_{t-1}} y_i^{(t-1)} \sigma_{N_{t-1}+l^{(t)}}^{N_{t-2}+i} \right) \sigma_{N_{t-1}+l^{(t)}}^{N_{t-2}+p}$$

$$\stackrel{(iii)}{=} \left( \gamma_{l^{(t)}}^p(\mathbf{y}^{(t-1)}) - \sum_{i=1}^{d_{t-1}} \gamma_{l^{(t)}}^i(\mathbf{y}^{(t-1)}) \sigma_{N_{t-1}+l^{(t)}}^{N_{t-2}+i} \right) \sigma_{N_{t-1}+l^{(t)}}^{N_{t-2}+p}, \tag{32}$$

where, denoting $\mathbf{y}_{l/}^{(t)} = (y_1^{(t)}, \ldots, y_{l-1}^{(t)}, y_{l+1}^{(t)}, \ldots, y_{d_t}^{(t)})$ as $\mathbf{y}^{(t)}$ without the $l$-th token, the equality $(i)$ is because

$$\sum_{\mathbf{y}^{(t)} \in \mathcal{Y}^{(t)}} \frac{p_{\mathbf{W}}(\mathbf{y}^{(t)}|\mathbf{y}^{(t-1)})}{p_{\mathbf{W}}(y_{l^{(t)}}^{(t)}|\mathbf{y}^{(t-1)})} r_t(\mathbf{y}^{(t)}, \mathbf{y}^{(t-1)}) y_{l^{(t)}}^{(t)}$$

$$= \frac{1}{k-1} \sum_{\mathbf{y}^{(t)} \in \mathcal{Y}^{(t)}} p_{\mathbf{W}}(\mathbf{y}_{l/^{(t)}}^{(t)}|\mathbf{y}^{(t-1)}) \left( \bar{y}_{l^{(t)}}^{(t)} + \sum_{l \neq l^{(t)}} y_l^{(t)} \bar{y}_l^{(t)} y_{l^{(t)}}^{(t)} \right)$$

$$= \frac{2}{k-1} \bar{y}_{l^{(t)}}^{(t)} + \frac{1}{k-1} \left( \sum_{y_{l^{(t)}}^{(t)} \in \{-1,+1\}} y_{l^{(t)}}^{(t)} \right) \sum_{\mathbf{y}_{l/^{(t)}}^{(t)}} p_{\mathbf{W}}(\mathbf{y}_{l/^{(t)}}^{(t)}|\mathbf{y}^{(t-1)}) \sum_{l \neq l^{(t)}} y_l^{(t)} \bar{y}_l^{(t)}$$

$$= \frac{2}{k-1} \bar{y}_{l^{(t)}}^{(t)}; \tag{33}$$

the equality $(ii)$ applies the definition of $\bar{y}$; the equality $(iii)$ applies the definition of the critical gradient component

$$\gamma_{l^{(t)}}^p(\mathbf{y}^{(t-1)}) := \frac{2}{k-1} \psi'(\xi_{l^{(t)}}) \phi_2\left( y_{i_1^{l^{(t)}}}^{(t-1)}, y_{i_2^{l^{(t)}}}^{(t-1)} \right) y_p^{(t-1)}. \tag{34}$$

Now, inserting Eq. (32) back to Eq. (26) proves the first claim of the lemma Eq. (22).

**Establishing the equivalence between the separations.** Now we establish the equivalence between the separation of the policy gradient and that of the critical gradient component. Under the setting of the lemma we can assume $\mathbf{W} = \mathbf{1}$ w.l.g such that $\sigma_{N_{t-1}+i}^{N_{t-2}+j} = 1/d_{t-2}$, then Eq. (32) implies that

$$\partial_{W_{N_{t-2}+p, N_{t-1}+l^{(t)}}} R(\mathbf{x}; \mathbf{W}) - \partial_{W_{N_{t-2}+p', N_{t-1}+l^{(t)}}} R(\mathbf{x}; \mathbf{W})$$

$$= \frac{1}{d_{t-2}} \mathbb{E}_{\mathbf{y}^{(:t-1)} \sim p_{\mathbf{W}}(\cdot|\mathbf{x})} \left[ \gamma_{l^{(t)}}^p(\mathbf{y}^{(t-1)}) - \gamma_{l^{(t)}}^{p'}(\mathbf{y}^{(t-1)}) \right]. \tag{35}$$

Hence, the separation of the critical gradient component is equivalent to the separation of the policy gradient as stated in the lemma. □

### B.3 PROOF OF THEOREM 3.1

For convenience, we restate Thm. 3.1 below.

**Theorem B.1** (Restated Thm. 3.1). *Given integers $d \geq k \geq 2$, consider a $k$-sparse Boolean function $\Phi_k(\cdot)$ with any subset $B \subseteq [d]$ as in Def. 2.1. Let $\boldsymbol{W}(0) = \mathbf{1}$ be the initialization and let $\boldsymbol{W}^\star = \arg\max_{\boldsymbol{W}} \mathcal{R}(\boldsymbol{W})$ be the optimal parameter that solves $\max_{\boldsymbol{W}} \mathcal{R}(\boldsymbol{W})$. Set learning rate $\eta = \Omega\left(\ln(d/\epsilon)\right)$ for any $\epsilon > 0$. If the separation of the critical gradient component $\gamma_{l^{(t)}}^p$ is satisfied for $\forall t \in [T], l^{(t)} \in [d_t]$ and $\forall p \in \{i_1^{l^{(t)}}, i_2^{l^{(t)}}\}, p' \in [d_{t-1}] \backslash \{i_1^{l^{(t)}}, i_2^{l^{(t)}}\}$ :*

$$\mathbb{E}_{\mathbf{x}, \mathbf{y}^{(:t-1)} \sim p_{\boldsymbol{W}}(\cdot|\mathbf{x})} \left[ \gamma_{l^{(t)}}^p(\mathbf{y}^{(t-1)}) - \gamma_{l^{(t)}}^{p'}(\mathbf{y}^{(t-1)}) \right] > 0 \tag{36}$$

*($p$ is a child node of $y_{l^{(t)}}^{(t)}$ while $p'$ is not), then fine-tuning the transformer $f(\cdot; \boldsymbol{W})$ via RL optimized by the sign of the policy gradient Eq. (8) after one update*

$$\boldsymbol{W}(1) = \boldsymbol{W}(0) + \eta \operatorname{sign}\left(\nabla_{\boldsymbol{W}} \mathcal{R}(\mathbf{x}; \boldsymbol{W})\right)$$

*achieves*

$$\|\operatorname{softmax}(\boldsymbol{W}(1)) - \operatorname{softmax}(\boldsymbol{W}^\star)\|_1 \leq \epsilon.$$

*Proof.* We first present a sketch.

**Proof sketch.** The proof follows a two step procedure: (i) we show that the separation of the critical gradient component implies that the policy gradient has positive values at relevant positions and negative values for all irrelevant positions; (ii) then we show that one update of the policy gradient is sufficient for the transformer to have low error. We now present the proof.

**Step (i): separation of $\gamma_{l^{(t)}}^p$ implies positive gradient at relevant positions.** First of all, Eq. (32) allows us to derive a simple relation

$$\sum_{p=1}^{d_{t-1}} \partial_{W_{N_{t-2}+p, N_{t-1}+l^{(t)}}} R(\mathbf{x}; \boldsymbol{W}) = 0 \tag{37}$$

by conducting some simple algebra. Furthermore, at initialization $\boldsymbol{W} = c\mathbf{1}$ and w.l.g we assume $c = 1$. If the separation of the critical gradient $\forall t \in [T], l^{(t)} \in [d_t]$ and $\forall p \in \{i_1^{l^{(t)}}, i_2^{l^{(t)}}\}, p' \in [d_{t-1}] \backslash \{i_1^{l^{(t)}}, i_2^{l^{(t)}}\}$ :

$$\mathbb{E}_{\mathbf{x}, \mathbf{y}^{(:t-1)} \sim p_{\boldsymbol{W}}(\cdot|\mathbf{x})} \left[ \gamma_{l^{(t)}}^p(\mathbf{y}^{(t-1)}) - \gamma_{l^{(t)}}^{p'}(\mathbf{y}^{(t-1)}) \right] > 0 \tag{38}$$

is satisfied, then Lem. 2 suggests that

$$\mathbb{E}_{\mathbf{x}} \left[ \partial_{W_{N_{t-2}+p, N_{t-1}+l^{(t)}}} R(\mathbf{x}; \boldsymbol{W}) - \partial_{W_{N_{t-2}+p', N_{t-1}+l^{(t)}}} R(\mathbf{x}; \boldsymbol{W}) \right] > 0. \tag{39}$$

According to the symmetry of $i_1^{l^{(t)}}$ and $i_2^{l^{(t)}}$ and that of all other positions (i.e., $\gamma_{l^{(t)}}^{i_1^{l^{(t)}}} = \gamma_{l^{(t)}}^{i_2^{l^{(t)}}}$ and $\forall p \in [d_{t-1}] \backslash \{i_1^{l^{(t)}}, i_2^{l^{(t)}}\} : \gamma_{l^{(t)}}^p = c_2$ for some constant $c_2$), we are able to write

$$\mathbb{E}_{\mathbf{x}} \left[ \partial_{W_{N_{t-2}+p, N_{t-1}+l^{(t)}}} R(\mathbf{x}; \boldsymbol{W}) \right] = \begin{cases} G_1, & p \in \{i_1^{l^{(t)}}, i_2^{l^{(t)}}\}, \\ G_2, & p \in [d_{t-1}] \backslash \{i_1^{l^{(t)}}, i_2^{l^{(t)}}\}, \end{cases} \tag{40}$$

using Eq. (32) where $G_1 > G_2$ according to Eq. (39). Now use Eq. (37), we obtain

$$2G_1 + (d_{t-1} - 2)G_2 = 0 \implies G_1 = -\frac{d_{t-1} - 2}{2} G_2. \tag{41}$$

Thus, we must have $G_1 > 0$ and $G_2 < 0$ since $d_{t-1} - 2 > 0$ while $G_1 > G_2$, i.e., the policy gradient has positive values at the relevant positions $p \in \{i_1^{l^{(t)}}, i_2^{l^{(t)}}\}$ (the child nodes of $l^{(t)}$-th token of $\mathbf{y}^{(t)}$) and negative values at irrelevant positions (all other nodes).

**Step (ii): one policy gradient update is sufficient.** We now directly use the sign of the policy gradient to update the model with learning rate $\eta > 0$:

$$W_{N_{t-2}+p,N_{t-1}+l^{(t)}}(1) = W_{N_{t-2}+p,N_{t-1}+l^{(t)}}(0) + \eta \operatorname{sign}\left(\mathbb{E}_{\mathbf{x}}\left[\partial_{W_{N_{t-2}+p,N_{t-1}+l^{(t)}}} R(\mathbf{x};\boldsymbol{W})\right]\right).$$

As the gradient has positive values at relevant positions and negative values at irrelevant positions, we conclude that after one-update of the policy gradient

$$W_{N_{t-2}+p,N_{t-1}+l^{(t)}}(1) = \begin{cases} 1+\eta, & p \in \{i_1^{l^{(t)}}, i_2^{l^{(t)}}\}, \\ 1-\eta, & p \in [d_{t-1}]\setminus\{i_1^{l^{(t)}}, i_2^{l^{(t)}}\}. \end{cases} \tag{42}$$

After the softmax we obtain

$$\forall t \in [T], l^{(t)} \in [d_t]: \ \sigma_{N_{t-1}+l^{(t)}}^{N_{t-2}+p}(1) = \begin{cases} \frac{1}{2}\frac{1}{1+\frac{d_{t-1}-2}{2}e^{-2\eta}}, & p \in \{i_1^{l^{(t)}}, i_2^{l^{(t)}}\}, \\ \frac{1}{d_{t-1}-2+2e^{2\eta}}, & p \in [d_{t-1}]\setminus\{i_1^{l^{(t)}}, i_2^{l^{(t)}}\}. \end{cases} \tag{43}$$

On the other hand, the model parameter $\boldsymbol{W}^\star$ that solves $\max_{\boldsymbol{W}} \mathcal{R}(\boldsymbol{W})$ has the formulation of

$$\forall t \in [T], l^{(t)} \in [d_t]: \ (\sigma^\star)_{N_{t-1}+l^{(t)}}^{N_{t-2}+p} = \begin{cases} \frac{1}{2}, & p \in \{i_1^{l^{(t)}}, i_2^{l^{(t)}}\}, \\ 0, & p \in [d_{t-1}]\setminus\{i_1^{l^{(t)}}, i_2^{l^{(t)}}\}, \end{cases} \tag{44}$$

such that the model $f(\cdot;\boldsymbol{W}^\star)$ only attends to the relevant positions when generating $y_{l^{(t)}}^{(t)}$ as long as the expressivity of the transformer $f(\cdot;\boldsymbol{W}^\star)$ is guaranteed (i.e., it can solve the $k$-sparse Boolean functions $\Phi_k(\cdot)$ perfectly). Therefore, we can calculate

$$\forall t \in [T], l^{(t)} \in [d_t]: \ \sum_{p=1}^{d_{t-1}} \left|\sigma_{N_{t-1}+l^{(t)}}^{N_{t-2}+p}(1) - (\sigma^\star)_{N_{t-1}+l^{(t)}}^{N_{t-2}+p}\right| = \frac{1}{\frac{1}{2}+\frac{e^{2\eta}}{d_{t-1}-2}}, \tag{45}$$

which gives us

$$\|\operatorname{softmax}(\boldsymbol{W}(1)) - \operatorname{softmax}(\boldsymbol{W}^\star)\|_1 = \max_t \frac{1}{\frac{1}{2}+\frac{e^{2\eta}}{d_{t-1}-2}} = \frac{1}{\frac{1}{2}+\frac{e^{2\eta}}{d-2}}. \tag{46}$$

To make $\frac{1}{\frac{1}{2}+\frac{e^{2\eta}}{d-2}} \leq \epsilon$ for some given $\epsilon > 0$, we need to ensure

$$\eta \geq \ln\left(\frac{(d-2)(2-\epsilon)}{4\epsilon}\right) \implies \eta = \Omega\left(\ln\frac{d}{\epsilon}\right). \tag{47}$$

This proves the claim. $\qquad\square$

### B.4 Hardness of RL with Final Reward

In this section, we prove Prop. 3.1 to show that the policy gradient contains negligible information of the objective so that it cannot tell relevant positions from irrelevant ones when using final reward.

*Proof.* We are interested in evaluating the variance

$$\operatorname{Var}(\mathscr{H};\boldsymbol{W}) := \mathbb{E}_{h\in\mathscr{H}}\left[\left\|\nabla_{\boldsymbol{W}}\mathcal{R}_h^{\mathrm{F}}(\boldsymbol{W}) - \mathbb{E}_{h'\in\mathscr{H}}[\nabla_{\boldsymbol{W}}\mathcal{R}_{h'}^{\mathrm{F}}(\boldsymbol{W})]\right\|^2\right]. \tag{48}$$

To derive an upper bound, it is sufficient to show that there exists some constant vector $\boldsymbol{a}$ and function $G(\mathscr{H})$ such that

$$\mathbb{E}_{h\in\mathscr{H}}\left[\left\|\nabla_{\boldsymbol{W}}\mathcal{R}_h^{\mathrm{F}}(\boldsymbol{W}) - \boldsymbol{a}\right\|^2\right] \leq G(\mathscr{H}). \tag{49}$$

Below we construct such a vector $\boldsymbol{a}$. For this purpose, we first evaluate the formulation of the policy gradient (we use $\nabla$ to denote $\nabla_{\boldsymbol{W}}$)

$$\nabla\mathcal{R}_h^{\mathrm{F}}(\boldsymbol{W})$$
$$= \mathbb{E}_{\mathbf{x}}\left[\sum_{\mathbf{y}\in\mathcal{Y}^{(:T)}} \nabla p_{\boldsymbol{W}}(\mathbf{y}|\mathbf{x})\mathbf{y}^{(T)}h(\mathbf{x})\right] \tag{50}$$
$$= \mathbb{E}_{\mathbf{x}}\left[\nabla_{\boldsymbol{W}}\mathbb{E}_{\mathbf{y}\sim p_{\boldsymbol{W}}(\cdot|\mathbf{x})}\left[\mathbf{y}^{(T)}\right]h(\mathbf{x})\right] := \mathbb{E}_{\mathbf{x}}\left[\boldsymbol{v}(\mathbf{x})h(\mathbf{x})\right],$$

where we define

$$\boldsymbol{v}(\mathbf{x}) := \nabla_{\boldsymbol{W}} \mathbb{E}_{\mathbf{y} \sim p_{\boldsymbol{W}}(\cdot | \mathbf{x})} \left[ \mathbf{y}^{(T)} \right].$$

Then, according to the assumption of bounded gradient in Prop. 3.1, we have $\|\boldsymbol{v}(\mathbf{x})\|^2 \leq M$. Now let $\boldsymbol{a} = \boldsymbol{0}$, then we obtain

$$\mathbb{E}_{h \in \mathscr{H}} \left[ \left\| \nabla_{\boldsymbol{W}} \mathcal{R}_h^{\mathrm{F}}(\boldsymbol{W}) - \boldsymbol{a} \right\|^2 \right] = \mathbb{E}_{h \in \mathscr{H}} \left[ \left\| \mathbb{E}_{\mathbf{x}} \left[ h(\mathbf{x}) \boldsymbol{v}(\mathbf{x}) \right] \right\|^2 \right]. \tag{51}$$

As long as we can bound the R.H.S of Eq. (51), we can prove our claim. To this end, we let $\langle h, v \rangle_{L_2} = \mathbb{E}_{\mathbf{x}}[h(\mathbf{x})v(\mathbf{x})]$ denote inner product in the $L_2$ space of square-integrable functions w.r.t the relevant distribution. Then the bound can be established as follows:

$$\begin{aligned}
\mathbb{E}_{h \in \mathscr{H}} \left[ \left\| \mathbb{E}_{\mathbf{x}} \left[ h(\mathbf{x}) \boldsymbol{v}(\mathbf{x}) \right] \right\|^2 \right] &= \mathbb{E}_{h \in \mathscr{H}} \left[ \sum_{\mu} \left( \mathbb{E}_{\mathbf{x}} \left[ h(\mathbf{x}) v_{\mu}(\mathbf{x}) \right] \right)^2 \right] \\
&= \sum_{\mu} \sum_{i} \frac{1}{|\mathscr{H}|} \left( \mathbb{E}_{\mathbf{x}} \left[ h_i(\mathbf{x}) v_{\mu}(\mathbf{x}) \right] \right)^2 \\
&\overset{(i)}{=} \sum_{\mu} \sum_{i} \frac{1}{|\mathscr{H}|} \langle h_i, v_{\mu} \rangle_{L_2}^2 \\
&\overset{(ii)}{\leq} \sum_{\mu} \frac{1}{|\mathscr{H}|} \sum_{i} \langle h_i, h_i \rangle_{L_2}^2 \langle v_{\mu}, v_{\mu} \rangle_{L_2}^2 \\
&\overset{(iii)}{\leq} \sum_{\mu} \frac{1}{|\mathscr{H}|} \langle v_{\mu}, v_{\mu} \rangle_{L_2}^2 \\
&= \frac{1}{|\mathscr{H}|} \mathbb{E}_{\mathbf{x}} \left[ \|\boldsymbol{v}(\mathbf{x})\|^2 \right] \leq \frac{2M}{|\mathscr{H}|},
\end{aligned} \tag{52}$$

where $(i)$ uses the definition of $\langle h, v \rangle$; $(ii)$ follows from $\mathbb{E}_{\mathbf{x}}[h(\mathbf{x})h'(\mathbf{x})] = 0$ for distinct $h, h' \in \mathscr{H}$; $(iii)$ is because $\sum_i \langle h_i, h_i \rangle_{L_2}^2 \leq 1$. $\qquad \square$

## C  PROOFS OF SECTION 3.2

We first present several helpful lemmas before proving Thm. 3.2, which is deferred to App. C.1.

**Lemma 3** (Equivalence between the separation of gradient and that of critical gradient component).
*For the population loss Eq. (13), given $t \in [T]$ and $l^{(t)} \in [d_t]$, if $\forall p \in [d_{t-1}]$ the parameter has the form $W_{N_{t-2}+p, N_{t-1}+l^{(t)}} = c$ for some constant $c$, then*

$$\begin{aligned}
& - \partial_{W_{N_{t-2}+p, N_{t-1}+l^{(t)}}} L_t(\mathbf{x}; \boldsymbol{W}) - \left( -\partial_{W_{N_{t-2}+p', N_{t-1}+l^{(t)}}} L_t(\mathbf{x}; \boldsymbol{W}) \right) \\
& = \gamma_{l^{(t)}}^p (\hat{\mathbf{y}}^{(t-1)}) - \gamma_{l^{(t)}}^{p'} (\hat{\mathbf{y}}^{(t-1)}).
\end{aligned} \tag{53}$$

*Proof.* To establish such equivalence, we need to derive the formulation of the gradient (recall that we use the hinge loss)

$$\begin{aligned}
\nabla_{\boldsymbol{W}} \mathcal{L}(\boldsymbol{W}) &:= \mathbb{E}_{\mathbf{x}} \left[ \nabla_{\boldsymbol{W}} \sum_{t=1}^{T} L_t(\mathbf{x}; \boldsymbol{W}) \right] \\
&= \frac{1}{k-1} \sum_{t=1}^{T} \sum_{l^{(t)}=1}^{d_t} \mathbb{E}_{\mathbf{x}} \left[ \nabla_{\boldsymbol{W}} \ell \left( \hat{q}_{l^{(t)}}^{(t)}, \tilde{y}_{l^{(t)}}^{(t)} \right) \right] \\
&= -\frac{1}{k-1} \sum_{t=1}^{T} \sum_{l^{(t)}=1}^{d_t} \mathbb{E}_{\mathbf{x}} \left[ \tilde{y}_{l^{(t)}}^{(t)} \nabla_{\boldsymbol{W}} \hat{q}_{l^{(t)}}^{(t)} \right],
\end{aligned} \tag{54}$$

where we use the fact that the activation function is assumed as $\psi : [-1, 1] \rightarrow [0, 1]$ to remove max of the hinge loss in the last equality. In addition, we note that $q_{l^{(t)}}^{(t)}$ only depends on $W_{i,j}$ with $j = N_{t-1} + l^{(t)}$ such that

$$\partial_{W_{i,j}} q_{l^{(t)}}^{(t)} = 0 \text{ if } j \neq N_{t-1} + l^{(t)}, \tag{55}$$

hence we will not consider these components. We now find the gradient of the score. Specifically, given $t \in [T], l^{(t)} \in [d_t]$, recall that only the components with $p \in [d_{t-1}]$,

$$W_{N_{t-2}+p, N_{t-1}+l^{(t)}} \neq -\infty \tag{56}$$

according to the pretrained mask discussed in Sec. 2.2, then (similar to Eq. (29))

$$\frac{\tilde{y}_{l^{(t)}}^{(t)}}{k-1} \partial_{W_{N_{t-2}+p, N_{t-1}+l^{(t)}}} \hat{q}_{l^{(t)}}^{(t)}$$

$$= \frac{2\tilde{y}_{l^{(t)}}^{(t)}}{k-1} \psi'\left(\xi_{l^{(t)}}\right) \left(\hat{y}_p^{(t-1)} - \sum_{i=1}^{d_{t-1}} \hat{y}_i^{(t-1)} \sigma_{N_{t-1}+l^{(t)}}^{N_{t-2}+i}\right) \sigma_{N_{t-1}+l^{(t)}}^{N_{t-2}+p} \tag{57}$$

$$= \left(\gamma_{l^{(t)}}^p(\hat{\mathbf{y}}^{(t-1)}) - \sum_{i=1}^{d_{t-1}} \gamma_{l^{(t)}}^i(\hat{\mathbf{y}}^{(t-1)})\sigma_{N_{t-1}+l^{(t)}}^{N_{t-2}+i}\right) \sigma_{N_{t-1}+l^{(t)}}^{N_{t-2}+p}.$$

Therefore, we can now establish the relation between the separation of the gradient of the population loss and that of the critical gradient component. Specifically, given $t \in [T], l^{(t)} \in [d_t]$ for any $p \in [d_{t-1}]$, if $W_{N_{t-2}+p, N_{t-1}+l^{(t)}} = c$ then all corresponding attention scores have the same value; thus

$$-\partial_{W_{N_{t-2}+p, N_{t-1}+l^{(t)}}} L_t(\mathbf{x}; \boldsymbol{W}) - \left(-\partial_{W_{N_{t-2}+p', N_{t-1}+l^{(t)}}} L_t(\mathbf{x}; \boldsymbol{W})\right)$$

$$= \gamma_{l^{(t)}}^p(\hat{\mathbf{y}}^{(t-1)}) - \gamma_{l^{(t)}}^{p'}(\hat{\mathbf{y}}^{(t-1)}). \tag{58}$$

$\square$

**Lemma 4** (Separation of critical gradient component $\gamma_{l^{(t)}}^p$ ensures positive gradient update at relevant positions and negative gradient update at irrelevant ones). *Given $t \in [T], \forall l^{(t)} \in [d_t]$ the gradient update $-\partial_{W_{N_{t-2}+p, N_{t-1}+l^{(t)}}} L_t(\mathbf{x}; \boldsymbol{W})$ has positive values at the relevant positions $p \in \{i_1^{l^{(t)}}, i_2^{l^{(t)}}\}$ (the child nodes of $l^{(t)}$-th token of $\mathbf{y}^{(t)}$) and negative values at irrelevant positions (all other nodes) if the following conditions hold:*

*(i) the model parameter $W_{N_{t-2}+p, N_{t-1}+l^{(t)}} = c$ for any $p \in [d_{t-1}]$ and $l^{(t)} \in [d_t]$;*

*(ii) $\hat{\mathbf{y}}^{(t-1)}$ is correctly generated as $\tilde{\mathbf{y}}^{(t-1)}$, i.e., $\hat{\mathbf{y}}^{(t-1)} = \tilde{\mathbf{y}}^{(t-1)}$;*

*(iii) the separation of the critical gradient component $\gamma_{l^{(t)}}^p$ in Lem. 3 is satisfied.*

*Proof.* We consider optimizing $L_t$ under the case where $\hat{\mathbf{y}}^{(t-1)} = \tilde{\mathbf{y}}^{(t-1)}$ (i.e., $\hat{\mathbf{y}}^{(t-1)}$ is correctly generated and is the same as the ground-truth label $\tilde{\mathbf{y}}^{(t-1)}$) and the parameter $W_{N_{t-2}+p, N_{t-1}+l^{(t)}} = c$ for any $p \in [d_{t-1}]$ and $l^{(t)} \in [d_t]$. As a result, if the separation of the critical gradient component is satisfied at the $t$-th step

$$\forall l^{(t)} \in [d_t], p \in \{i_1^{l^{(t)}}, i_2^{l^{(t)}}\}, p' \in [d_{t-1}] \backslash \{i_1^{l^{(t)}}, i_2^{l^{(t)}}\}:$$
$$\mathbb{E}_{\mathbf{x}}\left[\gamma_{l^{(t)}}^p(\tilde{\mathbf{y}}^{(t-1)}) - \gamma_{l^{(t)}}^{p'}(\tilde{\mathbf{y}}^{(t-1)})\right] > 0, \tag{59}$$

we can conclude that

$$\mathbb{E}_{\mathbf{x}}\left[-\partial_{W_{N_{t-2}+p, N_{t-1}+l^{(t)}}} L_t(\mathbf{x}; \boldsymbol{W}) - \left(-\partial_{W_{N_{t-2}+p', N_{t-1}+l^{(t)}}} L_t(\mathbf{x}; \boldsymbol{W})\right)\right] > 0 \tag{60}$$

by applying Lem. 3 and $\hat{\mathbf{y}}^{(t-1)} = \tilde{\mathbf{y}}^{(t-1)}$. Now, similar to the proof of RL (App. B.3), we observe that

$$\sum_{p=1}^{d_{t-1}} \partial_{W_{N_{t-2}+p, N_{t-1}+l^{(t)}}} L_t(\mathbf{x}; \boldsymbol{W}) = 0, \tag{61}$$

which further implies that

$$\mathbb{E}_{\mathbf{x}}\left[-\partial_{W_{N_{t-2}+p, N_{t-1}+l^{(t)}}} L_t(\mathbf{x}; \boldsymbol{W})\right] = \begin{cases} G_1, & p \in \{i_1^{l^{(t)}}, i_2^{l^{(t)}}\}, \\ G_2 & p \in [d_{t-1}] \backslash \{i_1^{l^{(t)}}, i_2^{l^{(t)}}\}, \end{cases} \tag{62}$$

with $G_1 > G_2$. Here we use the symmetry of $i_1^{l^{(t)}}$ and $i_2^{l^{(t)}}$ and that of all other positions (i.e., $\gamma_{l^{(t)}}^{i_1^{l^{(t)}}} = \gamma_{l^{(t)}}^{i_2^{l^{(t)}}}$ and $\forall p \in [d_{t-1}]\backslash\{i_1^{l^{(t)}}, i_2^{l^{(t)}}\} : \gamma_{l^{(t)}}^p = c_2$ for some constant $c_2$). Now use Eq. (61), then we must have $G_1 > 0$ and $G_2 < 0$. Therefore, the lemma is proved. $\qquad\square$

## C.1 PROOF OF THM. 3.2

We now prove Thm. 3.2. For convenience, we restate Thm. 3.2 below.

**Theorem C.1** (Restated Thm. 3.2). *Given integers $d \geq k \geq 2$, consider a k-sparse Boolean function $\Phi_k(\cdot)$ with any subset $B \in [d]$ as in Def. 2.1. Let $\boldsymbol{W}(0) = \boldsymbol{1}$ be the initialization and let*

$$\boldsymbol{W}^\star = \arg\min_{\boldsymbol{W}} \mathcal{L}(\boldsymbol{W})$$

*be the optimal parameter that solves $\min_{\boldsymbol{W}} \mathcal{L}(\boldsymbol{W})$. Set learning rate $\eta = \Omega\left(\ln(d/\epsilon)\right)$ for any $\epsilon > 0$. Let the transformer $f(\cdot; \boldsymbol{W})$ be fine-tuned via SFT by running sign gradient descent*

$$\boldsymbol{W}(s + 1) = \boldsymbol{W}(s) - \eta\,\mathrm{sign}\left(\nabla_{\boldsymbol{W}}\mathcal{L}(\boldsymbol{W}(s))\right). \tag{63}$$

*If the separation of the critical gradient component is satisfied for any $l^{(t)} \in [d_t]$ and any $t \in [T]$ in the sense that*

$$\forall p \in \{i_1^{l^{(t)}}, i_2^{l^{(t)}}\}, p' \in [d_{t-1}]\backslash\{i_1^{l^{(t)}}, i_2^{l^{(t)}}\} : \ \mathbb{E}_{\mathbf{x}}\left[\gamma_{l^{(t)}}^p(\tilde{\mathbf{y}}^{(t-1)}) - \gamma_{l^{(t)}}^{p'}(\tilde{\mathbf{y}}^{(t-1)})\right] > 0, \tag{64}$$

*where $\tilde{\mathbf{y}}^{(t-1)}$ is the ground-truth label of $\mathbf{y}^{(t-1)}$ given an input $\mathbf{x}$, then running sign gradient descent for $T$ iterations achieves*

$$\|\mathrm{softmax}(\boldsymbol{W}(T)) - \mathrm{softmax}(\boldsymbol{W}^\star)\|_1 \leq \epsilon. \tag{65}$$

*Proof.* Note that, different from RL, once the input sequence $\mathbf{x}$ is given, the ground-truth label $\tilde{\mathbf{y}}$ and the generated output sequence $\hat{\mathbf{y}}$ before each update of sign gradient descent are both determined. There will be two main steps of the proof, and we start with the first one.

**Step (i): Conditions of Lem. 4 at the step $t$ guarantee the learnability of $\mathbf{y}^{(t)}$.** Given $t \in [T]$, with Lem. 4, we are able to prove the learnability of $\mathbf{y}^{(t)}$, as shown below. Suppose that the model parameters are given by condition (i) and $\hat{\mathbf{y}}^{(t-1)} = \tilde{\mathbf{y}}^{(t-1)}$ as in condition (ii) of Lem. 4. If the condition (iii) is also satisfied, i.e., the separation of $\gamma_{l^{(t)}}^p$ is satisfied at the step $t$, then we can conclude that the gradient update $-\partial_{W_{N_{t-2}+p, N_{t-2}+l^{(t)}}} L_t(\mathbf{x}; \boldsymbol{W})$ has positive values at the relevant positions $p \in \{i_1^{l^{(t)}}, i_2^{l^{(t)}}\}$ and negative values at irrelevant positions for any $l^{(t)} \in [d_t]$. As a result, one update of sign gradient descent gives us

$$W_{N_{t-2}+p, N_{t-1}+l^{(t)}} = \begin{cases} 1 + \eta, & p \in \{i_1^{l^{(t)}}, i_2^{l^{(t)}}\}, \\ 1 - \eta, & p \in [d_t]\backslash\{i_1^{l^{(t)}}, i_2^{l^{(t)}}\}. \end{cases} \tag{66}$$

After the softmax we obtain

$$\forall l^{(t)} \in [d_t] : \ \sigma_{N_{t-1}+l^{(t)}}^{N_{t-2}+p} = \begin{cases} \frac{1}{2}\frac{1}{1 + \frac{d_{t-1}-2}{2}e^{-2\eta}}, & p \in \{i_1^{l^{(t)}}, i_2^{l^{(t)}}\}, \\ \frac{1}{d_{t-1}-2+2e^{2\eta}}, & p \in [d_{t-1}]\backslash\{i_1^{l^{(t)}}, i_2^{l^{(t)}}\}. \end{cases} \tag{67}$$

Compared to the optimal model parameter $(\sigma^\star)_{N_{t-1}+l^{(t)}}^{N_{t-2}+p}$ for $p \in [d_{t-1}]$ that solves $L_t$

$$\forall l^{(t)} \in [d_t] : \ (\sigma^\star)_{N_{t-1}+l^{(t)}}^{N_{t-2}+p} = \begin{cases} \frac{1}{2}, & p \in \{i_1^{l^{(t)}}, i_2^{l^{(t)}}\}, \\ 0, & p \in [d_{t-1}]\backslash\{i_1^{l^{(t)}}, i_2^{l^{(t)}}\}, \end{cases} \tag{68}$$

we can evaluate the error as

$$\forall l^{(t)} \in [d_t] : \ \sum_{p=1}^{d_{t-1}} \left| \sigma_{N_{t-1}+l^{(t)}}^{N_{t-2}+p} - (\sigma^\star)_{N_{t-1}+l^{(t)}}^{N_{t-2}+p} \right| = \frac{1}{\frac{1}{2} + \frac{e^{2\eta}}{d_{t-1}-2}} \leq \epsilon, \tag{69}$$

if $\eta = \Omega(\ln(d/\epsilon))$. Furthermore, noting that

$$\sum_{i=1}^{d_{t-1}} y_i^{(t-1)} \sigma_{N_{t-1}+l^{(t)}}^{N_{t-2}+i}$$

$$= \frac{y_{i_1^{l^{(t)}}}^{(t-1)} + y_{i_1^{l^{(t)}}}^{(t-1)}}{2} \left[ 1 - \frac{d_{t-1}-2}{d_{t-1}-2+2e^{2\eta}} \right] + \frac{1}{d_{t-1}-2+2e^{2\eta}} \sum_{p \in [d_{t-1}] \setminus \{i_1^{l^{(t)}}, i_2^{l^{(t)}}\}} y_p^{(t-1)} \tag{70}$$

$$= \sum_{i=1}^{d_{t-1}} y_i^{(t-1)} (\sigma^\star)_{N_{t-1}+l^{(t)}}^{N_{t-2}+i}$$

$$+ \frac{1}{d_{t-1}-2+2e^{2\eta}} \left[ -(d_{t-1}-2) \frac{y_{i_1^{l^{(t)}}}^{(t-1)} + y_{i_1^{l^{(t)}}}^{(t-1)}}{2} + \sum_{p \in [d_{t-1}] \setminus \{i_1^{l^{(t)}}, i_2^{l^{(t)}}\}} y_p^{(t-1)} \right],$$

we are able to conclude

$$\sum_{i=1}^{d_{t-1}} y_i^{(t-1)} (\sigma^\star)_{N_{t-1}+l^{(t)}}^{N_{t-2}+i} - 2\epsilon \leq \sum_{i=1}^{d_{t-1}} y_i^{(t-1)} \sigma_{N_{t-1}+l^{(t)}}^{N_{t-2}+i}$$

$$\leq \sum_{i=1}^{d_{t-1}} y_i^{(t-1)} (\sigma^\star)_{N_{t-1}+l^{(t)}}^{N_{t-2}+i} + 2\epsilon. \tag{71}$$

Therefore, if $\psi(\cdot)$ guarantees the flexibility of the expressibility of the transformer in the sense that

$$\phi_2(z_1, z_2) = \text{sign}\left( 2\psi\left( \frac{z_1 + z_2}{2} \right) - 1 \right) = \text{sign}\left( 2\psi\left( \frac{z_1 + z_2}{2} + \lambda \right) - 1 \right), \tag{72}$$

where we take $\lambda = -2\epsilon$ when $z_1 + z_2$ equals 1 or 0 and take $2\epsilon$ when $z_1 + z_2 = -1$ for sufficiently small $\epsilon$, one update of the sign gradient descent ensures that

$$\text{sign}\left( 2\psi\left( \sum_{i=1}^{d_{t-1}} y_i^{(t-1)} (\sigma^\star)_{N_{t-1}+l^{(t)}}^{N_{t-2}+i} \right) - 1 \right) = \text{sign}\left( 2\psi\left( \sum_{i=1}^{d_{t-1}} y_i^{(t-1)} \sigma_{N_{t-1}+l^{(t)}}^{N_{t-2}+i} \right) - 1 \right), \tag{73}$$

and, as a result, all tokens $\hat{y}_{l^{(t)}}^{(t)}$ for $l^{(t)} \in [d_t]$ will be generated correctly.

**Step (ii): $T$-updates of sign gradient descent can achieve small error.** We prove this claim in an induction manner. We use non-negative integers $s$ to count sign gradient descent, which is distinct from $t$ of the CoT chain. We let $\eta = \Omega(\ln(d/\epsilon))$.

1. First of all, at $s = 0$, the models parameters $W = \mathbf{1}c$, hence, condition (i) of Lem. 4 is satisfied. Furthermore, all $\hat{y}_{l^{(1)}}^{(1)}$ must be assigned to a same value, i.e.,

$$\text{either } \hat{y}_{l^{(1)}}^{(1)} = 1, \forall l^{(1)} \in [d_1] \text{ or } \hat{y}_{l^{(1)}}^{(1)} = -1, \forall l^{(1)} \in [d_1], \tag{74}$$

because $\sigma_{N_0+l^{(1)}}^{N_{-1}+p}(s=0) = 1/d_0$ for any $l^{(1)} \in [d_1]$ such that all $\xi_{l^{(1)}}$ for different $l^{(1)}$ are equal according to the definition which further implies that the outputs of the transformer $\psi(\xi_{l^{(1)}})$ for different $l^{(1)}$ are equal. As a result, Eq. (57) tells us $\partial_{W_{N_0+p,N_1+l^{(2)}}} \hat{q}_{l^{(2)}}^{(2)} = 0$ for any $l^{(2)} \in [d_2]$ and $p \in [d_1]$. Similar argument can give us $\partial_{W_{N_{t-2}+p,N_{t-1}+l^{(t)}}} \hat{q}_{l^{(t)}}^{(t)} = 0$ for any $t \in [2, T]$ and any $l^{(t)} \in [d_t]$. Hence, all parameters $W_{i,j}$ with $j \in [N_1 + 1, N_T]$ will not be updated at $s = 0$, and we only need to consider optimizing $\mathbb{E}_{\mathbf{x}}[L_1(\mathbf{x}, W)]$.

For the first step of the CoT chain $t = 1$, we have $\hat{\mathbf{y}}^{(t-1)} = \tilde{\mathbf{y}}^{(t-1)} = \mathbf{x}$, and thus condition (ii) of Lem. 4 is satisfied. If the condition (iii) of Lem. 4, the separation of the critical

gradient component, is further satisfied, then we can apply our statement in **Step (i)** to conclude that one update of sign gradient descent gives us

$$\sum_{p=1}^{d_0} \left| \sigma_{N_0+l^{(1)}}^{N_{-1}+p} - (\sigma^\star)_{N_0+l^{(1)}}^{N_{-1}+p} \right| = \frac{1}{\frac{1}{2} + \frac{e^{2\eta}}{d-2}} \le \epsilon, \tag{75}$$

and all tokens $\hat{y}_{l^{(1)}}^{(1)}$ for $l^{(1)} \in [d_1]$ will be generated correctly for sufficiently small $\epsilon$ that guarantees Eq. (72).

2. Now at $s = 1$, we consider the second step of the CoT chain $t = 2$, where $\hat{\mathbf{y}}^{(t-1)}$ is generated correctly. Therefore, following the argument for $s = 0$, we can easily conclude that only $\mathbb{E}_{\mathbf{x}}[L_2(\mathbf{x}; \mathbf{W})]$ will be optimized and, under the separation of the critical gradient component, we have that after one update of sign gradient descent

$$\forall l^{(2)} \in [d_2] : \sum_{p=1}^{d_1} \left| \sigma_{N_1+l^{(2)}}^{N_0+p} - (\sigma^\star)_{N_1+l^{(2)}}^{N_0+p} \right| = \frac{1}{\frac{1}{2} + \frac{e^{2\eta}}{d_1-2}} \le \frac{1}{\frac{1}{2} + \frac{e^{2\eta}}{d-2}} \le \epsilon, \tag{76}$$

and $\hat{\mathbf{y}}^{(2)}$ will be generated correctly.

3. Repeating the above argument, we can conclude that, if the separation of the critical gradient component is satisfied as in Thm. 3.2 and the parameter is initialized as $\mathbf{W} = \mathbf{1}c$, each update of the sign gradient descent solves and only solves one step $\mathbf{y}^{(t)}$ and $T$ updates solve the whole CoT chain, with an error

$$\| \text{softmax}(\mathbf{W}(T)) - \text{softmax}(\mathbf{W}^\star) \|_1 = \max_t \frac{1}{\frac{1}{2} + \frac{e^{2\eta}}{d_t-2}} = \frac{1}{\frac{1}{2} + \frac{e^{2\eta}}{d-2}} \le \epsilon. \tag{77}$$

$\square$

# D  PROOFS OF SECTION 4

In this section, for given $k$-sparse Boolean functions, we verify that they satisfy the conditions established in Thm. 3.1 and that in Thm. 3.2, which reveals that transformers with CoT provably learn these functions via RL or SFT.

- App. D.1 discusses the design of the activation functions listed in Tab. 1;
- App. D.2 discusses $k$-PARITY;
- App. D.3.1 discusses $k$-AND as well as $k$-OR.

## D.1  FORMULATIONS OF THE ACTIVATION FUNCTIONS

Recall that the activation function $\psi : [-1, 1] \to [0, 1]$ ensures that the output of the transformer Eq. (4) can be seen as the probability of generating $y = 1$ for RL as well as a score of the token for SFT. Therefore, we must make sure that $\psi((z_1 + z_2)/2)$ is large if $\phi_2(z_1, z_2) = 1$ such that the transformer has sufficient expressibility to capture the targeted $k$-sparse Boolean functions. Following this principle:

1. For $k$-PARITY, we have $\phi_2(z_1, z_2) = z_1 z_2$ in the sense that

$$\phi_2(z_1, z_2) = \begin{cases} 1, & z_1 = z_2, \\ -1, & z_1 \ne z_2; \end{cases} \tag{78}$$

therefore, we need $\psi(\pm 1) \approx 1$ while $\psi(0) \approx 0$. A natural choice will be $\psi(x) = x^2$.

2. For $k$-AND, we have

$$\phi_2(z_1, z_2) = \begin{cases} 1, & z_1 = z_2 = 1, \\ -1, & \text{otherwise}; \end{cases} \tag{79}$$

hence we need $\psi(1) \approx 1$ while $\psi(-1)$ and $\psi(0)$ are approximately 0. A simple function that satisfies this condition is $\psi(x) = \max(x, 0)$.

3. For $k$-OR, we have

$$\phi_2(z_1, z_2) = \begin{cases} -1, & z_1 = z_2 = -1, \\ 1, & \text{otherwise}; \end{cases} \tag{80}$$

hence we need $\psi(-1) \approx 0$ while $\psi(1)$ and $\psi(0)$ are approximately 1. A simple function that satisfies this condition will be $\psi(x) = 1 + \min(x, 0)$.

## D.2 PARITY

### D.2.1 FINE-TUNING VIA RL

For parity, we can give a more exact characterization of the learning dynamics as shown in Thm. 4.1. Recall that $\psi(z) = z^2$ and $\phi_2(z_1, z_2) = z_1 z_2$. We have

$$\psi'(\xi_{l^{(t)}}) = 2\xi_{l^{(t)}} = 2\sum_j \sigma_{N_{t-1}+l^{(t)}}^{N_{t-2}+j} y_j^{(t-1)}, \tag{81}$$

$$\phi\left(y_{i_1^{l^{(t)}}}^{(t-1)}, y_{i_2^{l^{(t)}}}^{(t-1)}\right) = y_{i_1^{l^{(t)}}}^{(t-1)} y_{i_2^{l^{(t)}}}^{(t-1)}. \tag{82}$$

We first present a helpful lemma.

**Lemma 5.** *Given $t \in [T]$, if the sum of the square of the attention score $\sigma_{N_{t-1}+l^{(t)}}^{N_{t-2}+q}$ over $q$ is the same for different $l^{(t)}$:*

$$\forall l^{(t)} \in [d_t] : \sum_{q=1}^{d_{t-1}} \left(\sigma_{N_{t-1}+l^{(t)}}^{N_{t-2}+q}\right)^2 = c_1^{(t)}, \tag{83}$$

*then we have that the expectations for different tokens $y_{l^{(t)}}^{(t)}$ in the same layer $t$ have a same value:*

$$\forall l^{(t)} \in [d_t] : \mathop{\mathbb{E}}_{\mathbf{x}, \mathbf{y}^{(:t)} \sim p_{\mathbf{W}}(\cdot|\mathbf{x})} \left[y_{l^{(t)}}^{(t)}\right] = c_2^{(t)}, \tag{84}$$

*with $|c_2^{(t)}| < 1$.*

We first discuss the proof of Thm. 4.1 before discussing the proof of the above lemma.

*Proof of Thm. 4.1.* The proof is based on the evaluation of the critical gradient component, which enables to verify that it satisfies the separation condition in Thm. 3.1. Below we calculate $\gamma_{l^{(t)}}^p$:

$$\mathop{\mathbb{E}}_{\mathbf{y}^{(:t-1)}} \left[\gamma_{l^{(t)}}^p(\mathbf{y}^{(:t-1)})\right] := \mathop{\mathbb{E}}_{\mathbf{y}^{(:t-1)}} \left[\left(\sum_{j=1}^{d_{t-1}} \sigma_{N_{t-1}+l^{(t)}}^{N_{t-2}+j} y_j^{(t-1)}\right) y_{i_1^{l^{(t)}}}^{(t-1)} y_{i_2^{l^{(t)}}}^{(t-1)} y_p^{(t-1)}\right]. \tag{85}$$

Consider optimizing the expected reward $\mathcal{R}(\mathbf{W})$ with the sign of the policy gradient. We use non-negative integers $s$ to count the iterations of the policy gradient. At the first iteration $s = 0$ we already know that

$$s = 0, \ \forall t \in [T], l^{(t)} \in [d_t] : \sum_q (\sigma_{N_{t-1}+l^{(t)}}^{N_{t-2}+q})^2 = c_1^{(t)}(0) \tag{86}$$

because $\mathbf{W}(0) = \mathbf{1}c$. Then Lem. 5 gives us

$$s = 0, \ \forall t \in [T], l^{(t)} \in [d_t] : \mathop{\mathbb{E}}_{\mathbf{x}, \mathbf{y}^{(:t)}} \left[y_{l^{(t)}}^{(t)}\right] = c_2^{(t)}(0). \tag{87}$$

Thus, according to Eq. (85), the critical gradient component can be calculated as

- if $p \in \{i_1^{l^{(t)}}, i_2^{l^{(t)}}\}$, say $p = i_2^{l^{(t)}}$, then

$$\mathbb{E}_{\mathbf{x},\mathbf{y}^{(:t-1)}} \left[ \gamma_{l^{(t)}}^p(\mathbf{y}^{(:t-1)}) \right]$$

$$= \sum_{j=1}^{d_{t-1}} \sigma_{N_{t-1}+l^{(t)}}^{N_{t-2}+j} \mathop{\mathbb{E}}_{\mathbf{x},\mathbf{y}^{(:t-1)}} \left[ y_j^{(t-1)} y_{i_1^{l^{(t)}}}^{(t-1)} \right]$$

$$\overset{(i)}{=} \sum_{j \neq i_1^{l^{(t)}}} \sigma_{N_{t-1}+l^{(t)}}^{N_{t-2}+j} \mathop{\mathbb{E}}_{\mathbf{x},\mathbf{y}^{(:t-1)}} \left[ y_j^{(t-1)} \right] \mathop{\mathbb{E}}_{\mathbf{x},\mathbf{y}^{(:t-1)}} \left[ y_{i_1^{l^{(t)}}}^{(t-1)} \right] + \sigma_{N_{t-1}+l^{(t)}}^{N_{t-2}+i_1^{l^{(t)}}} \tag{88}$$

$$\overset{(ii)}{=} \left( c_2^{(t-1)}(0) \right)^2 + \sigma_{N_{t-1}+l^{(t)}}^{N_{t-2}+i_1^{l^{(t)}}} \left( 1 - \left( c_2^{(t-1)}(0) \right)^2 \right),$$

where we use that $y_j^{(t)}$ and $y_{j'}^{(t)}$ are independent if $j \neq j'$ in $(i)$ and $\sum_j \sigma_{N_{t-1}+l^{(t)}}^{N_{t-2}+j} = 1$ in $(ii)$. Similarly, if $p = i_1^{l^{(t)}}$,

$$\mathbb{E}_{\mathbf{x},\mathbf{y}^{(:t-1)}} \left[ \gamma_{l^{(t)}}^p(\mathbf{y}^{(:t-1)}) \right]$$

$$= \sum_{j \neq i_2^{l^{(t)}}} \sigma_{N_{t-1}+l^{(t)}}^{N_{t-2}+j} \mathop{\mathbb{E}}_{\mathbf{x},\mathbf{y}^{(:t-1)}} \left[ y_j^{(t-1)} \right] \mathop{\mathbb{E}}_{\mathbf{x},\mathbf{y}^{(:t-1)}} \left[ y_{i_2^{l^{(t)}}}^{(t-1)} \right] + \sigma_{N_{t-1}+l^{(t)}}^{N_{t-2}+i_2^{l^{(t)}}} \tag{89}$$

$$= \left( c_2^{(t-1)}(0) \right)^2 + \sigma_{N_{t-1}+l^{(t)}}^{N_{t-2}+i_2^{l^{(t)}}} \left( 1 - \left( c_2^{(t-1)}(0) \right)^2 \right).$$

- if $p \notin \{i_1^{l^{(t)}}, i_2^{l^{(t)}}\}$, then (similar to the case of $p \in \{i_1^{l^{(t)}}, i_2^{l^{(t)}}\}$)

$$\mathbb{E}_{\mathbf{x},\mathbf{y}^{(:t-1)}} \left[ \gamma_{l^{(t)}}^p(\mathbf{y}^{(:t-1)}) \right]$$

$$= \sum_j \sigma_{N_{t-1}+l^{(t)}}^{N_{t-2}+j} \mathop{\mathbb{E}}_{\mathbf{x},\mathbf{y}^{(:t-1)}} \left[ y_j^{(t-1)} y_p^{(t-1)} y_{i_1^{l^{(t)}}}^{(t-1)} y_{i_2^{l^{(t)}}}^{(t-1)} \right]$$

$$= \sum_{j \neq i_1^{l^{(t)}}, i_2^{l^{(t)}}, p} \sigma_{N_{t-1}+l^{(t)}}^{N_{t-2}+j} \mathop{\mathbb{E}}_{\mathbf{x},\mathbf{y}^{(:t-1)}} \left[ y_j^{(t-1)} y_p^{(t-1)} y_{i_1^{l^{(t)}}}^{(t-1)} y_{i_2^{l^{(t)}}}^{(t-1)} \right] + \sigma_{N_{t-1}+l^{(t)}}^{N_{t-2}+p} \mathop{\mathbb{E}}_{\mathbf{x},\mathbf{y}^{(:t-1)}} \left[ y_{i_1^{l^{(t)}}}^{(t-1)} y_{i_2^{l^{(t)}}}^{(t-1)} \right]$$

$$+ \sigma_{N_{t-1}+l^{(t)}}^{N_{t-2}+i_1^{l^{(t)}}} \mathop{\mathbb{E}}_{\mathbf{x},\mathbf{y}^{(:t-1)}} \left[ y_p^{(t-1)} y_{i_1^{l^{(t)}}}^{(t-1)} \right] + \sigma_{N_{t-1}+l^{(t)}}^{N_{t-2}+i_2^{l^{(t)}}} \mathop{\mathbb{E}}_{\mathbf{x},\mathbf{y}^{(:t-1)}} \left[ y_p^{(t-1)} y_{i_2^{l^{(t)}}}^{(t-1)} \right]$$

$$= \left( c_2^{(t-1)}(0) \right)^2 \left[ \left( c_2^{(t-1)}(0) \right)^2 \right.$$

$$\left. + \left( 1 - \left( c_2^{(t-1)}(0) \right)^2 \right) \left( \sigma_{N_{t-1}+l^{(t)}}^{N_{t-2}+p} + \sigma_{N_{t-1}+l^{(t)}}^{N_{t-2}+i_1^{l^{(t)}}} + \sigma_{N_{t-1}+l^{(t)}}^{N_{t-2}+i_2^{l^{(t)}}} \right) \right]. \tag{90}$$

Since we assume $\boldsymbol{W}(0) = \mathbf{1}c$, all $\sigma_{N_{t-1}+l^{(t)}}^{N_{t-2}+p}$ for different $p \in [d_{t-1}]$ are equal and we can easily conclude that for $p \in \{i_1^{l^{(t)}}, i_2^{l^{(t)}}\}, p' \notin \{i_1^{l^{(t)}}, i_2^{l^{(t)}}\}$,

$$\partial_{W_{N_{t-2}+p, N_{t-1}+l^{(t)}}} R - \partial_{W_{N_{t-2}+p', N_{t-1}+l^{(t)}}} R$$

$$= \frac{1 - \left( c_2^{(t-1)}(0) \right)^2}{d_t} \frac{(d_t - 3) \left( c_2^{(t-1)}(0) \right)^2 + 1}{d_t} > 0. \tag{91}$$

This means that the separation of the critical gradient component and that of the gradient are satisfied. Therefore, as shown in Thm. 3.1, only the parameters in the position $\{i_1^{l^{(t)}}, i_2^{l^{(t)}}\}$ become larger and all other parameters become smaller after the first step of the sign policy gradient:

$$s = 1, \forall t \in [T], l^{(t)} \in [d_t] : W_{N_{t-2}+p, N_{t-1}+l^{(t)}} = \begin{cases} 1 + \eta, & p \in \{i_1^{l^{(t)}}, i_2^{l^{(t)}}\}, \\ 1 - \eta, & \text{otherwise}. \end{cases} \tag{92}$$

After the softmax we obtain

$$s = 1, \forall t \in [T], l^{(t)} \in [d_t] : \sigma_{N_{t-1}+l^{(t)}}^{N_{t-2}+p} = \begin{cases} \frac{1}{2} \frac{1}{1 + \frac{d_t - 2}{2} e^{-2\eta}} := \lambda_1^{l^{(t)}}, & p \in \{i_1^{l^{(t)}}, i_2^{l^{(t)}}\}, \\ \frac{1}{d_t - 2 + 2e^{2\eta}} := \lambda_2^{l^{(t)}}, & \text{otherwise}, \end{cases} \tag{93}$$

and we conclude that

$$s = 1, \forall t \in [T], l^{(t)} \in [d_t] : \sum_q (\sigma_{N_{t-1}+l^{(t)}}^{N_{t-2}+q})^2 = c_1^{(t)}(1) \tag{94}$$

by conducting simple algebra. This means that the condition Eq. (83) still holds at $s = 1$. Therefore, Lem. 5 can be applied again, and, following the computation of Eq. (88) and Eq. (90), we are able to conclude that at $s = 1$:

- if $p \in \{i_1^{l^{(t)}}, i_2^{l^{(t)}}\}$, then

$$\sigma_{N_{t-1}+l^{(t)}}^{N_{t-2}+p} \mathbb{E}_{\mathbf{x}, \mathbf{y}^{(:t-1)}} \left[ \gamma_{l^{(t)}}^p (\mathbf{y}^{(:t-1)}) \right]$$
$$= \lambda_1^{l^{(t)}} \left( c_2^{(t-1)}(1) \right)^2 + \left( \lambda_1^{l^{(t)}} \right)^2 \left( 1 - \left( c_2^{(t-1)}(1) \right)^2 \right); \tag{95}$$

- if $p \notin \{i_1^{l^{(t)}}, i_2^{l^{(t)}}\}$, then

$$\sigma_{N_{t-1}+l^{(t)}}^{N_{t-2}+p} \mathbb{E}_{\mathbf{x}, \mathbf{y}^{(:t-1)}} \left[ \gamma_{l^{(t)}}^p (\mathbf{y}^{(:t-1)}) \right]$$
$$= \lambda_2^{l^{(t)}} \left( c_2^{(t-1)}(1) \right)^2 \left[ \left( c_2^{(t-1)}(1) \right)^2 + \left( 1 - \left( c_2^{(t-1)}(1) \right)^2 \right) \left( \lambda_2^{l^{(t)}} + 2\lambda_1^{l^{(t)}} \right) \right]. \tag{96}$$

According to the form of the policy gradient Eq. (32), we also need to evaluate

$$\mathbb{E}_{\mathbf{x}, \mathbf{y}^{(:t-1)}} \left[ \sum_{i=1}^{d_{t-1}} \gamma_{l^{(t)}}^i (\mathbf{y}^{(t-1)}) \right] \sigma_{N_{t-1}+l^{(t)}}^{N_{t-2}+i}$$
$$= 2\lambda_1^{l^{(t)}} \left[ \left( c_2^{(t-1)}(1) \right)^2 + \lambda_1^{l^{(t)}} \left( 1 - \left( c_2^{(t-1)}(1) \right)^2 \right) \right] \tag{97}$$
$$+ (d_t - 2)\lambda_2^{l^{(t)}} \left( c_2^{(t-1)}(1) \right)^2 \left[ \left( c_2^{(t-1)}(1) \right)^2 + \left( 1 - \left( c_2^{(t-1)}(1) \right)^2 \right) \left( \lambda_2^{l^{(t)}} + 2\lambda_1^{l^{(t)}} \right) \right].$$

Combined these equations, noting that $2\lambda_1^{l^{(t)}} + (d_t - 2)\lambda_2^{l^{(t)}} = 1$, we are now able to find the expression of the gradient when $p \in \{i_1^{l^{(t)}}, i_2^{l^{(t)}}\}$:

$$\partial_{W_{N_{t-2}+p, N_{t-1}+l^{(t)}}} R$$
$$= \lambda_1^{l^{(t)}} \left( 1 - 2\lambda_1^{l^{(t)}} \right) \left\{ \left( c_2^{(t-1)}(1) \right)^2 + \lambda_1^{l^{(t)}} \left( 1 - \left( c_2^{(t-1)}(1) \right)^2 \right) \right.$$
$$\left. - \left( c_2^{(t-1)}(1) \right)^2 \left[ \left( c_2^{(t-1)}(1) \right)^2 + \left( 1 - \left( c_2^{(t-1)}(1) \right)^2 \right) \left( \lambda_2^{l^{(t)}} + 2\lambda_1^{l^{(t)}} \right) \right] \right\}$$
$$= \lambda_1^{l^{(t)}} \left( 1 - 2\lambda_1^{l^{(t)}} \right) \left( 1 - \left( c_2^{(t-1)}(1) \right)^2 \right) \left[ \left( c_2^{(t-1)}(1) \right)^2 (d_t - 3) \lambda_2^{l^{(t)}} \left( c_2^{(t-1)}(1) \right)^2 + \lambda_1^{l^{(t)}} \right]$$
$$> 0, \tag{98}$$

as $d_t > 3$ (the case for $d_t = 2$ does not need any update) and $2\lambda_1^{l^{(t)}} < 1$. Hence the policy has positive gradient at relevant positions (Eq. (98)) and negative gradient at irrelevant positions (using Eq. (37)). Similar to the proof of Thm. 3.1, we can verify that

$$s = 2, \forall t \in [T], l^{(t)} \in [d_t] : W_{N_{t-2}+p, N_{t-1}+l^{(t)}} = \begin{cases} 1 + 2\eta, & p \in \{i_1^{l^{(t)}}, i_2^{l^{(t)}}\}, \\ 1 - 2\eta, & \text{otherwise}. \end{cases} \tag{99}$$

Following the same argument, we can eventually show that after $S$-steps of the sign policy gradient update the model parameters are simply

$$W_{N_{t-2}+p, N_{t-1}+l^{(t)}}(S) = \begin{cases} 1 + \eta S, & p \in \{i_1^{l^{(t)}}, i_2^{l^{(t)}}\}, \\ 1 - \eta S, & \text{otherwise .} \end{cases} \tag{100}$$

After the softmax we obtain

$$\sigma_{N_{t-1}+l^{(t)}}^{N_{t-2}+p}(S) = \begin{cases} \frac{1}{2} \frac{1}{1 + \frac{d_t-2}{2}e^{-2\eta S}}, & p \in \{i_1^{l^{(t)}}, i_2^{l^{(t)}}\}, \\ \frac{1}{d_t - 2 + 2e^{2\eta S}}, & \text{otherwise .} \end{cases} \tag{101}$$

$\square$

*Proof of Lem. 5.* We prove this lemma by induction. For $t = 0$, we directly have

$$\forall l^{(0)} \in [d_0] : \mathbb{E}_{\mathbf{x}}[x_{l^{(0)}}] = 0. \tag{102}$$

Denote

$$\chi_{l^{(t)}}(\mathbf{y}^{(t-1)}) = p_{\mathbf{W}}\left(y_{l^{(t)}}^{(t)} = 1 \middle| \mathbf{y}^{(t-1)}\right) = (\xi_{l^{(t)}})^2. \tag{103}$$

We first verify the claim for $t = 1$, where the expectation can be written as

$$\begin{aligned}
\mathbb{E}_{\mathbf{x}, \mathbf{y}^{(1)}}\left[y_{l^{(1)}}^{(1)}\right] &= \sum_{\mathbf{x}} p(\mathbf{x})\left(2\chi_{l^{(1)}}(\mathbf{x}) - 1\right) \\
&\overset{(i)}{=} 2\sum_{\mathbf{x}} p(\mathbf{x})\left(\sum_{p,q} x_p x_q \sigma_{N_0+l^{(1)}}^{N_{-1}+p} \sigma_{N_0+l^{(1)}}^{N_{-1}+q}\right) - 1 \\
&\overset{(ii)}{=} 2\sum_{p \neq q} \mathbb{E}_{\mathbf{x}}[x_p] \mathbb{E}_{\mathbf{x}}[x_q] \sigma_{N_0+l^{(1)}}^{N_{-1}+p} \sigma_{N_0+l^{(1)}}^{N_{-1}+q} + 2\sum_m (\sigma_{N_0+l^{(1)}}^{N_{-1}+m})^2 - 1 \\
&\overset{(iii)}{=} 2\sum_m (\sigma_{N_0+l^{(1)}}^{N_{-1}+m})^2 - 1 = 2c_1^{(t)} - 1,
\end{aligned} \tag{104}$$

where we use the definition of $\xi_{l^{(1)}}$ in $(i)$; we use that $x_j$ for different $j$ are independent w.r.t $p(\mathbf{x})$ in $(ii)$; we apply $\mathbb{E}_{\mathbf{x}}[x_p] = 0$ in $(iii)$. Therefore, with the condition Eq. (83), the statement is established for $t = 1$. Suppose that the statement is established for $(t-1)$-th step

$$\forall l^{(t-1)} \in [d^{(t-1)}] : \mathbb{E}_{\mathbf{x}, \mathbf{y}^{(:t-1)}}\left[y_{l^{(t-1)}}^{(t-1)}\right] = c_2^{(t-1)}, \tag{105}$$

then

$$\begin{aligned}
&\mathbb{E}_{\mathbf{x}, \mathbf{y}^{(:t)}}\left[y_{l^{(t)}}^{(t)}\right] \\
&= \sum_{\mathbf{x}} p(\mathbf{x}) \sum_{\mathbf{y}^{(:t-1)}} p_{\mathbf{W}}(\mathbf{y}^{(:t-1)}|\mathbf{x}) \sum_{\mathbf{y}^{(t)}} p_{\mathbf{W}}(\mathbf{y}^{(t)}|\mathbf{y}^{(t-1)}) y_{l^{(t)}}^{(t)} \\
&= 2\sum_{\mathbf{x}} p(\mathbf{x}) \sum_{\mathbf{y}^{(:t-1)}} p_{\mathbf{W}}(\mathbf{y}^{(:t-1)}|\mathbf{x}) \chi_{l^{(t)}}(\mathbf{y}^{(t-1)}) - 1 \\
&= 2\sum_{p^{(t-1)} \neq q^{(t-1)}} \mathbb{E}_{\mathbf{x}, \mathbf{y}^{(:t-1)},}\left[y_{p^{(t-1)}}^{(t-1)}\right] \mathbb{E}_{\mathbf{x}, \mathbf{y}^{(:t-1)},}\left[y_{q^{(t-1)}}^{(t-1)}\right] \sigma_{N_{t-1}+l^{(t)}}^{N_{t-2}+p^{(t-1)}} \sigma_{N_{t-1}+l^{(t)}}^{N_{t-2}+q^{(t-1)}} \\
&\quad + 2\sum_{m^{(t-1)}} \left(\sigma_{N_{t-1}+l^{(t)}}^{N_{t-2}+m^{(t-1)}}\right)^2 - 1 \\
&= 2\left(c_2^{(t-1)}\right)^2 \sum_{p^{(t-1)} \neq q^{(t-1)}} \sigma_{N_{t-1}+l^{(t)}}^{N_{t-2}+p^{(t-1)}} \sigma_{N_{t-1}+l^{(t)}}^{N_{t-2}+q^{(t-1)}} + 2\sum_{m^{(t-1)}} \left(\sigma_{N_{t-1}+l^{(t)}}^{N_{t-2}+m^{(t-1)}}\right)^2 - 1 \\
&= 2\left(c_2^{(t-1)}\right)^2 \left(\sum_{p^{(t-1)}} \sigma_{N_{t-1}+l^{(t)}}^{N_{t-2}+p^{(t-1)}}\right)^2 + 2\left[1 - \left(c_2^{(t-1)}\right)^2\right] \sum_{m^{(t-1)}} \left(\sigma_{N_{t-1}+l^{(t)}}^{N_{t-2}+m^{(t-1)}}\right)^2 - 1 \\
&= 2\left(c_2^{(t-1)}\right)^2 + 2\left[1 - \left(c_2^{(t-1)}\right)^2\right] \left(c_1^{(t-1)}\right)^2 - 1
\end{aligned}$$

does not depend on $l^{(t)}$. Thus we conclude the proof. $\square$

### D.2.2 LEARNABILITY OF SFT: PROOF OF CLAIM. 4.1

*Proof.* To prove this claim, we only need to evaluate whether the separation of the critical gradient component is satisfied, which can be done by inspecting the forms of $\gamma^p_{l^{(t)}}$ similar to the case for RL. Specifically, according to Eq. (85) and recalling that $\tilde{\mathbf{y}}^{(t)}$ is the ground-truth of $\mathbf{y}^{(t)}$ given $\mathbf{x}$, we have

$$\mathbb{E}_{\mathbf{x}}\left[\gamma^p_{l^{(t)}}(\tilde{\mathbf{y}}^{(t-1)})\right] := \mathbb{E}_{\mathbf{x}}\left[\left(\sum_{j=1}^{d_{t-1}} \sigma^{N_{t-2}+j}_{N_{t-1}+l^{(t)}} \tilde{y}^{(t-1)}_j\right) \tilde{y}^{(t-1)}_{i_1^{l^{(t)}}} \tilde{y}^{(t-1)}_{i_2^{l^{(t)}}} \tilde{y}^{(t-1)}_p\right]. \tag{106}$$

As this is similar to the case for RL, we can easily evaluate it at the relevant positions and irrelevant ones, respectively, as shown below:

- if $p \in \{i_1^{l^{(t)}}, i_2^{l^{(t)}}\}$, say $p = i_2^{l^{(t)}}$, then

$$\mathbb{E}_{\mathbf{x}}\left[\gamma^p_{l^{(t)}}(\tilde{\mathbf{y}}^{(t-1)})\right]$$
$$= \sum_{j=1}^{d_{t-1}} \sigma^{N_{t-2}+j}_{N_{t-1}+l^{(t)}} \mathbb{E}_{\mathbf{x}}\left[\tilde{y}^{(t-1)}_j \tilde{y}^{(t-1)}_{i_1^{l^{(t)}}}\right]$$
$$\overset{(i)}{=} \sum_{j \neq i_1^{l^{(t)}}} \sigma^{N_{t-2}+j}_{N_{t-1}+l^{(t)}} \mathbb{E}_{\mathbf{x}}\left[\tilde{y}^{(t-1)}_j\right] \mathbb{E}_{\mathbf{x}}\left[\tilde{y}^{(t-1)}_{i_1^{l^{(t)}}}\right] + \sigma^{N_{t-2}+i_1^{l^{(t)}}}_{N_{t-1}+l^{(t)}}$$
$$\overset{(ii)}{=} \sigma^{N_{t-2}+i_1^{l^{(t)}}}_{N_{t-1}+l^{(t)}}, \tag{107}$$

where we use that $\tilde{y}^{(t)}_j$ and $\tilde{y}^{(t)}_{j'}$ are independent if $j \neq j'$ in $(i)$ and $\mathbb{E}_{\mathbf{x}}[\tilde{y}^{(t-1)}_j] = 0$ in $(ii)$. Similarly, if $p = i_1^{l^{(t)}}$, we can similarly obtain that

$$\mathbb{E}_{\mathbf{x}}\left[\gamma^p_{l^{(t)}}(\tilde{\mathbf{y}}^{(t-1)})\right] = \sigma^{N_{t-2}+i_2^{l^{(t)}}}_{N_{t-1}+l^{(t)}}. \tag{108}$$

- if $p \notin \{i_1^{l^{(t)}}, i_2^{l^{(t)}}\}$, then (similar to the case of $p \in \{i_1^{l^{(t)}}, i_2^{l^{(t)}}\}$)

$$\mathbb{E}_{\mathbf{x}}\left[\gamma^p_{l^{(t)}}(\tilde{\mathbf{y}}^{(t-1)})\right]$$
$$= \sum_{j} \sigma^{N_{t-2}+j}_{N_{t-1}+l^{(t)}} \mathbb{E}_{\mathbf{x}}\left[\tilde{y}^{(t-1)}_j y^{(t-1)}_p \tilde{y}^{(t-1)}_{i_1^{l^{(t)}}} \tilde{y}^{(t-1)}_{i_2^{l^{(t)}}}\right]$$
$$= \sum_{j \neq i_1^{l^{(t)}}, i_2^{l^{(t)}}, p} \sigma^{N_{t-2}+j}_{N_{t-1}+l^{(t)}} \mathbb{E}_{\mathbf{x}}\left[\tilde{y}^{(t-1)}_j y^{(t-1)}_p \tilde{y}^{(t-1)}_{i_1^{l^{(t)}}} \tilde{y}^{(t-1)}_{i_2^{l^{(t)}}}\right] + \sigma^{N_{t-2}+p}_{N_{t-1}+l^{(t)}} \mathbb{E}_{\mathbf{x}}\left[\tilde{y}^{(t-1)}_{i_1^{l^{(t)}}} \tilde{y}^{(t-1)}_{i_2^{l^{(t)}}}\right]$$
$$+ \sigma^{N_{t-2}+i_1^{l^{(t)}}}_{N_{t-1}+l^{(t)}} \mathbb{E}_{\mathbf{x}}\left[\tilde{y}^{(t-1)}_p \tilde{y}^{(t-1)}_{i_1^{l^{(t)}}}\right] + \sigma^{N_{t-2}+i_2^{l^{(t)}}}_{N_{t-1}+l^{(t)}} \mathbb{E}_{\mathbf{x}}\left[\tilde{y}^{(t-1)}_p \tilde{y}^{(t-1)}_{i_2^{l^{(t)}}}\right] = 0. \tag{109}$$

Therefore, for any $t \in [T]$ and any $l^{(t)} \in [d_t]$,

$$\forall p \in \{i_1^{l^{(t)}}, i_2^{l^{(t)}}\}, p' \in [d_{t-1}]\backslash\{i_1^{l^{(t)}}, i_2^{l^{(t)}}\} : \mathbb{E}_{\mathbf{x}}\left[\gamma^p_{l^{(t)}}(\tilde{\mathbf{y}}^{(t-1)}) - \gamma^{p'}_{l^{(t)}}(\tilde{\mathbf{y}}^{(t-1)})\right] > 0, \tag{110}$$

which is exactly the separation condition in Thm. 3.2, and hence the learnability of SFT is established for $k$-PARITY. $\qquad\square$

### D.3 PROOFS OF SEC. 4.2

We discuss the learnability of $k$-AND for transformers via RL or SFT in App. D.3.1, and discuss that of $k$-OR in App. D.3.2.

### D.3.1 AND

As in the case for $k$-PARITY, we start with evaluating the formulation of the critical gradient component $\gamma_{l^{(t)}}^p(\mathbf{y}^{(t-1)})$, which can then be used to verify whether the separation condition in Thm. 3.1 or Thm. 3.2 is satisfied to indicate the learnability. Recall that for $k$-AND (Sec. 2.1),

$$\psi'(\xi_{l^{(t)}}) = \mathbb{I}(\xi_{l^{(t)}} > 0), \tag{111}$$

$$\phi\left(y_{i_1^{l^{(t)}}}^{(t-1)}, y_{i_2^{l^{(t)}}}^{(t-1)}\right) = \frac{y_{i_1^{l^{(t)}}}^{(t-1)}y_{i_2^{l^{(t)}}}^{(t-1)} + y_{i_1^{l^{(t)}}}^{(t-1)} + y_{i_2^{l^{(t)}}}^{(t-1)} - 1}{2}, \tag{112}$$

where $\mathbb{I}(\cdot)$ is the indictor function. This gives us the expression of $\gamma$ (see Eq. (9) for its definition) as follows

$$\gamma_{l^{(t)}}^p(\mathbf{y}^{(t-1)})$$
$$= \mathbb{I}(\xi_{l^{(t)}} > 0)\left(y_{i_1^{l^{(t)}}}^{(t-1)}y_{i_2^{l^{(t)}}}^{(t-1)}y_p^{(t-1)} + y_{i_1^{l^{(t)}}}^{(t-1)}y_p^{(t-1)} + y_{i_2^{l^{(t)}}}^{(t-1)}y_p^{(t-1)} - y_p^{(t-1)}\right). \tag{113}$$

We now check the separation for the critical gradient component $\gamma_{l^{(t)}}^p$.

**RL.** Given $t \in [T]$, when $p \in \{i_1^{l^{(t)}}, i_2^{l^{(t)}}\}$, say $i_2^{l^{(t)}}$, according to Eq. (113), we have (the case for $i_1^{l^{(t)}}$ is similar)

$$\mathbb{E}_{\mathbf{y}^{(:t-1)}}[\gamma_{l^{(t)}}^p(\mathbf{y}^{(t-1)})] = \frac{1}{2}\mathbb{E}_{\mathbf{y}^{(:t-1)}}\left[\mathbb{I}(\xi_{l^{(t)}} > 0)y_{i_1^{l^{(t)}}}^{(t-1)}y_{i_2^{l^{(t)}}}^{(t-1)}\right] + \frac{1}{2}\mathbb{E}_{\mathbf{y}^{(:t-1)}}\left[\mathbb{I}(\xi_{l^{(t)}} > 0)\right]; \tag{114}$$

as a comparison, when $p \notin \{i_1^{l^{(t)}}, i_2^{l^{(t)}}\}$, we obtain that

$$\mathbb{E}_{\mathbf{y}^{(:t-1)}}\left[\gamma_{l^{(t)}}^p(\mathbf{y}^{(t-1)})\right]$$
$$= \frac{1}{2}\mathbb{E}_{\mathbf{y}^{(:t-1)}}\left[\mathbb{I}(\xi_{l^{(t)}} > 0)\left(y_{i_1^{l^{(t)}}}^{(t-1)}y_{i_2^{l^{(t)}}}^{(t-1)}y_p^{(t-1)} + y_{i_1^{l^{(t)}}}^{(t-1)}y_p^{(t-1)} + y_{i_2^{l^{(t)}}}^{(t-1)}y_p^{(t-1)} - y_p^{(t-1)}\right)\right]. \tag{115}$$

It is now left for us to compare the difference between these critical gradient components. For convenience and ease of notation, we define $A := \{i_1^{l^{(t)}}, i_2^{l^{(t)}}, p\}$ and $\mathbf{y}_{\not{A}}^{(t-1)}$ as the sequence $\mathbf{y}^{(t-1)}$ without the tokens $y_{i_1^{l^{(t)}}}^{(t-1)}, y_{i_2^{l^{(t)}}}^{(t-1)}$, and $y_p^{(t-1)}$. In addition, let $D_\lambda := \{\mathbf{y}_{\not{A}}^{(t-1)} | \sum_j y_{\not{A},j}^{(t-1)} \geq \lambda\}$, i.e., the sum of tokens of $\mathbf{y}_{\not{A}}^{(t-1)}$ is larger than or equal to $\lambda$. Finally, we simplify $p_W$ as $p$ with $p_\pm = p(y_j^{(t-1)} = \pm 1 | \mathbf{y}^{(:t-2)})$. With these new symbols, we define four random variables depending on $\mathbf{y}^{(:t-2)}$ as follows:

$$a = \sum_{\mathbf{y}_{\not{A}}^{(t-1)} \in D_{-3}} p_+^3 p(\mathbf{y}_{\not{A}}^{(t-1)} | \mathbf{y}^{(:t-2)}), \tag{116}$$

$$b = \sum_{\mathbf{y}_{\not{A}}^{(t-1)} \in D_{-1}} p_+^2 p_- p(\mathbf{y}_{\not{A}}^{(t-1)} | \mathbf{y}^{(:t-2)}), \tag{117}$$

$$c = \sum_{\mathbf{y}_{\not{A}}^{(t-1)} \in D_{+1}} p_+ p_-^2 p(\mathbf{y}_{\not{A}}^{(t-1)} | \mathbf{y}^{(:t-2)}), \tag{118}$$

$$d = \sum_{\mathbf{y}_{\not{A}}^{(t-1)} \in D_{+3}} p_-^3 p(\mathbf{y}_{\not{A}}^{(t-1)} | \mathbf{y}^{(:t-2)}). \tag{119}$$

In Eq. (114) and Eq. (115), all $y_j$ can be treated equally as they are independent of each other. We omit all the subscripts and write the difference between $p \in \{i_1^{l^{(t)}}, i_2^{l^{(t)}}\}$ and $p' \in [d_{t-1}]\setminus\{i_1^{l^{(t)}}, i_2^{l^{(t)}}\}$

as

$$\mathbb{E}_{\mathbf{y}^{(:t-1)}}[\Delta] := \mathbb{E}_{\mathbf{y}^{(:t-1)}}[\gamma_{l^{(t)}}^{p}(\mathbf{y}^{(t-1)})] - \mathbb{E}_{\mathbf{y}^{(:t-1)}}[\gamma_{l^{(t)}}^{p'}(\mathbf{y}^{(t-1)})]$$

$$\propto \underbrace{\mathbb{E}_{\mathbf{y}^{(:t-1)}}[\mathbb{I}(\xi_{l^{(t)}} > 0)]}_{(i)} + \underbrace{\mathbb{E}_{\mathbf{y}^{(:t-1)}}[\mathbb{I}(\xi_{l^{(t)}} > 0)y]}_{(ii)} \tag{120}$$

$$- \underbrace{\mathbb{E}_{\mathbf{y}^{(:t-1)}}[\mathbb{I}(\xi_{l^{(t)}} > 0)yy]}_{(iii)} - \underbrace{\mathbb{E}_{\mathbf{y}^{(:t-1)}}[\mathbb{I}(\xi_{l^{(t)}} > 0)yyy]}_{(iv)}.$$

We below analyze each term of $\Delta$. The first term and the last term can be easily evaluated as follows:

$$(i) = \sum_{\mathbf{y}^{(:t-2)}} p(\mathbf{y}^{(:t-2)}) \left[ \sum_{\mathbf{y}_{\not{A}}^{(t-1)} \in D_{-3}} p_+^3 p(\mathbf{y}_{\not{A}}^{(t-1)} | \mathbf{y}^{(:t-2)}) \right.$$

$$+ 3 \sum_{\mathbf{y}_{\not{A}}^{(t-1)} \in D_{-1}} p_+^2 p_- p(\mathbf{y}_{\not{A}}^{(t-1)} | \mathbf{y}^{(:t-2)}) \tag{121}$$

$$\left. + 3 \sum_{\mathbf{y}_{\not{A}}^{(t-1)} \in D_{+1}} p_+ p_-^2 p(\mathbf{y}_{\not{A}}^{(t-1)} | \mathbf{y}^{(:t-2)}) + \sum_{\mathbf{y}_{\not{A}}^{(t-1)} \in D_{+3}} p_-^3 p(\mathbf{y}_{\not{A}}^{(t-1)} | \mathbf{y}^{(:t-2)}) \right]$$

$$= \mathbb{E}_{\mathbf{y}^{(:t-2)}}[a + 3b + 3c + d],$$

and

$$(iv) = \sum_{\mathbf{y}^{(:t-2)}} p(\mathbf{y}^{(:t-2)}) \left[ \sum_{\mathbf{y}_{\not{A}}^{(t-1)} \in D_{-3}} p_+^3 p(\mathbf{y}_{\not{A}}^{(t-1)} | \mathbf{y}^{(:t-2)}) \right.$$

$$- 3 \sum_{\mathbf{y}_{\not{A}}^{(t-1)} \in D_{-1}} p_+^2 p_- p(\mathbf{y}_{\not{A}}^{(t-1)} | \mathbf{y}^{(:t-2)}) + 3 \sum_{\mathbf{y}_{\not{A}}^{(t-1)} \in D_{+1}} p_+ p_-^2 p(\mathbf{y}_{\not{A}}^{(t-1)} | \mathbf{y}^{(:t-2)}) \tag{122}$$

$$\left. - \sum_{\mathbf{y}_{\not{A}}^{(t-1)} \in D_{+3}} p_-^3 p(\mathbf{y}_{\not{A}}^{(t-1)} | \mathbf{y}^{(:t-2)}) \right]$$

$$= \mathbb{E}_{\mathbf{y}^{(:t-2)}}[a - 3b + 3c - d].$$

For the rest terms, we express them with the four constants $a, b, c, d$ defined above such that we are able to evaluate $\Delta$. To this end, let $\mathbf{y}_{\not{Y}}^{(t-1)}$ denote the sequence obtained by removing one token in $A$ from $\mathbf{y}^{(t-1)}$, $\mathbf{y}_{\not{2}}^{(t-1)}$ be obtained by removing two tokens of $A$ from $\mathbf{y}^{(t-1)}$, and let $D_\lambda$. This allows us to establish the following relation:

$$\sum_{\mathbf{y}_{\not{Y}}^{(t-1)} \in D_0} p(\mathbf{y}_{\not{Y}}^{(t-1)} | \mathbf{y}^{(:t-2)})$$

$$= \sum_{\mathbf{y}_{\not{2}}^{(t-1)} \in D_{-1}} p_+ p(\mathbf{y}_{\not{2}}^{(t-1)} | \mathbf{y}^{(:t-2)}) + \sum_{\mathbf{y}_{\not{2}}^{(t-1)} \in D_{+1}} p_- p(\mathbf{y}_{\not{2}}^{(t-1)} | \mathbf{y}^{(:t-2)}), \tag{123}$$

which can be further decomposed to be finally expressed by $a, b, c,$ and $d$. With this relation, we are able to derive

$$(ii) = \sum_{\mathbf{y}^{(:t-2)}} p(\mathbf{y}^{(:t-2)}) \left[ \sum_{\mathbf{y}_{\not{Y}}^{(t-1)} \in D_{-1}} p_+ p(\mathbf{y}_{\not{Y}}^{(t-1)} | \mathbf{y}^{(:t-2)}) - \sum_{\mathbf{y}_{\not{Y}}^{(t-1)} \in D_{+1}} p_- p(\mathbf{y}_{\not{Y}}^{(t-1)} | \mathbf{y}^{(:t-2)}) \right]$$

$$= \sum_{\mathbf{y}^{(:t-2)}} p(\mathbf{y}^{(:t-2)}) \left[ \sum_{\mathbf{y}_{\not{A}}^{(t-1)} \in D_{-3}} p_+^3 p(\mathbf{y}_{\not{A}}^{(t-1)} | \mathbf{y}^{(:t-2)}) + \sum_{\mathbf{y}_{\not{A}}^{(t-1)} \in D_{-1}} p_+^2 p_- p(\mathbf{y}_{\not{A}}^{(t-1)} | \mathbf{y}^{(:t-2)}) \right.$$

$$\left. - \sum_{\mathbf{y}_{\not{A}}^{(t-1)} \in D_{+1}} p_+ p_-^2 p(\mathbf{y}_{\not{A}}^{(t-1)} | \mathbf{y}^{(:t-2)}) - \sum_{\mathbf{y}_{\not{A}}^{(t-1)} \in D_{+3}} p_-^3 p(\mathbf{y}_{\not{A}}^{(t-1)} | \mathbf{y}^{(:t-2)}) \right]$$

$$= \mathbb{E}_{\mathbf{y}^{(:t-2)}}[a + b - c - d]. \tag{124}$$

Similarly, for the second term we can write

$$
\begin{aligned}
(iii) &= \sum_{\mathbf{y}^{(:t-2)}} p(\mathbf{y}^{(:t-2)}) \Big[ \sum_{\mathbf{y}_{\cancel{2}}^{(t-1)} \in D_{-2}} p_+^2 p(\mathbf{y}_{\cancel{2}}^{(t-1)} | \mathbf{y}^{(:t-2)}) \\
&\quad - 2 \sum_{\mathbf{y}_{\cancel{2}}^{(t-1)} \in D_0} p_+ p_- p(\mathbf{y}_{\cancel{2}}^{(t-1)} | \mathbf{y}^{(:t-2)}) + \sum_{\mathbf{y}_{\cancel{2}}^{(t-1)} \in D_{+1}} p_-^2 p(\mathbf{y}_{\cancel{2}}^{(t-1)} | \mathbf{y}^{(:t-2)}) \\
&= \sum_{\mathbf{y}^{(:t-2)}} p(\mathbf{y}^{(:t-2)}) \Big[ \sum_{\mathbf{y}_{\cancel{A}}^{(t-1)} \in D_{-3}} p_+^3 p(\mathbf{y}_{\cancel{2}}^{(t-1)} | \mathbf{y}^{(:t-2)}) \\
&\quad - \sum_{\mathbf{y}_{\cancel{A}}^{(t-1)} \in D_{-1}} p_+^2 p_- p(\mathbf{y}_{\cancel{A}}^{(t-1)} | \mathbf{y}^{(:t-2)}) - \sum_{\mathbf{y}_{\cancel{A}}^{(t-1)} \in D^{+1}} p_+ p_-^2 p(\mathbf{y}_{\cancel{A}}^{(t-1)} | \mathbf{y}^{(:t-2)}) \\
&\quad + \sum_{\mathbf{y}_{\cancel{A}}^{(t-1)} \in D_{+3}} p_-^3 p(\mathbf{y}_{\cancel{A}}^{(t-1)} | \mathbf{y}^{(:t-2)}) \Big] \\
&= \mathbb{E}_{\mathbf{y}^{(:t-2)}}[a - b - c + d].
\end{aligned}
\tag{125}
$$

Now combining these terms, we can easily show that

$$
\Delta \propto 8b > 0;
\tag{126}
$$

hence the separation of the critical gradient component is satisfied, and the learnability is guaranteed for RL.

**SFT.** Compared to the case for fine-tuning via RL discussed above, the only difference brought by evaluating the critical gradient component for SFT is that we are now computing

$$
\mathbb{E}_{\mathbf{x}}[\gamma_{l^{(t)}}^p(\tilde{\mathbf{y}}^{(t-1)})],
\tag{127}
$$

where $\tilde{\mathbf{y}}^{(t-1)}$ is the ground-truth of $\mathbf{y}^{(t-1)}$. This is equivalent to changing $\mathbf{y}^{(t-1)}$ to $\tilde{\mathbf{y}}^{(t-1)}$ and all the dependence on $\mathbf{y}^{(:t-2)}$ to the dependence on $\mathbf{x}$. It can be easily seen that these do not change the conclusion that $\Delta \propto 8b > 0$ (as it is valid for any distribution $p(\mathbf{y}^{(t-1)} | \mathbf{y}^{(:t-2)})$ according to the calculation of Eq. (120), which includes $p_{\text{True}}(\mathbf{y}^{(t-1)} | \mathbf{x})$—the ground-truth distribution—as an example); thus the separation of the critical gradient component in Thm. 3.2 is also satisfied, which further guarantees the learnability via SFT.

### D.3.2   OR

**RL.** The analysis for $k$-OR is highly similar to that for $k$-AND in App. D.3.1, except for that $\psi(\cdot)$ and $\phi_2(\cdot, \cdot)$ have different formulations. Specifically,

$$
\psi'(\xi_{l^{(t)}}) = \mathbb{I}(\xi_{l^{(t)}} < 0),
\tag{128}
$$

$$
\phi\left(y_{i_1^{l^{(t)}}}^{(t-1)}, y_{i_2^{l^{(t)}}}^{(t-1)}\right) = \frac{-y_{i_1^{l^{(t)}}}^{(t-1)} y_{i_2^{l^{(t)}}}^{(t-1)} + y_{i_1^{l^{(t)}}}^{(t-1)} + y_{i_2^{l^{(t)}}}^{(t-1)} + 1}{2}.
\tag{129}
$$

This gives us the expression for $\gamma$

$$
\begin{aligned}
&\mathbb{E}_{\mathbf{y}^{(:t-1)}}[\gamma_{l^{(t)}}^p] \\
&:= \frac{1}{2} \mathbb{E}_{\mathbf{y}^{(:t-1)}} \left[ \mathbb{I}(\xi_{l^{(t)}} < 0) \left( -y_{i_1^{l^{(t)}}}^{(t-1)} y_{i_2^{l^{(t)}}}^{(t-1)} y_p^{(t-1)} + y_{i_1^{l^{(t)}}}^{(t-1)} y_p^{(t-1)} + y_{i_2^{l^{(t)}}}^{(t-1)} y_p^{(t-1)} + y_p^{(t-1)} \right) \right].
\end{aligned}
\tag{130}
$$

When $p \in \{i_1^{l^{(t)}}, i_2^{l^{(t)}}\}$, say $j_2$,

$$
\begin{aligned}
\mathbb{E}_{\mathbf{y}^{(:t-1)}}[\gamma_{l^{(t)}}^p] &:= \frac{1}{2} \mathbb{E}_{\mathbf{y}^{(:t-1)}} \left[ \mathbb{I}(\xi_{l^{(t)}} < 0) \left( -y_{i_1^{l^{(t)}}}^{(t-1)} + y_{i_1^{l^{(t)}}}^{(t-1)} y_{i_2^{l^{(t)}}}^{(t-1)} + 1 + y_{i_2^{l^{(t)}}}^{(t-1)} \right) \right] \\
&= \frac{1}{2} \mathbb{E}_{\mathbf{y}^{(:t-1)}} \left[ \mathbb{I}(\xi_{l^{(t)}} < 0) y_{i_1^{l^{(t)}}}^{(t-1)} y_{i_2^{l^{(t)}}}^{(t-1)} \right] + \frac{1}{2} \mathbb{E}_{\mathbf{y}^{(:t-1)}} \left[ \mathbb{I}(\xi_{l^{(t)}} < 0) \right].
\end{aligned}
\tag{131}
$$

As a comparison, when $p \notin \{i_1^{l^{(t)}}, i_2^{l^{(t)}}\}$, $\gamma_{l^{(t)}}^p$ is simply Eq. (130). As in the case of $k$-AND (App. D.3.1), we also use the definition of $\mathbf{y}_{\mathcal{A}}^{(t-1)}$ and denote $\bar{D}_\lambda := \{\mathbf{y}_{\mathcal{A}}^{(t-1)} | \sum_j y_{\mathcal{A},j}^{(t-1)} \leq \lambda\}$. Furthermore, we also define four random variables depending on $\mathbf{y}^{(:t-2)}$, i.e.,

$$\bar{a} = \sum_{\mathbf{y}_{\mathcal{A}}^{(t-1)} \in \bar{D}_{-3}} p_+^3 p(\mathbf{y}_{\mathcal{A}}^{(t-1)} | \mathbf{y}^{(:t-2)}), \tag{132}$$

$$\bar{b} = \sum_{\mathbf{y}_{\mathcal{A}}^{(t-1)} \in \bar{D}_{-1}} p_+^2 p_- p(\mathbf{y}_{\mathcal{A}}^{(t-1)} | \mathbf{y}^{(:t-2)}), \tag{133}$$

$$\bar{c} = \sum_{\mathbf{y}_{\mathcal{A}}^{(t-1)} \in \bar{D}_{+1}} p_+ p_-^2 p(\mathbf{y}_{\mathcal{A}}^{(t-1)} | \mathbf{y}^{(:t-2)}), \tag{134}$$

$$\bar{d} = \sum_{\mathbf{y}_{\mathcal{A}}^{(t-1)} \in \bar{D}_{+3}} p_-^3 p(\mathbf{y}_{\mathcal{A}}^{(t-1)} | \mathbf{y}^{(:t-2)}). \tag{135}$$

In this case, the difference of the critical gradient component has the form of

$$\mathbb{E}_{\mathbf{y}^{(:t-1)}}[\bar{\Delta}] \propto \underbrace{\mathbb{E}_{\mathbf{y}^{(:t-1)}}[\mathbb{I}(\xi_{l^{(t)}} < 0)]}_{(i)} - \underbrace{\mathbb{E}_{\mathbf{y}^{(:t-1)}}[\mathbb{I}(\xi_{l^{(t)}} < 0)y]}_{(ii)} \\ - \underbrace{\mathbb{E}_{\mathbf{y}^{(:t-1)}}[\mathbb{I}(\xi_{l^{(t)}} < 0)yy]}_{(iii)} + \underbrace{\mathbb{E}_{\mathbf{y}^{(:t-1)}}[\mathbb{I}(\xi_{l^{(t)}} < 0)yyy]}_{(iv)}. \tag{136}$$

Each term is computed as follows (similar to App. D.3.1):

$$(i) = \sum_{\mathbf{y}^{(:t-2)}} p(\mathbf{y}^{(:t-2)}) \Big[ \sum_{\mathbf{y}_{\mathcal{A}}^{(t-1)} \in \bar{D}_{-3}} p_+^3 p(\mathbf{y}_{\not{\mathcal{A}}}^{(t-1)} | \mathbf{y}^{(:t-2)}) + 3 \sum_{\mathbf{y}_{\mathcal{A}}^{(t-1)} \in \bar{D}_{-1}} p_+^2 p_- p(\mathbf{y}_{\mathcal{A}}^{(t-1)} | \mathbf{y}^{(:t-2)}) \\ + 3 \sum_{\mathbf{y}_{\mathcal{A}}^{(t-1)} \in \bar{D}_{+1}} p_+ p_-^2 p(\mathbf{y}_{\mathcal{A}}^{(t-1)} | \mathbf{y}^{(:t-2)}) + \sum_{\mathbf{y}_{\mathcal{A}}^{(t-1)} \in \bar{D}_{+3}} p_-^3 p(\mathbf{y}_{\mathcal{A}}^{(t-1)} | \mathbf{y}^{(:t-2)}) \Big] \\ = \mathbb{E}_{\mathbf{y}^{(:t-2)}}[\bar{a} + 3\bar{b} + 3\bar{c} + \bar{d}]; \tag{137}$$

$$(ii) = \sum_{\mathbf{y}^{(:t-2)}} p(\mathbf{y}^{(:t-2)}) \Big[ \sum_{\mathbf{y}_{\mathcal{A}}^{(t-1)} \in \bar{D}_{-3}} p_+^3 p(\mathbf{y}_{\not{\mathcal{A}}}^{(t-1)} | \mathbf{y}^{(:t-2)}) + \sum_{\mathbf{y}_{\mathcal{A}}^{(t-1)} \in \bar{D}_{-1}} p_+^2 p_- p(\mathbf{y}_{\mathcal{A}}^{(t-1)} | \mathbf{y}^{(:t-2)}) \\ - \sum_{\mathbf{y}_{\mathcal{A}}^{(t-1)} \in \bar{D}_{+1}} p_+ p_-^2 p(\mathbf{y}_{\mathcal{A}}^{(t-1)} | \mathbf{y}^{(:t-2)}) - \sum_{\mathbf{y}_{\mathcal{A}}^{(t-1)} \in \bar{D}_{+3}} p_-^3 p(\mathbf{y}_{\mathcal{A}}^{(t-1)} | \mathbf{y}^{(:t-2)}) \Big] \\ = \mathbb{E}_{\mathbf{y}^{(:t-2)}}[\bar{a} + \bar{b} - \bar{c} - \bar{d}]; \tag{138}$$

$$(iii) = \sum_{\mathbf{y}^{(:t-2)}} p(\mathbf{y}^{(:t-2)}) \Big[ \sum_{\mathbf{y}_{\mathcal{A}}^{(t-1)} \in \bar{D}_{-3}} p_+^3 p(\mathbf{y}_{\not{\mathcal{A}}}^{(t-1)} | \mathbf{y}^{(:t-2)}) \\ - \sum_{\mathbf{y}_{\mathcal{A}}^{(t-1)} \in \bar{D}_{-1}} p_+^2 p_- p(\mathbf{y}_{\mathcal{A}}^{(t-1)} | \mathbf{y}^{(:t-2)}) - \sum_{\mathbf{y}_{\mathcal{A}}^{(t-1)} \in \bar{D}_{+1}} p_+ p_-^2 p(\mathbf{y}_{\mathcal{A}}^{(t-1)} | \mathbf{y}^{(:t-2)}) \\ + \sum_{\mathbf{y}_{\mathcal{A}}^{(t-1)} \in \bar{D}_{+3}} p_-^3 p(\mathbf{y}_{\mathcal{A}}^{(t-1)} | \mathbf{y}^{(:t-2)}) \Big] \\ = \mathbb{E}_{\mathbf{y}^{(:t-2)}}[\bar{a} - \bar{b} - \bar{c} + \bar{d}]; \tag{139}$$

and

$$(iv) = \sum_{\mathbf{y}^{(:t-2)}} p(\mathbf{y}^{(:t-2)}) \Big[ \sum_{\mathbf{y}_{\mathcal{A}}^{(t-1)} \in \bar{D}_{-3}} p_+^3 p(\mathbf{y}_{\mathcal{A}}^{(t-1)}|\mathbf{y}^{(:t-2)})$$

$$- 3 \sum_{\mathbf{y}_{\mathcal{A}}^{(t-1)} \in \bar{D}_{-1}} p_+^2 p_- p(\mathbf{y}_{\mathcal{A}}^{(t-1)}|\mathbf{y}^{(:t-2)}) + 3 \sum_{\mathbf{y}_{\mathcal{A}}^{(t-1)} \in \bar{D}_{+1}} p_+ p_-^2 p(\mathbf{y}_{\mathcal{A}}^{(t-1)}|\mathbf{y}^{(:t-2)}) \tag{140}$$

$$- \sum_{\mathbf{y}_{\mathcal{A}}^{(t-1)} \in \bar{D}_{+3}} p_-^3 p(\mathbf{y}_{\mathcal{A}}^{(t-1)}|\mathbf{y}^{(:t-2)}) \Big]$$

$$= \mathbb{E}_{\mathbf{y}^{(:t-2)}}[\bar{a} - 3\bar{b} + 3\bar{c} - \bar{d}].$$

Combining these expressions gives us the final result

$$\bar{\Delta} \propto 8c > 0; \tag{141}$$

hence the separation of the critical gradient in Thm. 3.1 is satisfied such that $k$-OR can also be learned perfectly via RL.

**SFT.** As discussed earlier in App. D.3.1, compared to the case for fine-tuning via RL, the only difference brought by evaluating the critical gradient component for SFT is that we are now computing

$$\mathbb{E}_{\mathbf{x}}[\gamma_{l^{(t)}}^p(\tilde{\mathbf{y}}^{(t-1)})], \tag{142}$$

where $\tilde{\mathbf{y}}^{(t-1)}$ is the ground-truth of $\mathbf{y}^{(t-1)}$, which does not change the conclusion that $\bar{\Delta} \propto 8c > 0$; thus the separation of the critical gradient component in Thm. 3.2 is also satisfied, which further guarantees the learnability via SFT.

# E NUMERICAL EXPERIMENTS

We conduct straightforward numerical experiments to empirically verify our theoretical claims. We specifically consider $k$-PARITY. To learn it, we use a one-layer transformer $f(\cdot; \boldsymbol{W})$ (with $\boldsymbol{W} = \mathbf{1}$ at initialization) as specified in Sec. 2.2. The input data $\mathbf{x} \in \{-1, +1\}^d$ and we construct a random subset $B \subseteq [d]$ with $|B| = k$ for the $k$-PARITY. We let $d = 20$ and $k = 16$; hence the CoT chain will have 4 steps, namely $\mathbf{y} = (\mathbf{y}^{(1)}, \mathbf{y}^{(2)}, \mathbf{y}^{(3)}, \mathbf{y}^{(4)})$. We uniformly sample $50,000$ samples of $\mathbf{x}$ to be our training dataset, where we also build the corresponding ground-truth reasoning sequence $\tilde{\mathbf{y}}$ for SFT as discussed in Sec. 3.2. In Fig. 3a, we plot the ground-truth attention score that can build $\tilde{\mathbf{y}}$, where each white box in a column $y_j^{(t)}$ denotes one child node of $y_j^{(t)}$ such that the attention score has value 0.5 at each white box and 0 at gray boxes. We omit the attention scores for $W_{i,j}$ with $j \leq d$ as we will not generate $\mathbf{x} \in \{-1, +1\}^d$.

**Fine-tuning via RL.** We fine-tune $f(\cdot; \boldsymbol{W})$ via RL optimized by the sign of the policy gradient following Sec. 3.1. We plot $\text{sign}(\nabla_{\boldsymbol{W}} \mathcal{R}(\boldsymbol{W}))$ at $\boldsymbol{W}(0)$ in Fig. 3b. Compared to Fig. 3a, we can clearly see that $\nabla_{\boldsymbol{W}} \mathcal{R}(\boldsymbol{W})$ has positive values in all relevant positions and negative values in irrelevant positions, i.e., a distinct separation that enables the transformer to learn $k$-PARITY after one update.

**SFT.** In contrast to the one-update learning of RL, SFT exhibits a stepwise learning behavior, as shown in Fig. 4. We use $s$ to count the iteration number. In particular, the gradient $\nabla_{\boldsymbol{W}} \mathcal{L}(\boldsymbol{W})$ only has nonzero values for the first reasoning step $\mathbf{y}^{(1)}$ at $s = 0$, where it has positive values at relevant positions of each token of $\mathbf{y}^{(1)}$ and negative values at irrelevant positions. Hence SFT can and only can learn the first CoT step at the first update of sign gradient descent. Similarly, the transformer can and only can learn $\mathbf{y}^{(2)}$ at $s = 1$, etc.

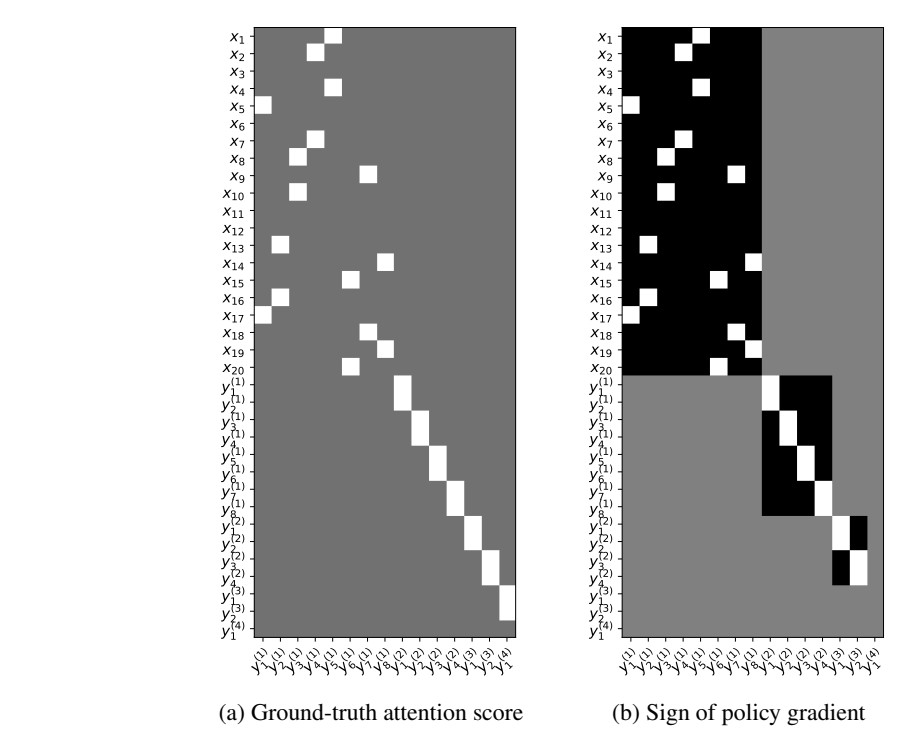

(a) Ground-truth attention score    (b) Sign of policy gradient

Figure 3: **(a)** The ground truth $(\sigma^\star)_{N_{t-1}+l^{(t)}}^{N_{t-2}+p}$. Each white box is $0.5$ and each gray box is $0$. **(b)** $\text{sign}(\nabla_{\boldsymbol{W}} \mathcal{L}(\boldsymbol{W}))$ at $\boldsymbol{W}(0) = \boldsymbol{1}$. Each white box has value $+1$ and each black box has value $-1$. Gray boxes have value $0$ coming from causal mask and pretrained mask.

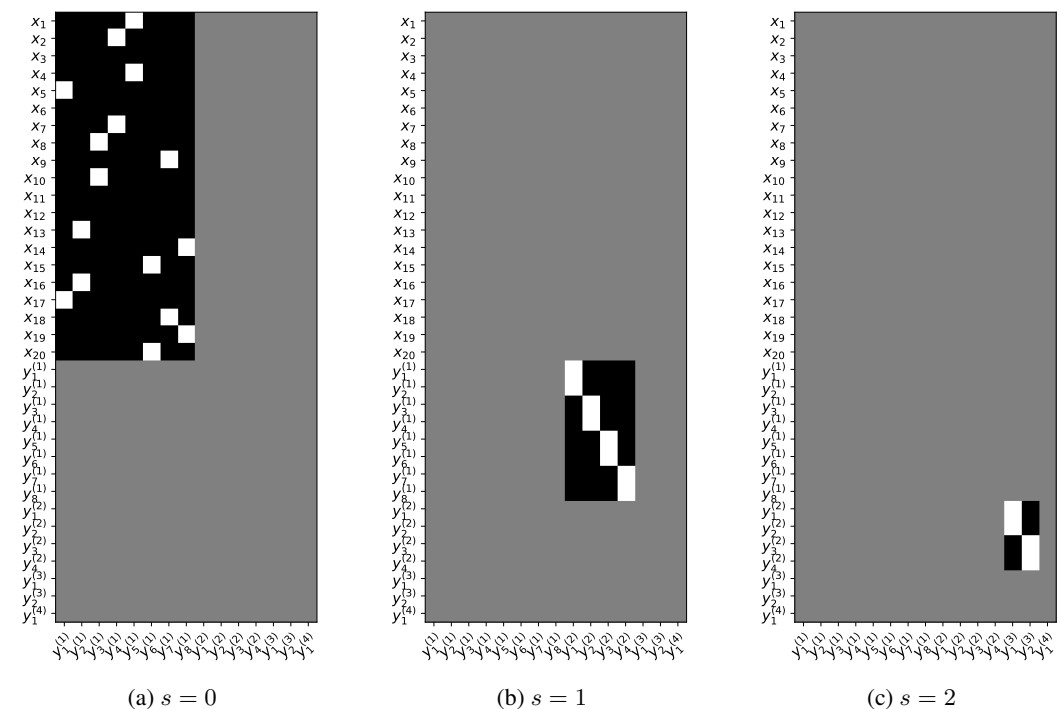

(a) $s = 0$    (b) $s = 1$    (c) $s = 2$

Figure 4: $\text{sign}(-\nabla_{\boldsymbol{W}} \mathcal{L}(\boldsymbol{W}))$ computed by $\boldsymbol{W}(s)$ for different updating step $s$. Each white box has value $+1$, each black box has value $-1$, and grey boxes are $0$.

