# OpenReview forum: "Transformers with RL or SFT  Provably Learn Sparse Boolean Functions, But Differently"
_ICLR.cc/2026/Conference — Submitted to ICLR 2026_

### Official Review · Reviewer_GHnk · 2025-10-20

**Soundness:** 4
**Presentation:** 3
**Contribution:** 2
**Rating:** 2
**Confidence:** 4

**Summary:**

The paper studies the learnability of $k$-sparse boolean functions (parity, AND, OR) using a one-layer transformer model by generating intermediate steps in CoT fashion. For fine-tuning via RL and policy gradient, given access to immediate rewards, it is shown that transformers can learn certain functions in one gradient step. For SFT, the model can also learn these functions but in a stepwise manner, requiring compute equal to chain length.

**Strengths:**

* The setup is very similar to the CoT-based parity learning formulation introduced in [KS25] but makes some technical changes which allow for improved analysis. The intermediate tokens are generated via sampling as $p(y=1|z)=model(z)$ rather than simply setting $y=model(z)$, which better captures the autoregressive nature of CoT.
* This also makes error analysis easier. Compared to [WLS23,KS25], the paper does not require teacher forcing nor self-verification/filtering to control error amplification for the SFT learning result.
* The task can be more general than parity. A general condition is identified (critical gradient component) which implies learnability.

[WLS23] Noam Wies, Yoav Levine, and Amnon Shashua. Sub-task decomposition enables learning in sequence to sequence tasks, 2023.

[KS25] Juno Kim and Taiji Suzuki. Transformers provably solve parity efficiently with chain of thought. ICLR, 2025.

**Weaknesses:**

The main weakness is novelty, the setup and results are quite similar to [KS25]. While comparisons are given in the paper, I feel the differences are mostly technical and not significant from either an SQ learning perspective or helping to understanding post-training.
* RL with immediate rewards is very similar to teacher forcing in that each step decouples into its own problem, and so one GD step solves all steps simultaneously. While RL in this paper does not require intermediate states during generation as in [KS25], this is because exact process reward is available for the computation of each step regardless of the data from the previous step. This is an extremely strong assumption which sidesteps all the aspects of RL which make it challenging/interesting (e.g. how to obtain a process reward model, how to deal with imperfect rewards, how to do exploration, how to do credit assignment).
* SFT learning is very similar to the no teacher forcing result in [KS25]. While the paper does a cleaner setup/analysis and does not require the augmented data trick in [KS25] (which forces curriculum learning), the overall message is the same: step-by-step learning arises from training on CoT loss. Also, a similar result was proven for k-hop composition tasks by [WNBD+25] without teacher forcing or data augmentation. Hence the result does not seem to give particularly new insights into SFT.
* Besides the dynamical analysis, the results do not help us understand the differences of models fine-tuned via RL v.s. SFT since the converged solutions are the same. For example, there is no finite-sample or generalization analysis. Moreover, "pretraining" here simply refers to adding the step-wise causal mask and does not seem to have connection to real-world pretraining.
* Wording throughout the paper makes it seem as the results apply to general $k$-sparse boolean functions $\Phi_k$, but the results only apply to functions which can be seen as repeated composition of a fixed 2-sparse function $\phi_2$ (this is a much smaller class due to the standard circuit counting argument). This restriction is not made explicit in the abstract, introduction, and even the problem setup, which is misleading; $\phi_2$ is first mentioned in eq.(1) long after the definition of $k$-sparse function is given, and even the wording there does not make it clear that $\phi_2$ must be fixed across nodes (since the paragraph is talking about how to decompose general $k$-sparse functions into binary trees). This must be made clear in the introduction and Definition 2.1.

[WNBD+25] Learning Compositional Functions with Transformers from Easy-to-Hard Data. Zixuan Wang, Eshaan Nichani, Alberto Bietti, Alex Damian, Daniel Hsu, Jason D Lee, Denny Wu. COLT 2025.

**Questions:**

* Can the result be generalized to more general $k$-sparse functions?
* Can the setup be modified to obtain length or sparsity generalization results for the finetuned models?

---

> ### Author Response · Authors · 2025-11-19
> **Our Response, Part I/II**
>
> Thank you for your comment.
>
> However, we respectfully disagree with the reviewer's assertion that our work lacks novelty or that our contributions are *"mostly technical and not significant"*. This underestimates the conceptual and theoretical advances made in this paper.
>
> We highlight that our work establishes the first framework for theoretically understanding RL fine-tuning for transformers, which are largely absent in the current literature. This is clearly not only *"very similar to ... introduced in [KS25]"*. In addition, we provide a novel characterization of the distinct learning behaviors between RL and SFT without teacher forcing as well as data augmentation (or curriculum learning). In contrast to prior works, the step-wise learning of SFT naturally arises from transformer: it is an intrinsic property not relying on external tricks. This is a novel mechanical insight.
>
> Therefore, we disagree with the reviewer on that these aspects are only *"some technical changes"*. Instead, we believe our results provide **a novel and significant extension and foundation** for future works to further explore the learning dynamics of RL fine-tuning and its comparison with SFT (without teacher forcing or other techniques).
>
> In addition, regarding the first point provided in the Weaknesses part, we respectfully disagree with the reviewer's argument that *"RL with immediate rewards is very similar to teacher forcing in that each step decouples into its own problem"*. Instead, the similarity of providing step-wise guidance is only superficial. Their mechanisms and objectives are fundamentally distinct. In particular:
>
>   - For SFT with teacher forcing, the model is always fed the correct previous steps and trained to correctly predict the next step. Teacher forcing provides a perfect input for each step, ensuring the learning target for that step is well-defined. And the model learns by matching patterns.
>
>   - For RL with immediate reward, as a comparison, the model generates complete reasoning chains through its current policy, receives immediate rewards for each step, and updates to maximize expected reward. The model learns from its own generated trajectories by exploring different actions, even when its prior steps are imperfect.
>
> Therefore, the formal theoretical equivalence between the learnability of RL with immediate reward and that of SFT with teacher forcing is a novel theoretical insight---two distinct mechanisms lead to similar efficient learning results---rather than a merely superficial comparison that underestimates our contribution.
>
> Below we present our response to the rest of Weaknesses.
>
> ---
> ## Response to Weaknesses Part
>
> 1. **Comment:** *"While RL in this paper does not require ... This is an extremely strong assumption which sidesteps all the aspects of RL which make it challenging/interesting"*
>
>     **Response:** We respectfully disagree with the reviewer's critique.
>
>     In this paper, we focus on the challenge of learning. The interesting aspects we study are **the learnability and the corresponding learning dynamics analysis** for RL through the transformer's architecture, considering its distinct mechanism. The assumption of a perfect step-wise reward is necessary for a provable baseline, allowing us to establish highly non-trivial formal guarantees about learnability and reveal fundamental differences in learning dynamics. This theoretical finding is independent of the reward source.
>
>     In addition, we emphasize that the mentioned challenges in the comment are still highly challenging open problems. By providing the first theoretical guarantees for RL with CoT, we provide possible foundation upon which solutions to these challenges can be systematically built.

---

> ### Author Response · Authors · 2025-11-19
> **Our Response Part II/II**
>
> ## Continuing Response to Weaknesses Part
>
> 2. **Comment:** *"SFT learning is very similar to the no teacher forcing result in [KS25]. ... Hence the result does not seem to give particularly new insights into SFT."*
>
>     **Response:** We respectfully disagree with the reviewer's assertion that *"SFT learning is very similar to ..."*, which underestimates our message.
>
>     As acknowledged by the reviewer, we do not require augmented data as well as the corresponding filter of SFT without teacher forcing in Kim \& Suzuki, (2025). This is a special curriculum learning. As a comparison, our approach addresses the mismatch between training and inference, where the training does not rely on any forms of curriculum learning or data augmentation tricks. We show that the stepwise learning emerges naturally from autoregressive training.
>
>    The **overall message is more**: the stepwise learning can be an intrinsic property of how transformers acquire reasoning capabilities, without relying on external tricks.
>
>     Finally, thank you for suggesting Wang et al., (2025). The learning results in Wang et al., (2025) still relies on easy-to-hard strategies (Theorem 3-4 there). As a comparison, our setting does not rely on such strategies (e.g., curriculum learning, data mixture), and we show that stepwise learning can emerge naturally in SFT without external tricks, as discussed above. We also include a discussion of this in the revision (Appendix A, page 14).
>
> 3. **Comment:** *"Besides the dynamical analysis, the results do not help us understand the differences of models fine-tuned via RL v.s. SFT since the converged solutions are the same. For example, there is no finite-sample or generalization analysis. Moreover, "pretraining" here simply refers to adding the step-wise causal mask and does not seem to have connection to real-world pretraining."*
>
>     **Response:** We would like to firstly clarify that our primary focus is the comparative analysis between RL and SFT learning dynamics on the same tasks. We provide our response to this comment below point by point.
>
>     - *"the converged solutions are the same"*: While it is true, the learning dynamics, i.e., how they arrive at the solution, are fundamentally different. Our analysis highlights a qualitative difference in learning behavior that is not apparent from the solution alone.
>
>     - *"there is no finite-sample or generalization analysis"*: Finite-sample and generalization analysis would definitely be valuable extensions, as we have acknowledged in Sec. 5 (Limitations and future directions). However, our work is a first-step theoretical analysis, where studying population-level gradients is a common and necessary starting point.
>
>       This helps isolate the fundamental differences between RL and SFT without the confounding effects of finite-sample noise. The results can be extended to finite-sample analysis by upper bounding the sample noise by concentration, which should be an interesting future direction.
>
>     - *"Pretraining here is just masking, not real-world pretraining"*: We first clarify that our purpose is not to replicate all the real-world systems but to rigorously prove for the first time the distinct learning behaviors of RL and SFT in in transformers with CoT. Our paper has explicitly stated that the "pretrained mask" is distinct from the conventional pretraining.
>
>        This is not an oversight. Specifically, our "pretraining" mask encodes a very simple prior inductive bias. In real-world models, pretraining also provides models with certain structural priors. While our setup is abstract, it captures the idea that pretraining induces certain priors. In addition, we believe our framework can be interestingly applied as case study to explore how different pretraining priors can affect RL and SFT tuning dynamics.
> 4. **Comment:** *"Wording throughout ... in the introduction and Definition 2.1."*
>
>     **Response:** Thank you for this suggestion. We implicitly use $\phi_2$ to indicate that during the decomposition the 2-sparse Boolean function is fixed. To make further improvement, in the revision (abstract, introduction, setup(Page 3)), we explicitly state that (1). $\phi_2$ is fixed; (2). the results only apply to functions that can be expressed as repeated compositions of a fixed $\phi_2$.
>
> ---
> ## Response to Questions Part
> 1. **Comment:** *"Can the result be generalized to more general $k$-sparse functions?"*
>
>     **Response:**If the $k$-sparse Boolean functions admit a similar binary tree decomposition, then the results can be generalized by examining the separation of critical gradient component to validate the learnability.
> 2. **Comment:** *"Can the setup be modified to obtain ... results for the finetuned models?"*
>
>     **Response:** For the length generalization setup, the adaptive positional encoding is necessary. But currently we use fixed positional encoding. And we believe generalizing to such setup would be interesting.

---

### Official Review · Reviewer_zXpU · 2025-10-29

**Soundness:** 2
**Presentation:** 2
**Contribution:** 2
**Rating:** 2
**Confidence:** 2

**Summary:**

The paper is concerned with learnability of k-sparse Boolean functions by transformers with chain of thought.

A k-sparse Boolean function is a function that depends on k input bits. For example, a k-sparse parity on d bits is a function that is equal to the parity of inputs from some k-element subset of coordinates B.

The paper considers transformers with the following chain of thought. The first step produces k tokens. The second step produces k/2 tokens. The third produces k/4, and so on. Each token is supposed to attend exactly two tokens from the previous step (and, in case of parity, to compute their 2-wise parity). After log k steps, the final token is supposed to give the output of the function.

The paper shows results about learnability of this architecture via reinforcement learning and supervised
fine-tuning approaches.

**Strengths:**

The paper proposes potential approaches to circumvent limitations of learnability of k-sparse parities, established in

**Weaknesses:**

My major concern with the paper is that the set of k-coordinates, B, on which the function depends, seems to be known to the architecture. At least at the line 152, the lowest k tokens of the chain of thought are directly attending the corresponding indices of the set B. In this case, the role of the other coordinates is not clear (so that we are left with simply parity instead of k-sparse parity).

Now, in the interesting setting when B is not known in advance, I don't see how the proposed RL reward, for example, takes into account which coordinates belong to B and which are not. More generally, it is not clear how is the optimal solution W^* in Theorem 3.1, for example, for the initial problem of learning the parity of an unknown set of inputs.

**Questions:**

Is the set of k coordinates, where, say, the parity is computed, is known to the architecture?  If not, then how is it reflected, how well the architecture learns B? In particular, how is r_t  defined in equation 6 for t = 1? Is it possible to obtain some bounds on how well are optimal solutions W^* in Theorems 3.1 an 3.2 for the initial problem of computing a k-sparse parity?

---

> ### Author Response · Authors · 2025-11-19
> **Our Response**
>
> Thank you for your comment.
>
> We feel that there are misunderstandings regarding our core contributions in the comment. We appreciate this opportunity to provide a clarification below.
>
>    1. The assessment of strengths consists of only a single sentence which is incomplete:
>
>         > *"The paper proposes potential approaches to circumvent limitations of learnability of k-sparse parities, established in"*
>
>       In addition, this misunderstands our work. We do not merely *"proposes potential approaches"*. We conduct definitive and novel theoretical analysis on comparing how RL and SFT without teacher forcing as well as data augmentation exhibit different learning dynamics for transformers.
>
>       And we theoretically characterize the distinct learning behaviors of RL and SFT, the simultaneous learning versus step-wise learning. This sheds lights on the mechanism of RL and SFT for learning CoT. Thus, describing *"proposes potential approaches"* is a understatement that misrepresents our core contributions.
>
>    2. The major concern in the Weaknesses part
>       >*"the set of k-coordinates, B, on which the function depends, seems to be known to the architecture"*
>
>       is also a misunderstanding of our framework (as addressed later). We clarify that showing that the transformer can find these $k$-coordinates through RL or SFT is one of the main conclusion of our results. This clearly gives a negative answer to the major concern.
>
> We now provide a detailed response to address the misunderstandings.
>
> ---
> ## Response to Weaknesses Part
>
> 1. **Comment:** *"... the set of k-coordinates, B, on which the function depends, seems to be known to the architecture."*
>
>    **Response:** The model does not know any coordinates in $B$. Instead, the model learns these $k$ coordinates in $B$ from data, which is one of our main theoretical conclusions.
>
>     In particular, given a $k$-sparse Boolean function $\Phi_k(\cdot)$ with a random set $B$ (this set no longer changes subsequently, otherwise learning is impossible), $\Phi_k(\cdot)$ can be solved by following the intermediate steps in Fig.1a. Then we can collect some data $x$ with the corresponding ground-truth labels (and intermediate steps as CoT for SFT) by using this $\Phi_k(\cdot)$.
>
>     Now the goal of learning is to train the transformer $f(\cdot; W)$ to correctly predict $\Phi_k(\cdot)$ with such data. The point is that coordinates of $B$ are not revealed to the model directly---such information is encoded in the data. This is where the learnability comes in: it measures whether or not the model can learn to correctly predict $\Phi_k(\cdot)$ with the given data. This certainly includes learning which coordinates are in $B$.
>
> 2. **Comment:** *"At least at the line 152, the lowest k tokens of the chain of thought are directly attending the corresponding indices of the set B."*
>
>    **Response:** As discussed in the previous point, line 152 is for how $\Phi_k(\cdot)$ is recursively solved. It is not a discussion on the model or the learning process. And the model does not know which coordinates are in $B$, i.e., the model parameter $W$ is not specially adapted to $B$.
>
> 3. **Comment:** *"I don't see how the proposed RL reward, for example, takes into account which coordinates belong to B and which are not."*
>
>    **Response:** We clarify that the reward is not a part of the model. It belongs to the "environment", and it has to know the coordinates in $B$ to determine which action is a good one based on the current state.
>
> 4. **Comment:** *"More generally, it is not clear how is the optimal solution $W^\star$ in Theorem 3.1, for example, for the initial problem of learning the parity of an unknown set of inputs."*
>
>    **Response:** Once $\Phi_k(\cdot)$ and its recursive decomposition are determined, $W^\star $ is uniquely determined (line 272-278, page 6) by requiring the transformer to only attend to the relevant positions (child nodes of a token in Fig.1a) when generating CoT tokens. For the formulation, please refer to Eq.(44) (Page 20 of the revision). Please also see Fig.3a  (Page 36 of the revision) for $W^\star$ obtained in the numerical experiment.
>
> ---
> ## Response to Questions Part
>
> 1. **Comment:** *"Is the set of k coordinates, where, say, the parity is computed, is known to the architecture?"*
>
>    **Response:** No. Please see our response to the first point of the Weaknesses.
>
> 2. **Comment:** *"If not, then how is it reflected, how well the architecture learns B?... defined in equation 6 for t = 1?"*
>
>    **Response:** As discussed above, the reward is a part of the environment, not model. The optimal $W^\star $ has the highest reward, and how well the architecture learns B depends on how close its parameter is to $W^{\star}$ (the $\ell_1$-error is certainly such a measure).
>
> 3. **Comment:** *"Is it possible to obtain some bounds ... a k-sparse parity?"*
>
>    **Response:** Yes. Please see our response to the fourth point of the Weaknesses.

---

### Official Review · Reviewer_2mWM · 2025-11-02

**Soundness:** 2
**Presentation:** 2
**Contribution:** 2
**Rating:** 4
**Confidence:** 2

**Summary:**

The paper analyzes how a one‑layer transformer acquires chain‑of‑thought (CoT) style reasoning when fine‑tuned either with reinforcement learning (RL; sign policy‑gradient) or supervised fine‑tuning (SFT; hinge loss, no teacher forcing). It derives sufficient “gradient separation” conditions under which both procedures provably learn k-sparse Boolean functions, and verifies them for parity, AND, and OR. A key takeaway is a qualitative difference in learning dynamics: RL can learn the whole CoT chain at once, whereas SFT learns it step‑by‑step, needing one update per reasoning step. Closed‑form attention dynamics are given for parity (Eq. 15). Experiments in the appendix support the theory.

**Strengths:**

The paper cleanly formalizes why RL can propagate credit to all CoT steps while SFT induces an inductive, stagewise trajectory, yielding an intuitive and provable contrast between the two regimes.

**Weaknesses:**

1. Limited realism of the learning setting. Results hold for a one‑layer transformer with designed activations and structured masking aligned to the recursive CoT decomposition; this constrains architectural/general modeling fidelity to modern LLMs.
2. Population‑gradient analysis and simplified optimizers. The theory is for population (infinite‑data) gradients and sign policy‑gradient/SGD dynamics; practicality for finite‑sample training and commonly used RL (e.g., PPO with KL control) is not addressed.
3. Practical significance is unclear. While the paper sheds light on mechanisms in toy settings, the path to improved real‑world reasoning is not demonstrated.

**Questions:**

See weaknesses

---

> ### Author Response · Authors · 2025-11-19
> **Our Response**
>
> We thank the reviewer for the comment.
>
> First of all, we would like to clarify that **we do not aim to directly contribute to the general large scale LLMs**, which is clearly a very challenging and open problem. Our goal is not to replicate the full complexity of a modern LLM. We emphasize that the analysis of modern LLMs, PPO with KL control, and real-world reasoning are still open problems. Instead, we focus on theoretically understanding the underlying learning dynamics of RL and how it is different from SFT under a tractable setting that can be rigorously analyzed.
>
> Thus, we believe the weaknesses in the comment are somewhat out of the scope of our paper and misunderstands our goals.
>
> We acknowledge the limitations of our results, as we have summarized in Section 5 (page 9 of our paper). However, we would like to highlight that the significance and novelty of our contribution should not be overlooked. In particular, while fine-tuning via RL for transformers has demonstrated remarkable empirical success in enhancing LLM reasoning, its theoretical understandings were largely absent.
>
> Our work takes the first essential step towards building its theoretical foundations, and further proposes the timely theoretical comparison between RL and SFT in learning CoT. Our result firstly sheds lights on their distinct learning behaviors: simultaneous learning versus stepwise learning that naturally arises from transformer. This is a formal, mechanistic insight.
>
> The setting is simplified, yet it is nontrivial: we build on the established theoretical work Kim \& Suzuki, (2025), and extend to RL and SFT without teacher-forcing as well as augmented data. We believe this will provide a solid reference for future works to further explore the learning dynamics of RL (e.g., variants of policy gradient) and SFT as well as their difference.

---

### Official Review · Reviewer_kGV3 · 2025-11-04

**Soundness:** 3
**Presentation:** 2
**Contribution:** 2
**Rating:** 4
**Confidence:** 3

**Summary:**

This paper studies the task of learning $k$ sparse Boolean functions given intermediate chain of thought data with both RL and SFT (without teacher forcing). They study a simplified 1-layer masked transformer and show that RL succeeds in a single step of sign-GD under a separation condition on a quantity they call the critical gradient component. They show that under the same condition, SFT without teacher forcing succeeds in $T = \log_2(k)$ steps.

**Strengths:**

- End to end analysis of sign-GD in both policy gradient and an SFT setting without teacher forcing. In addition, the SFT analysis is able to analyze multiple steps of sign-GD.
- Without making any direct assumptions on the distribution of the data, the paper identifies a criterion in terms of the critical gradient component which determines whether one gradient step is sufficient for learnability.
- The results are written with some generality and easily extend to k-parity, k-AND, and k-OR.

**Weaknesses:**

- The RL analysis is restricted to taking one very large step of sign-GD to immediately saturate the softmax, which significantly simplifies the analysis.
- Throughout the paper, the authors use SFT to refer to SFT without teacher forcing which changes the main conclusion. As the authors acknowledge, GD succeeds in one step in the more standard setting of SFT with teacher forcing.
- It may be more accurate to refer to the "pre-trained" transformer as a simplified model intended to ease the analysis. In particular, the hard-coded causal masks seem like a reasonable theoretical starting point but it's not clear to me how they would arise from pretraining. In addition, the activation function is required to depend on the target function, which limits the scope of the analysis.
- The final guarantees are on the L1 error of the attention patterns, which doesn't guarantee low loss.
- The paper does not analyze the sample complexity of learning $k$-parity. For example, even under random inputs and without chain of thought supervision, parity is learnable in a single gradient descent step using $d^k$ samples [Barak et al. 2022 Hidden Progress in Deep Learning: SGD Learns Parities Near the Computational Limit]. To argue for the benefit of chain of thought, you would need to prove that parity can be learned with dimension independent sample complexity.
- There is no lower bound that demonstrates that $T$ steps of sign-GD are actually necessary, only that $T$ steps are sufficient which doesn't immediately imply a separation (although from the argument it seems clear that $T$ steps really are sufficient due to the lack of teacher forcing).

**Questions:**

- Why is it necessary to go from equation 7 to 8 for the policy gradient? Is this a modeling decision (e.g. equation 8 is easier to analyze) or are these mathematically equivalent in some sense? The paper only mentions that the $r_t$ term is "considerably more important."
- Is there any intuition for the critical gradient component? Equation 9 is particularly hard to read because $\xi$ depends on $\sigma$ which depends on the transformer's weights $W$. Since this paper focuses on the initial step of gradient descent, it may be informative to write out a simpler expression for the critical gradient component for parity/k-AND/k-OR when $W$ is $0$ so that $\sigma$ is constant.
- In Proposition 3.1, why does this variance bound imply a lower bound for learnability via gradient descent?
- Why is the hinge loss important in section 3.2? If logistic loss were used instead, I think that the initial gradient step would be the same. Is the issue controlling the following $T-1$ gradient steps?

---

> ### Author Response · Authors · 2025-11-19
> **Our Response Part I/II**
>
> We thank the reviewer for the comment.
>
> First of all, we would like to highlight the significance and novelty of our contributions, which should not be diminished by limitations or assumptions brought by theoretical tractability that are often necessary for theoretical works.
>
> In particular, we provide the first theoretical analysis that compares how transformers learn CoT under RL and SFT without teacher forcing or data augmentation in a controllable setup. This is a general framework that identifies a unifying critical gradient component for investigating the corresponding learnability of RL and SFT. And we establish a fundamental distinction between their inherent learning behaviors (simultaneous learning versus step-by-step). We believe this will provide a solid foundational framework for future works to further explore the  learning dynamics of RL and its comparison against SFT.
>
> ----
> ## Response to Weaknesses Part
>
> 1. **Comment:** *"The RL analysis is restricted ... significantly simplifies the analysis."*
>
>     **Response:** As in Kim \& Suzuki, (2025) which studied one-step GD learning, we analyze one step update in Theorem 3.1 because it is sufficient for the model to learn, as long as the separation condition is satisfied. Furthermore, as this theorem is a general one---it does not restrict the formulations of either activation function $\psi$ or 2-sparse Boolean function $\phi_2$, we believe that it already provides sufficiently meaningful results in a clean manner.
>
>     In addition, we highlight that multi-step analysis is possible for specific problems, which, albeit complicated, is fundamentally similar to one-step analysis in its core principle. In fact, Theorem 4.1 establishes learning dynamics of RL with arbitrary steps of sign policy gradient for $k$-PARITY. This removes the restriction of one-step sign-GD and still has provable learnability.
>
> 2. **Comment:** *"Throughout the paper, ... teacher forcing."*
>
>     **Response:** We argue that SFT without teacher forcing is more theoretically natural. Because transformers generate tokens autoregressively at inference. Remove teacher forcing in training can address the mismatch.
>
>     We have modified the manuscript to clearly state that we refer to SFT without teacher forcing.
>
> 3. **Comment:** *"It may be more accurate to refer to the "pre-trained" ... the scope of the analysis."*
>
>     **Response:** The "pretrained" transformer is essentially established to guarantee that each action only depends on the most recent state for RL, which is a prior. We acknowledge that it is distinct from the standard pretraining as we have noted in the paper, hence the quotation marks on it.
>
>     However, we emphasize that our goal is not to explain pretraining, and the "pretrained mask" is a theoretical abstraction that captures the idea of providing a prior by pretraining. Our work is an important start point. Future work can relax this assumption and explore interesting questions such as how this mask can be obtained via pretraining and how different pretraining can affect the later fine-tuning via RL or SFT in our framework.
>
>     In the most general case Theorem 3.1 and Theorem 3.2, we do not require the specific form of the activation function. It is only required when solving a certain Boolean function, as it must guarantee the expressibility of the transformer to correctly predict a given Boolean function.
>
> 4. **Comment:** *"The final guarantees are on ... guarantee low loss."*
>
>     **Response:** There is only one exact optimal softmax$(W)$ given the recursive decomposition, which achieves the highest accuracy. Hence we believe low $\ell_1$-error is a sufficient measure for the learnability.
>
> 5. **Comment:** *"The paper does not analyze the sample ... independent sample complexity."*
>
>     **Response:** The hardness of learning without chain of thought has been proved in Kim \& Suzuki, (2025) (Theorem 2, with finite sample). Our focus in this paper is the comparison between learning dynamics of RL and that of SF.
>
> 6.  **Comment:** *"There is no lower bound that demonstrates ... to the lack of teacher forcing)."*
>
>     **Response:** As we show in the proof of Theorem 3.2 (Appendix C.1, page 23), each sign gradient update can and only can learn one step of the CoT chain. Hence, it at least needs $T$ steps of sign gradient in total to learn the whole chain, which is also the lower bound.

---

> ### Author Response · Authors · 2025-11-19
> **Our Response Part II/II**
>
> ## Response to Questions Part
>
> 1. **Comment:** *"Why is it necessary to go from equation 7 ... term is "considerably more important.""*
>
>     **Response:** On one hand, equation 8 is easier to analyze. On the other hand, equation 8 is equivalent to setting the discount factor of reward to 0, which we call immediate reward as it is not affected by later step of learning. This is reasonable: the model can achieve a high reward in a given step, say the $t$-th step, even if it fails in later steps $t + 1, t+2, ...$ according to the recursive decomposition (Fig.1a). Hence we use equation 8 for training.
>
> 2. **Comment:** *"Is there any intuition for the critical gradient component? ... so that $\sigma$ is constant."*
>
>     **Response:** The name is because gradient can be completely determined by the critical gradient component and the attention score. Intuitively, it is a composite quantity that almost contains all the useful signal of the gradient, by measuring how the model's output is affected by the prior tokens (coming from $\psi'(\xi)$ in Eq.(9)) and how each prior token contributes to the true label of the current tokens (coming from $\phi_2(\cdot, \cdot)y_j^{(t - 1)}$ in Eq.(9)).
>
>     Below we write the expressions explicitly for specific Boolean functions (with $\psi$ and $\phi_2$ listed in Tab. 1), which in fact admit very simple forms.
>
>     When $W$ is 0, we have $\xi_{l^{(t)}} = \frac{1}{d_{t - 1}}\sum_{j = 1}^{d_{t - 1}} y_{j}^{(t - 1)}$.
>     Given $\mathbf{y}^{(t - 1)}$, for each of the Boolean function (we hide unimportant constants for brevity):
>    - $k$-PARITY:  $$
>             \gamma_{l^{(t)}}^{p}(\mathbf{y}^{(t - 1)}) \propto \left( \sum_{j = 1}^{d_{t-1}} y_{j}^{(t - 1)}\right) y_{i_1^{l^{(t)}}}^{(t - 1)} y_{i_2^{l^{(t)}}}^{(t - 1)} y_p^{(t - 1)}$$
>    - $k$-AND: $$
>             \gamma_{l^{(t)}}^{p}(\mathbf{y}^{(t - 1)}) \propto \mathbb{I}\left[ \left( \sum_{j = 1}^{d_{t-1}} y_{j}^{(t - 1)}\right) > 0\right] \left[ \left( y_{i_1^{l^{(t)}}}^{(t - 1)} + 1\right)\left( y_{i_2^{l^{(t)}}}^{(t - 1)} + 1\right) - 2\right] y_p^{(t - 1)} $$
>       where $\mathbb{I}(\cdot)$ is the indicator function.
>    - $k$-OR: $$
>              \gamma_{l^{(t)}}^{p}(\mathbf{y}^{(t - 1)}) \propto \mathbb{I}\left[ \left( \sum_{j = 1}^{d_{t-1}} y_{j}^{(t - 1)}\right) < 0\right] \left[ 2 - \left( 1 - y_{i_1^{l^{(t)}}}^{(t - 1)} \right)\left(1 -  y_{i_2^{l^{(t)}}}^{(t - 1)} \right)\right] y_p^{(t - 1)}$$.
>
> 3. **Comment:** *"In Proposition 3.1, why ... gradient descent?"*
>
>     **Response:** The variance bound w.r.t the target function intuitively measures the expected amount of signal about the target function in the gradient. If this variance is very small, then target signal contained in the gradient can be drowned out by noise. Hence, it is hard for the model to learn relying on gradient.
>
> 4. **Comment:** *"Why is the hinge loss important in section 3.2? ... gradient steps?"*
>
>     **Response:** Hinge loss is not necessary. We use it because its simple derivative will lead to a similar critical gradient component for SFT as in the analysis of RL. This makes the results for SFT cleaner. The results can be generalized to logistic loss with a slightly different critical gradient component (due to the additional derivative $\ell'(\hat{a}, a)$ brought by the logistic loss) following a very similar essence.

---

### Meta-Review · Area_Chair_EYP3 · 2026-01-05

**Summary:**

This paper gives an interesting theoretical analysis of learning sparse Boolean functions that can be decomposed into a binary tree of binary Boolean components. The reviewers raised a number of useful points and concerns.

Reviewer kGV3's review is the most technical. They point out that (1) the RL analysis is restricted to a one very large step of sign-GD, which may be overly simplistic; (2) the SFT setting does not use teacher forcing, which differs from the more common setup; (3) the hard-coded causal masks are part of the model design and may not be appropriately described as pretraining; (4) there is no sample complexity analysis; and (5) the paper lacks justification for requiring T steps of sign GD, where T is the depth of the computation tree. Reviewer 2mWM's comments focus mainly on the theory-practice gap, such as concerns about the use of population-level gradient analysis. Reviewer zXpU is mainly concerned with the simplified setting, in particular, the belief that the set of coordinates used by the sparse Boolean function is known to the transformer. Reviewer GHnk is mainly concerned that the theoretical setting is too similar to prior work by Kim & Suzuki (2025).

**Reviewer Concerns:**

The authors provide a reasonable rebuttal addressing many of these concerns. In response to Reviewer kGV3, the authors answer each point clearly; for example, one-step gradient descent learning is enough and multi-step analysis is fundamentally similar, and SFT without teacher forcing, while potentially subject to naming confusion, is not necessarily less natural. In response to Reviewer 2mWM, the authors highlight the distinction between theoretical studies of LLMs and practical large-scale LLM training. The authors argue that Reviewer zXpU's concerns are largely based on misunderstandings, clarifying that the model itself does not know the coordinates in B. Reviewer GHnk's concerns are the hardest to address, as the contribution does appear to rely heavily on prior work by Kim & Suzuki (2025).

**Reviewer Scores:**

Given the rebuttal, it seems plausible that the reviewer scores 4422 could improve to something like 6442 or higher. However, it is unlikely that the authors can fundamentally change Reviewer GHnk's assessment. I believe the authors can still use the feedback to substantially revise the paper and submit it to a future venue. At the same time, I agree that an average score of 3 for this paper is overly harsh. If I were a reviewer, I would likely give a borderline score.

---

### Decision · Program_Chairs · 2026-01-26

Reject